# Organizing Unstructured Image Collections using Natural Language

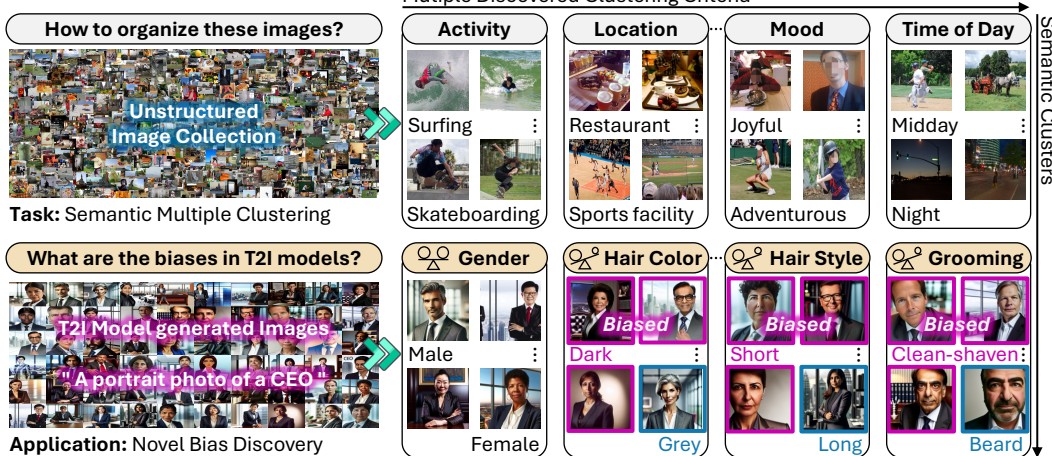

Figure 1: **Top: Semantic Multiple Clustering (SMC)** deals with automatically organizing an unstructured image collection into semantically meaningful and human interpretable groups (or semantic clusters), under multiple shared themes (or clustering criteria), without requiring any prior information. **Bottom:** Our proposed SMC system enables various applications like discovering novel biases in text-to-image generative (T2I) models.

## Abstract

Organizing unstructured visual data into semantic clusters is a key challenge in computer vision. Traditional deep clustering (DC) approaches focus on a single partition of data, while multiple clustering (MC) methods address this limitation by uncovering distinct clustering solutions. The rise of large language models (LLMs) and multimodal LLMs (MLLMs) has enhanced MC by allowing users to define clustering criteria in natural language. However, manually specifying criteria for large datasets is impractical. In this work, we introduce the task Semantic Multiple Clustering (SMC) that aims to automatically discover clustering criteria from large image collections, uncovering interpretable substructures *without requiring human input*. Our framework, **T**ext **D**riv**e**n **S**emantic Multiple **C**lustering (TeDeSC), uses text as a proxy to concurrently reason over large image collections, discover partitioning criteria, expressed in natural language, and reveal semantic substructures. To evaluate TeDeSC, we introduce the COCO-4c and Food-4c benchmarks, each containing four grouping criteria and ground-truth annotations. We apply TeDeSC to various applications, such as discovering biases and analyzing social media image popularity, demonstrating its utility as a tool for automatically organizing image collections and revealing novel insights.

**Disclaimer:** Potentially sensitive content.

## 1 Introduction

Organizing large volumes of unstructured visual data into semantic clusters is a crucial problem in computer vision and has traditionally been addressed through the lens of unsupervised deep clustering (DC) (Caron et al., 2018; Van Gansbeke et al., 2020). Despite the advancements and improved performance, DC remains limited due to its inherently ill-posed nature (Estivill-Castro, 2002) as it assumes a single clustering structure. For instance, a dataset can often be clustered in multiple ways (*e.g.*, a dataset of a deck of cards can be grouped by *suit* or *rank*). What forms a clustering depends on the needs of the users. Furthermore, the clusters discovered by DC rely on factors such as the inductive biases of the network, augmentations, and pretext tasks; and do not necessarily adhere to a particular partitioning criteria that a user may have in mind (*e.g.*, *suit*).

The limitation of DC in exploring visual data only along a single partition is addressed by multiple clustering (MC) methods (Yu et al., 2024), which uncover distinct clustering solutions. Multiple partitions reveal hidden patterns in the data and allow for analysis from different perspectives. The recent advent of large language models (LLMs) and multimodal LLMs (MLLMs) has further enhanced MC by enabling *users to specify* the clustering criterion (*e.g.*, group by `suit`) in natural language (Kwon et al., 2024; Yao et al., 2024). While incorporating a human in the loop is appealing, as it grants the user control over the clustering results, specifying a meaningful text criterion still requires *prior understanding* of the *entire* image collections. This task becomes cumbersome, especially due to the proliferation and high scene complexity inherent in large volumes of visual data.

In this work, we propose to ease the burden of specifying text criteria off the user, and introduce the task of *Semantic Multiple Clustering (SMC)*. As shown in Fig. 1(top), given an unstructured image collection, SMC aims to comprehensively *discover* all clustering criteria and their corresponding semantic clusters in natural language, entirely *without* requiring human priors. Tab. 1 outlines key distinctions between SMC and existing clustering solutions, with further discussions provided in App. A. SMC presents unique challenges to current machine learning systems. First, it requires reasoning over all provided images *concurrently*

Table 1: **Overview of clustering solutions and their key differences**. Unlike Deep Clustering (Caron et al., 2018) and Multiple Clustering methods (IC|TC (Kwon et al., 2024) and MMaP (Yao et al., 2024)), the proposed TeDeSC (Ours) requires *no* auxiliary prior knowledge, while offering interpretable outputs.

|  |  | DC | MMaP | IC\|TC | Ours |
|---|---|---|---|---|---|
| Prior | User-provided Criteria | ✗ | ✓ | ✓ | ✗ |
|  | Knowledge # Clusters | ✓ | ✓ | ✓ | ✗ |
| Out. | Multiple Clustering | ✗ | ✓ | ✓ | ✓ |
|  | Interpretability | ✗ | ✗ | ✓ | ✓ |

to identify valid clustering criteria. This capability is beyond the reach of existing vision-and-language models, which cannot effectively process thousands of images simultaneously. Second, SMC does not assume prior knowledge of user-preferred clustering granularity (*i.e.*, the number of clusters). Instead, it should adaptively determine the appropriate clustering structure for each discovered criterion, ensuring that the resulting clusters align with the underlying semantic substructure of the data.

To tackle SMC, we propose a general two-stage framework **Te**xt **D**riven **S**emantic Multiple **C**lustering (**TeDeSC**), powered by cutting-edge MLLMs and LLMs, that first *discovers* latent criteria (*e.g.*, `Activity` and `Location`) from a given collection of images, and then *groups* the images into semantic clusters (*e.g.*, "Surfing","Skateboarding" under `Activity`) for every discovered clustering criterion. To reason at a dataset level, TeDeSC first translates the visual data into textual data (*e.g.*, captions), which in turn are then collectively used as a proxy for the LLM to discover the hidden patterns in the image collection. The uniqueness of TeDeSC lies in its: (i) *comprehensiveness*, as it can exhaustively discover multiple clustering criteria that may not be evident to a user; (ii) *interpretability*, as it outputs cluster names in natural language as opposed to traditional MC methods; (iii) *flexibility*, as it can discover clusters at multiple granularities (coarse to fine-grained), without needing to specify the number of clusters a priori; and (iv) *generality*, as, unlike the existing work (Yao et al., 2024), it is not limited to object-centric images but can handle complex scenes having fine-grained details.

Current benchmarks for evaluating multiple clustering either lack realism (Clevr-4c, Fruit-2c, Card-2c) or offer a limited number of criteria (Action-3c), as shown in Fig. 2. To advance research in SMC, we introduce two challenging new benchmarks, COCO-4c and Food-4c, which depict images in daily life contexts and allow the dataset to be clustered as per four distinct criteria. Extensive experimental analyses on both existing and novel benchmarks demonstrate that the proposed TeDeSC can effectively discover meaningful clustering criteria and successfully group semantically similar images (some examples of clustering results by TeDeSC are shown in Fig. 1 (top)).

Lastly, we apply TeDeSC to a variety of applications, including discovering biases in real-world datasets and text-to-image generative models, as well as analyzing the popularity of social media images. For example, as shown in Fig. 1(bottom), TeDeSC can uncover less commonly studied biases (*e.g.*, `Hair color`) in DALL·E3-generated (Betker et al., 2023) images, beyond the well-known biases (*e.g.*, `Gender`) that may have already been corrected for. It achieves this by discovering various human-interpretable semantic clusters (*e.g.*, "Dark Hair", "Grey Hair") across different discovered criteria and identifying overpopulated clusters. Similarly, it can identify the semantic visual elements that contribute to the popularity of social images, offering valuable insights to related practitioners. These results suggest that TeDeSC is an automatic, versatile, and highly practical tool, opening up numerous new application opportunities for future research and providing the potential to generate insights from unstructured visual data at scale across various domains.

## 2 SEMANTIC MULTIPLE CLUSTERING

In this section, we first formally define the task of *Semantic Multiple Clustering (SMC)*, and then introduce the existing and newly proposed benchmarks for SMC.

**Task Definition:** Given a collection of $N$ unlabeled images $\mathcal{D} = \{\mathbf{x}_n\}_{n=1}^{N}$, the goal of SMC is to categorize $\mathcal{D}$ into semantic clusters based on multiple criteria $\mathcal{R} = \{R_l\}_{l=1}^{L}$, where $L$ is the total number of criteria. In detail, a criterion $R_l$ refers to the grouping *theme* according to which a set of images can be organized (or partitioned), such as Activity, Location, Mood, or Time of Day. All semantic clusters $\mathcal{C}_l$ under a given criterion $R_l$ should align with the theme of that criterion. For instance, the semantic clusters under the criterion Activity should reflect activities depicted in the images (*e.g.*, "Kayaking", "Cooking"). The same dataset $\mathcal{D}$, when grouped by a different criterion such as Location, could be clustered into categories like "Outdoor", "Indoor", and so on.

Unlike IC|TC (Kwon et al., 2024) and MMaP (Yao et al., 2024), where a human operator pre-defines the clustering criteria and the number of clusters $K_l$, both the criteria names and the cluster counts are *unknown* in SMC. *A SMC framework must automatically discover both the criteria and the corresponding clusters from $\mathcal{D}$.* In contrast to classic MC setting, SMC not only *discovers* both the criteria and cluster names but also *describes* them in natural language, making the discovered substructures human-interpretable.

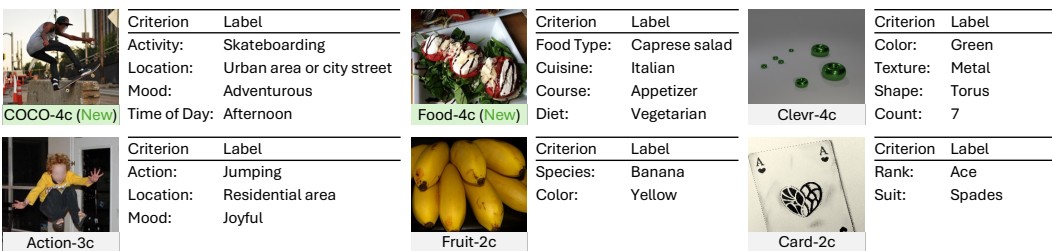

Figure 2: **Overview of the SMC benchmarks.** We show all clustering criteria and the corresponding ground-truth labels for the example images. We introduce two new challenging benchmarks: COCO-4c and Food-4c.

**Benchmarks:** Evaluating SMC methods requires benchmarks that can be partitioned under multiple valid criteria. Currently, only a few benchmarks (Yu et al., 2024) support the evaluation of SMC methods: Fruit-2c (Muresan & Oltean, 2018), Card-2c (Kaggle, 2022), Action-3c (Kwon et al., 2024), and Clevr-4c (Vaze et al., 2024). As shown in Fig. 2, these benchmarks are limited by their object-centric nature with simple backgrounds (*e.g.*, Fruit-2c), an insufficient number of criteria (*e.g.*, up to three in Action-3c), and a lack of photorealism due to synthetic generation (*e.g.*, Clevr-4c).

Given that the data encountered in real-world applications is more complex, we propose two *new* benchmarks for SMC: Food-4c and COCO-4c. Food-4c is sourced from Food-101 (Bossard et al., 2014), which includes 101 Food type (original annotations), along with new annotations for 15 Cuisine types, 5 Courses types, and 4 Diet preferences, totaling *four* clustering criteria. Additionally, we constructed COCO-4c using images from COCO-val (Lin et al., 2014), where we annotated *four* criteria with varying number of clusters: 64 Activity, 19 Location, 20 Mood, and 6 Time of day. Examples of these newly constructed benchmarks are shown in Fig. 2. Further details, including the full list of cluster names and the annotation pipeline, are provided in App. B.

## 3 METHODOLOGY: TEXT DRIVEN SEMANTIC MULTIPLE CLUSTERING

Partitioning an unstructured image collection into semantic clusters based on different (unknown) criteria is challenging, as it requires reasoning over the visual contents of all the images *concurrently*. Specifically, the system needs to first find commonalities among the images for discovering the partitioning criterion (or theme) and then group the images into semantic clusters according to the discovered criterion. To tackle the challenging SMC task, we propose a two-stage framework, named Text Driven Semantic Multiple Clustering (TeDeSC), that significantly *deviates* from the commonly used technique of deep feature-based clustering, and instead uses *text descriptions as a proxy* to reason over the images and uncover hidden patterns in $\mathcal{D}$. We elaborate the key design differences between TeDeSC and related work in App. A.

Figure 3: **TeDeSC** is composed of a *Criteria Proposer* and a *Semantic Grouper*. (**Left**) The Proposer processes the entire image set to discover and propose diverse grouping criteria expressed in natural language, which are accumulated in the Criteria Pool. (**Middle**) The Grouper takes these proposed criteria from the pool, discovers semantic clusters linked to the criteria at three different granularity levels, and assigns images to their respective clusters. (**Right**) The discovered criteria reveal multiple substructures of the image set across different granularities by aggregating cluster assignments. We explored various design choices and set the best-performing method (marked ⭐) as the main configuration of TeDeSC. Click the hyperlink in the figure for prompt details.

**System overview:** As illustrated in Fig. 3, the proposed TeDeSC consists of two modules: Criteria Proposer and Semantic Grouper. The Criteria Proposer (or Proposer in short) processes the entire image set $\mathcal{D}$ to discover diverse common themes among the images and proposes grouping criteria $\mathcal{R}$ in natural language (*e.g.*, Location). Once the criteria are proposed, the Semantic Grouper (or Grouper in short) discovers distinct semantic clusters that adhere to each criterion $R_l$ at varying levels of semantic granularity and assigns images to their respective clusters (*e.g.*, "Climbing gym"). In this work, we explore various design choices for both the Proposer and Grouper, detailed in the following subsections. Full implementation details, including exact prompts, are provided in App. C.

## 3.1 CRITERIA PROPOSER

As shown in Fig. 3(left), the Proposer processes the input image collection to generate grouping criteria in natural language. The core of the Proposer design lies in its ability to *concurrently* analyze and reason over a large set of images. With this in mind, we explore three systematic approaches detailed below. We provide exact prompt details used in the proposer in App. C.1.

**Image-based Proposer:** We start with a baseline that leverages a state-of-the-art MLLM (Li et al., 2024) designed for multi-image reasoning out-of-the-box to directly infer the criteria given a set of images. To reason over a set of images, which is crucial for discovering the criterion, we stitch a batch of images into a $8 \times 8$ image grid and input it to the MLLM as a single image. We then prompt the MLLM to propose grouping criteria for the images in the grid. All the resulting criteria from all such image grids are accumulated in a criteria pool, denoted as $\widetilde{\mathcal{R}}$.

**Tag-based Proposer:** Next, we explore a *tag*-based approach that uses an open-vocabulary tagger (Radford et al., 2021) to assign 10 tags to each image in $\mathcal{D}$, using the vocabulary from Word-Net (Miller, 1995). These tags serve as semantic descriptors, translating the visual content of each image into textual form. We then gather all the assigned tags from the images, input them into a LLM, and prompt it to discover grouping criteria based on these tags.

**Caption-based Proposer (Main):** While image tags effectively capture certain visual semantics, we found that they predominantly reflect object-related content. To encompass a broader spectrum of visual information—such as *environmental settings* or *interactions*—we instead use a MLLM to generate captions for each image in $\mathcal{D}$. Descriptive captions provide a richer and more holistic semantic context. Staying within the 128k token limit of modern LLMs (Meta, 2024b; OpenAI, 2024), we feed a subset of captions into a LLM, which we prompt to elicit partitioning criteria. The criteria generated for each subset are accumulated in $\widetilde{\mathcal{R}}$. Experiments in Sec. 4.1 show that the Caption-based Proposer is the most effective; so we use it as our main method and consider the others as baselines.

**Criteria refinement:** Since the Proposers operate on subsets, the criteria accumulated in $\widetilde{\mathcal{R}}$ may include redundant or noisy entries. To address this, we input all initially proposed criteria from $\widetilde{\mathcal{R}}$ into a LLM (Meta, 2024b), prompting it to consolidate similar criteria and discard noisy ones. This step refines and updates the criteria pool into $\mathcal{R}$, which is ready to be used in the subsequent stage.

## 3.2 SEMANTIC GROUPER

The automatically discovered criteria $\mathcal{R}$ serve as thematic indicators for revealing the semantic substructures (or clusters) within the image set $\mathcal{D}$. To uncover these substructures, as shown in Fig. 3(right), the Grouper takes $\mathcal{D}$ and each criterion $R_l$ as inputs. It then discovers cluster names $\mathcal{C}_l = \{s_k^l\}_{k=1}^{K_l}$ conditioned on $R_l$ and assigns each image to its corresponding cluster. The core design of the Grouper focuses on *aligning* semantic substructure discovery with a specified partitioning criterion. Similar to the Proposer, we also explore three distinct approaches for the Grouper.

Furthermore, as clusters under a given criterion can be formed at varying semantic granularities, depending on user preferences, we have designed our Grouper to cluster $\mathcal{D}$ at three levels of granularity: coarse, middle, and fine-grained. This design enables TeDeSC to provide new insights into the data at different granularities. For example, for the criterion food `Cuisine`, TeDeSC can organize images at a coarse-grained continental level (*e.g.*, "European" or "Asian"), a middle-grained regional level (*e.g.*, "Mediterranean" or "Southeast Asian"), or a fine-grained national level (*e.g.*, "Italian" or "Thai"). We provide the implementation details for the groupers and their multi-granularity design in App. C.2.

**Image-based Grouper:** Given a target criterion $R_l \in \mathcal{R}$, we first prompt a LLM to generate a question $q_l$ specific to $R_l$-*i.e.*, for criterion `Mood` the generated question is: "What mood is conveyed by this image? Answer with an abstract, common, and specific category name, respectively." We then use $q_l$ to guide a visual question answering (VQA) model to infer semantic cluster names for each image as $\left(s_{\text{coarse}}^l, s_{\text{mid}}^l, s_{\text{fine}}^l\right) = \text{VQA}(\mathbf{x}_n, q_l)$. Consequently, by aggregating the cluster assignment results at each granularity level across $\mathcal{D}$, we can derive multi-granularity semantic substructures.

**Tag-based Grouper:** Given a criterion, we prompt a LLM to generate a list of common, middle-grained tags specific to that criterion (*e.g.*, "Recreational facility"). Following Liu et al. (2024d), we then obtain coarse- and fine-grained tags by querying the LLM to generate super- and sub-categories (*e.g.*, "Indoor" and "Climbing gym") for each middle-grained tag. Unlike lexical databases such as WordNet or ConceptNet (Speer et al., 2017), which do not support free-form input and may be limited in accommodating certain criteria, tag synthesis using a LLM provides a flexible and reliable alternative. Subsequently, we use an image tagger to assign the most relevant tag from the candidate tags at each granularity to each image, resulting in multi-granularity substructures after aggregation.

**Caption-based Grouper (Main):** We prompt a MLLM to generate captions that specifically focus on the target criterion for each image, as $e_n^l = \text{MLLM}(\mathbf{x}_n, R_l)$. Next, we use a LLM in a three-step process to assign images to clusters at multiple granularity levels: *i) Initial Naming*: First, we prompt the LLM to assign a class name to each caption as $s_n^l = \text{LLM}(e_n^l, R_l)$, resulting in an initial set of names $\mathcal{S}_{\text{init}}^l$ of size $N$; *ii) Multi-granularity Cluster Refinement*: Using these initial names as basis, we prompt the LLM to refine them into three structured granularity levels: $\left(\mathcal{S}_{\text{coarse}}^l, \mathcal{S}_{\text{mid}}^l, \mathcal{S}_{\text{fine}}^l\right) = \text{LLM}(\mathcal{S}_{\text{init}}^l, R_l)$. These structured names serve as candidates for cluster assignment; *iii) Final Assignment*: Each image is then assigned a class name from each granularity level based on its caption, as $\left(s_{\text{coarse}}^l, s_{\text{mid}}^l, s_{\text{fine}}^l\right) = \text{LLM}(e_n^l, \mathcal{S}_{\text{coarse}}, \mathcal{S}_{\text{mid}}, \mathcal{S}_{\text{fine}})$. Experiments in Sec. 4.2 show that the Caption-based Grouper performs the best; thus we use it as our main method.

## 4 EXPERIMENTS

**Implementation details:** We run with our proposed TeDeSC framework using: *i)* CLIP ViT-L/14 (Radford et al., 2021) as the Tagger, *ii)* LLaVA-NeXT-7B (Liu et al., 2024b) as the MLLM, *iii)* Llama-3.1-8B (Meta, 2024b) as the LLM, and *iv)* BLIP-2 Flan-T5$_{\text{XXL}}$ (Li et al., 2023a) as the VQA model. For the Image-based Proposer we use LLaVA-NeXT-Interleave-7B (Li et al., 2024) as the MLLM due to its strong multi-image reasoning capability. Additionally, we explore a variant of the Image-based Grouper using LLaVA-NeXT-7B as the VQA model. Further implementation details, including exact prompts, are provided in App. C.

**Evaluation metrics for criteria discovery:** We asses the quality of the criteria $\mathcal{R}$ discovered by the Proposer from two dimensions: *i) Comprehensiveness:* We compute True Positive Rate (TPR) (Csurka et al., 2024) for the predicted criteria relative to the annotated ones as TPR $= \frac{|\mathcal{R} \cap \mathcal{Y}|}{|\mathcal{Y}|}$, to assess to what extent the predicted set covers the ground-truth set $\mathcal{Y}$; *ii) Diversity:* We compute the pairwise semantic similarity between the predicted criteria $\mathcal{R}$, using Sentence-BERT (Reimers & Gurevych, 2019), and convert it into a diversity measure by subtracting the similarity from 1. The final score is

the average of these pairwise values, reflecting how well the proposed criteria avoid redundancy and capture distinct aspects of the data—*i.e.*, criteria such as `Location` and `Place` are nearly identical and provide little additional insight. A higher score indicates better diversity. It is important to note that the number of valid grouping criteria is subjective and potentially unlimited, making False Positives difficult to define. Thus, we use TPR as the primary evaluation metric.

**Evaluation metrics for substructure uncovering:** Following Conti et al. (2023) and Liu et al. (2024e), for each criterion and its substructure uncovered by the Grouper, we evaluate its alignment with the substructure defined by the ground-truth labels along two dimensions: *i) Semantic Consistency:* For each image $x_n \in \mathcal{D}$, we compute the semantic similarity between its assigned cluster name $p_l \in \mathcal{P}_l$ and the ground-truth label $c_l \in \mathcal{C}_l$ under the current criterion $R_l$ as $\langle \mathcal{E}(p_l), \mathcal{E}(c_l) \rangle$, where $\mathcal{E}$ is the Sentence-BERT encoder and $\langle \cdot, \cdot \rangle$ represents the cosine similarity function. The average similarity across the dataset is reported as Semantic Accuracy (SAcc), reflecting how well the predicted substructure semantically aligns with the ground-truth. *ii) Structural Consistency:* We compute the clustering accuracy (CAcc) using the Hungarian algorithm that finds the optimal permutation between the ground-truth label and clustering assignment of each image (Han et al., 2021). CAcc and SAcc complement each other in evaluating the overall semantic clustering quality.

Since we do not process ground-truth annotations at different levels of granularity – coarse, medium and fine – we choose the substructure level that achieves the highest CAcc as the final predictions of the model. We deem this evaluation strategy as fair when compared to existing methods (Kwon et al., 2024; Yao et al., 2024) that rely on the knowledge of the number of ground-truth clusters.

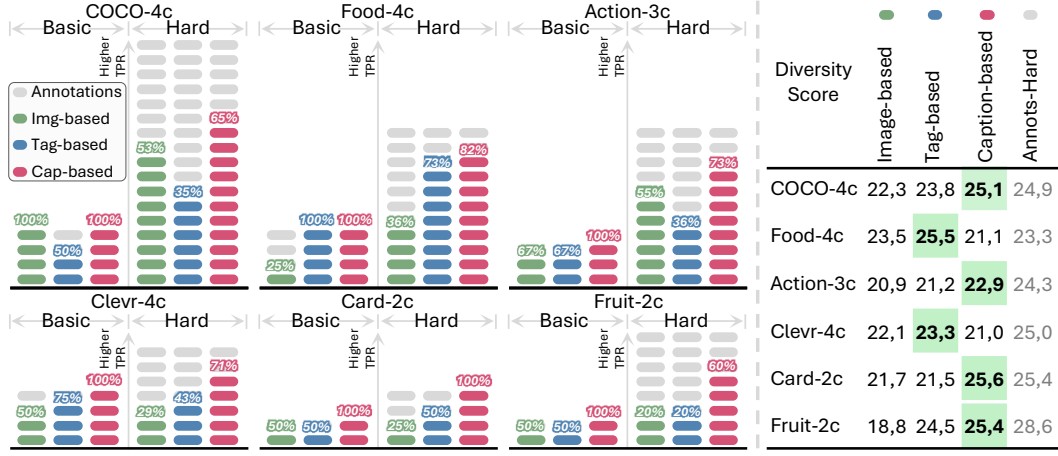

(a) Comprehensiveness Comparison: True Positive Rate (TPR)      (b) Diversity Comparison

Figure 4: **Comparison of Criteria Proposers. (a) Comprehensiveness:** TPR performance of each proposer is evaluated against Basic and Hard ground-truth criteria and visualized using a Progress Bar Chart. Each block represents one ground-truth criterion, with **Colored** blocks indicating successfully discovered criteria and **Gray** blocks representing undiscovered criteria. **(b) Diversity:** We report the Diversity Score measured from the criteria discovered by each proposer. Best diversity is highlight in **Green**. See expanded results in App. E.1.

### 4.1 STUDY OF THE CRITERIA PROPOSER

**Expanding ground-truth criteria for comprehensive evaluation:** For complex image collections like COCO-4c, four ground-truth criteria may not cover all valid grouping options. To address this, we expanded the ground-truth criteria for each of the six benchmarks in Sec. 2 using human annotators, resulting in {10, 4, 11, 7, 17, 11} distinct criteria for {Fruit-2c, Card-2c, Action-3c, Clevr-4c, COCO-4c, Food-4c}. We refer to the original per-image annotated criteria set (see Fig. 2) as **Basic** ground truth and the expanded set as **Hard** during evaluation. See App. B.2 for annotations.

**Which Criteria Proposer is the best?** In Fig. 4(a-b) we compare different variants of the Proposer along the dimensions of comprehensiveness and diversity. From Fig. 4(a) we observe that in terms of comprehensiveness the Caption-based Proposer consistently outperforms its counterparts in both the Hard set and Basic set across all six benchmarks. Its superior performance is particularly evident under the Hard criteria set, where it surpasses the second-best Tag-based Proposer by +32.2% TPR. Intuitively, the Caption-based Proposer works better because captions capture more diverse and

nuanced aspects of the image set, which further guides the LLM to comprehensively discover different grouping criteria. Contrarily, the Tag-based Proposer is less effective in complex benchmarks (*e.g.*, COCO-4c and Action-3c) since tags provide less contextual and descriptive information. Similarly, the Image-based Proposer is subpar in terms of performance since it is limited to reasoning over a small subset of images and loses visual details when combining images into a grid.

In Fig. 4(b) we notice similar trends for the diversity metric, where the Caption-based Proposer shows greater diversity on average across all the benchmarks when compared with the other two counterparts. Interestingly, the Tag-based Proposer works the best in the object centric benchmarks, such as Clevr-4c and Food-4c, since the foreground objects convey the bulk of the information.

**Studying the influence of image quantity on Criteria Discovery:** In Fig. 5 we show the TPR performance of the caption-based proposer across varying image scales used for criteria proposing, tested on the Hard criteria sets of six benchmarks. Interestingly, satisfactory performance is achieved with just a few images in *object-centric* benchmarks like Card-2c and Clevr-4c. In fact, even a single image often provides sufficient information for reasonable criteria discovery in these object-centric datasets, where a brief glimpse often represents the entirety adequately. For example, seeing one playing card allows the proposer to easily suggest criteria like "Rank" and "Suit" in the Card-2c dataset. However, this does not hold for more complex datasets like COCO-4c, Food-4c, and Action-3c, which contain diverse and realistic scenarios. Here, a reduction in image scale leads to a clear drop in TPR performance, as these datasets require a larger set of images to capture their intricate and varied thematic criteria. Since TeDeSC operates *without* prior knowledge of the dataset, we default to using the *entire* dataset to ensure comprehensive criteria discovery.

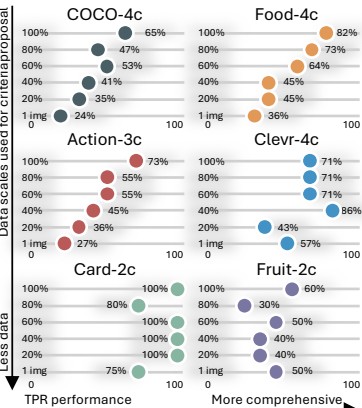

Figure 5: **Impact of image quantity on criteria discovery.** TPR of the *caption-based* proposer is reported for Hard ground-truth criteria set across varying image scales used for discovery

## 4.2 STUDY OF THE SEMANTIC GROUPER

**Which Semantic Grouper is the best?** We evaluate this by using the Harmonic Mean (HM) of CAcc and SAcc under each criterion, as these metrics complement each other. To provide context for our framework's performance, we establish a *pseudo* upper-bound reference using CLIP ViT-L14 in a zero-shot classification setup with ground-truth labels, where grouping criteria, cluster names, and the number of clusters are *all known*. Additionally, we use K-means with ground-truth cluster numbers and visual features from CLIP ViT-L14, DINOv1-B/16 (Li et al., 2022), and DINOv2-G/14 (Oquab et al., 2023) as baselines for clustering performance.

From Fig. 6, we observe that *the Caption-based Grouper performs best*, ranking first in 12 out of 19 tested criteria based on the HM across six benchmarks. It achieves an average CAcc of 59.9%, closely matching the pseudo upper-bound of 58.1% highlighting the effectiveness of our text-driven approach. For SAcc, the Caption-based Grouper achieves an average of 60.5%, surpassing its counterparts but falling short of the upper-bound 74.2%, which benefits from exact ground-truth class names. This gap is expected due to the open nature of the semantic space—*e.g.*,, terms like "Joyful," "Happy," and "Cheerful" often describe the same Mood but lack full semantic equivalence. The BLIP-2 image-based grouper ranks first in 5 out of 19 criteria. Its criterion-customized prompts help label visual content accurately, though its per-image labeling can lead to noisy clusters. In contrast, the tag-based grouper lags across all benchmarks, likely due to mismatches between generated tags and dataset concepts.

**Comparison with criterion-conditioned clustering methods:** We compare our top-performing Caption-based Grouper with two recent text-conditioned clustering methods: IC|TC (Kwon et al., 2024), which clusters images using LLaVA (Liu et al., 2024c) and GPT-4 (OpenAI, 2023) based on user-specified criteria, and MMaP (Yao et al., 2024), which generates pseudo prototypes with GPT-4 for user-specified criteria, then applies prompt-tuned CLIP (Radford et al., 2021) and KMeans clustering. Note that *both IC|TC and MMaP require user-provided (ground-truth) criteria and the number of clusters ($K_l$) as auxiliary prior input to work.* In stark contrast, our grouper uses the *criteria discovered* by the proposer and requires *no* pre-set cluster counts to forge high-quality clusters, operating entirely automatically. As shown in Tab. 2, our Caption-based Grouper outperforms

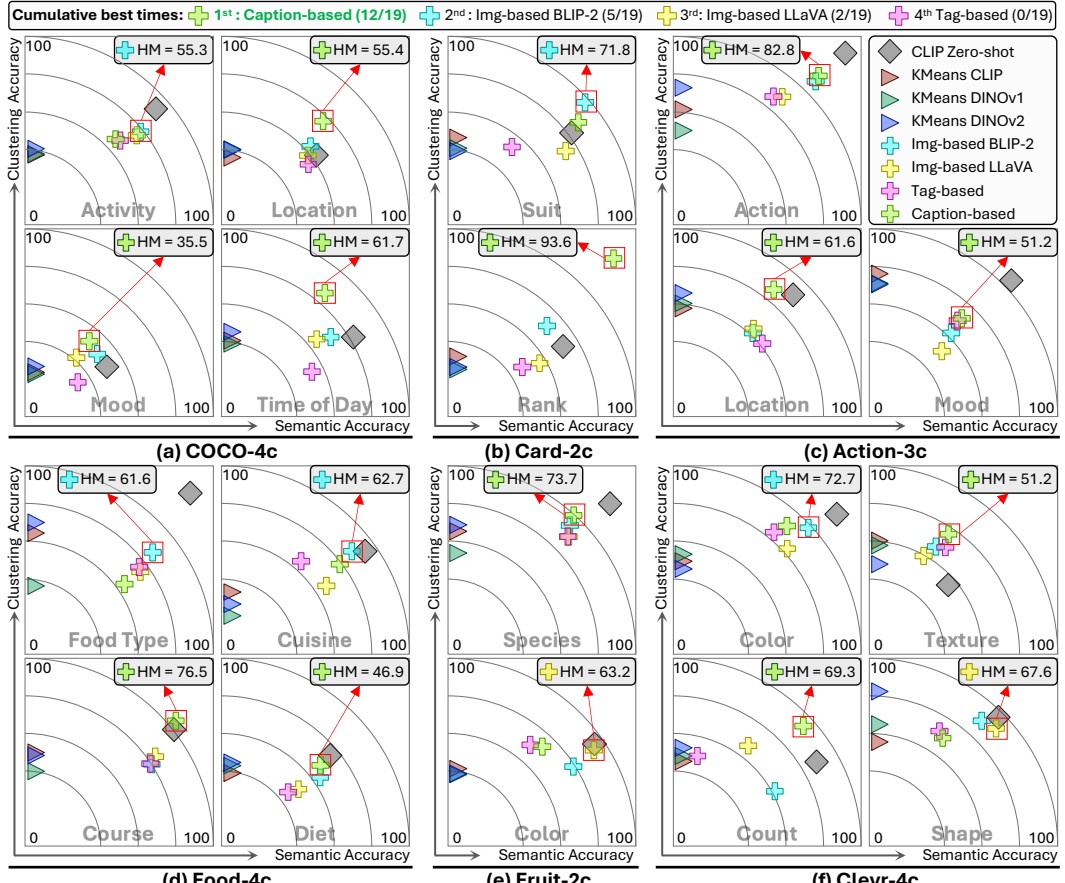

Figure 6: **Comparison of the Semantic Groupers.** CAcc, SAcc, and their Harmonic Mean (HM) scores are reported for different Semantic Groupers (⊕) on the basic criteria across six benchmarks. CLIP zero-shot classification (◇) performance is included as a *pseudo* upper bound, while KMeans (▷) using various strong visual features is provided as a baseline for CAcc. See expanded results in App. E.2.

Table 2: **Comparison with criterion-conditioned clustering methods.** For each benchmark, we report the average CAcc (%) and SAcc (%) across all criteria. We provide CLIP ViT-L/14 zero-shot performance as the pseudo upper-bound reference (UB). See expanded per-criterion results in App. E.3.

| | COCO-4c | | Food-4c | | Clevr-4c | | Action-3c | | Card-2c | | Fruit-2c | | Avg | |
| | CAcc | SAcc | CAcc | SAcc | CAcc | SAcc | CAcc | SAcc | CAcc | SAcc | CAcc | SAcc | CAcc | SAcc |
|---|---|---|---|---|---|---|---|---|---|---|---|---|---|---|
| UB | 40.1 | 60.6 | 64.1 | 80.2 | 56.7 | 72.5 | 79.8 | 82.3 | 41.4 | 66.9 | 69.4 | 88.3 | 50.2 | 64.4 |
| IC\|TC | 48.9 | **53.2** | **50.5** | 61.7 | 58.3 | 36.8 | **76.4** | 56.3 | **74.8** | 81.2 | 63.3 | 55.1 | **53.1** | 49.2 |
| MMaP | 33.9 | - | 43.8 | - | 62.8 | - | 60.6 | - | 36.9 | - | 51.0 | - | 41.3 | - |
| TeDeSC (Ours) | **51.2** | 48.4 | 48.1 | **64.9** | **64.9** | **54.3** | 68.3 | **60.6** | 73.3 | **84.3** | **65.1** | **61.1** | 53.0 | **53.4** |

MMaP and delivers results comparable to IC|TC across six benchmarks. This demonstrates that our framework achieves high-quality clusters for the SMC task *without* requiring users to pre-define criteria or cluster counts. Implementation details for IC|TC and MMaP are provided in App. D.

**Necessity of multi-granularity cluster refinement:** To validate the effectiveness of the multi-granularity cluster refinement design, we design control experiments with our Caption-based Grouper, using three different methods for constructing cluster names to organize images: *i) Initial Names:* using the initially assigned names as the final output, *ii) Flat Refinement:* prompting the LLM to refine the initial names into a flat list with a unified granularity, and our *iii) Multi-*

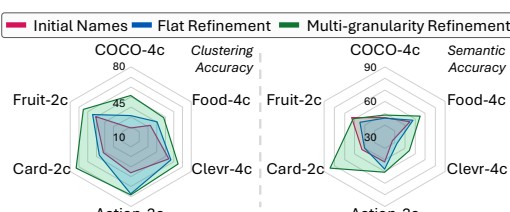

Figure 7: **Ablation study of multi-granularity refinement.** See expanded results in App. E.4.

*granularity Refinement*. We then compare the performance of these approaches in Fig. 7. We observe that both refinement methods significantly improve clustering accuracy compared to using noisy initial names, highlighting the importance of having granularity-consistent cluster names for accurately revealing substructures. Additionally, our proposed multi-granularity method surpasses flat refinement, as it enables the grouper to generate clustering results at different granularities, providing greater flexibility in aligning with the label granularity-the user-preferred level of detail.

**Additional studies:** In App. F, we report studies on: *i) Fine-grained Scenario:* We examine how to integrate our framework with more advanced cross-modal Chain-of-Thought prompting strategies to better handle fine-grained criteria. *ii) Sensitivity Analysis:* We analyze the system sensitivity to various MLLMs and LLMs. Additionally, Sec. G offers a *Qualitative Analysis* by visualizing clustering results for different criteria, and Sec. H includes a *Failure Case Analysis*.

## 5 APPLICATIONS

**Discovering and mitigating dataset bias:** Given an image collection that contains *spurious correlations* (Geirhos et al., 2020), we are curious whether we can proactively find this issue caused by data bias directly from the training images *without* relying on either the annotations (Sagawa et al., 2020) or *post hoc* misclassified images (Kim et al., 2024). As a case study, we applied the proposed TeDeSC framework to the 162k training images of the CelebA (Liu et al., 2015) dataset—a binary hair color classification dataset where the target label "Blond" is spuriously correlated with the demographic attribute "Female" in its training split. Additional details are provided in App. I.1.

**Findings:** Our method successfully identified the grouping criteria `Hair color` and `Gender`. We then analyzed the predicted gender distributions within the "Blond" and "Not Blond" clusters. As shown in Fig. 8(a), the "Blond" cluster is highly skewed, with 86.5% of the images depicting females, closely aligning with the ground-truth distribution (94.3%). This confirms the spurious correlation between "Blond" and "Female". To further validate this observation, we followed B2T (Kim et al., 2024) by using the predicted distributions to train a debiased model with GroupDRO (Sagawa et al., 2020) and compared it with

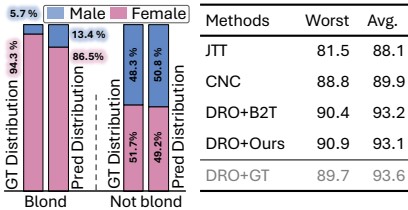

| Methods | Worst | Avg. |
|---------|-------|------|
| JTT | 81.5 | 88.1 |
| CNC | 88.8 | 89.9 |
| DRO+B2T | 90.4 | 93.2 |
| DRO+Ours | 90.9 | 93.1 |
| DRO+GT | 89.7 | 93.6 |

(a) Bias confirmation  (b) Debiasing results

Figure 8: **Results of dataset bias discovery and mitigation.** Worst group and average accuracies(%) are reported.

other unsupervised bias mitigation methods (JTT (Liu et al., 2021), CNC (Zhang et al., 2022), B2T, and GroupDRO with ground-truth labels). As shown in Fig. 8(b), our model achieved robust performance comparable to B2T, demonstrating the reliability of the discovered distributions.

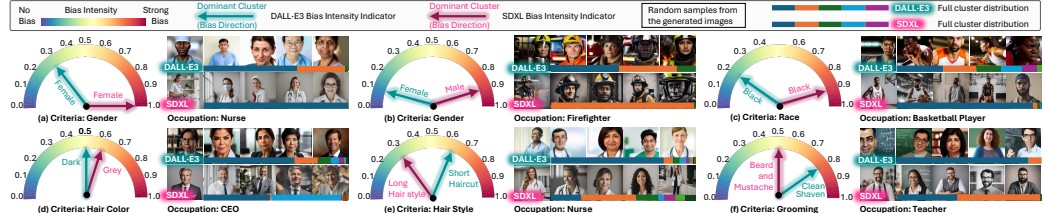

Figure 9: **Bias discovery results in T2I models.** Bias evaluation results are shown for the associated occupations.

**Discovering Novel Bias in Text-to-Image Diffusion Models:** Stereotypical biases related to gender or race (Naik & Nushi, 2023) in images generated by text-to-image (T2I) models like Stable Diffusion XL (SDXL) (Podell et al., 2024) and DALL·E3 (Betker et al., 2023) have been widely studied (Nicoletti & Bass, 2023; Bianchi et al., 2023). However, we ask: *Are there other biases present in T2I-generated images?* To explore this, we selected *nine* occupations (*e.g.*,, Nurse, CEO) from prior studies (Bianchi et al., 2023; Bolukbasi et al., 2016) and generated 100 images for each using the prompt "A portrait photo of a <OCCUPATION>," resulting in 1.8k images from both DALL-E3 and SDXL (see Fig. 21 and Fig. 22). Using TeDeSC, we automatically discovered 10 grouping criteria (bias dimensions) and their predicted distributions for each occupation. To measure bias, we quantified the normalized entropy of each distribution (D'Incà et al., 2024) as bias intensity and identified the dominant cluster (with the most images) as the potential bias direction. We also conducted a user study with 54 participants to evaluate our findings. The system's bias intensity scores closely matched human ratings, with an Absolute Mean Error (AME) of 0.1396 (scale: 0 to 1),

and its predicted bias directions aligned with human evaluations 72.3% of the time. Key findings are summarized below, with full results in App. I.2.

**Findings:** In Fig. 9, we present key findings. Without predefined biases, our method identifies known social biases in occupations. For example, as shown in Fig. 9(a–c), SDXL-generated images display pronounced gender and racial imbalances for roles like Nurse, Firefighter, and Basketball Player, exceeding official statistics (U.S. Bureau of Labor Statistics, 2021). In contrast, DALL·E3 demonstrates improved bias mitigation, likely due to its "guardrails" (OpenAI, 2022b). More notably, as shown in Fig. 9(d–f), our method uncovers novel, previously unrecognized bias dimensions. For instance, SDXL strongly associates CEOs with "Grey" hair, while DALL·E3 favors "Dark" hair. Interestingly, DALL·E3 exhibits stronger biases than SDXL in Hair style and Grooming for occupations like Nurse (Fig. 9(e)) and Teacher (Fig. 9(f)). These findings suggest that industrial T2I models, even with guardrail systems, may address well-known biases while overlooking novel ones, emphasizing the need for broader bias examination.

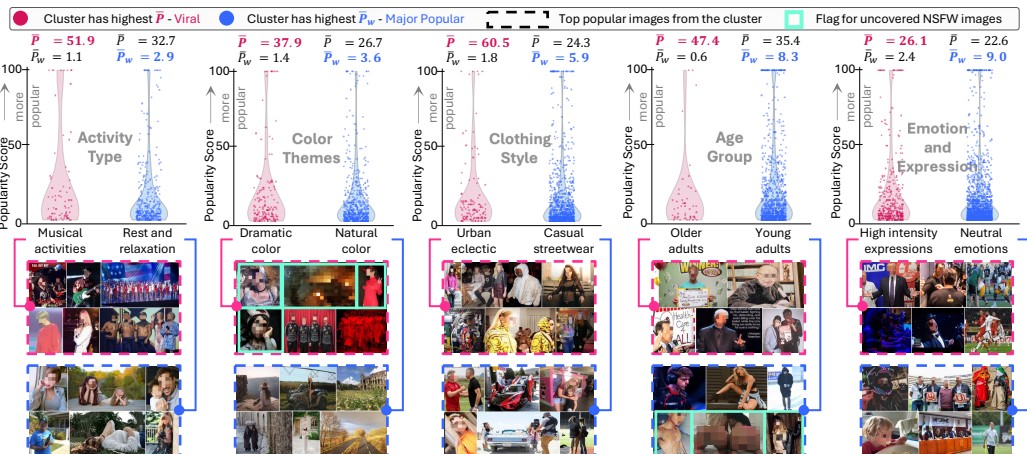

Figure 10: **Analysis of social media photo popularity on SPID dataset.** We show the *viral* and *major popular* clusters along with the popularity distribution of data points within these clusters across five criteria (in Grey).

**Analyzing Social Media Image Popularity:** *What visual elements make a photo popular?* To explore this, we applied TeDeSC to 4.1k Flickr photos from the SPID dataset (Ortis et al., 2019), each with a popularity score based on the number of views. Our method grouped the photos into semantic clusters based on 10 discovered criteria. For each cluster, we calculated: *i)* $\bar{P}$, the average popularity of photos in the cluster, and *ii)* $\bar{P}_w$, the weighted average popularity based on the cluster's proportion in the dataset. The cluster with the highest $\bar{P}$ is identified as *viral* (highly popular but few in number), while the one with the highest $\bar{P}_w$ is *major popular* (concentrating most of the general popular photos). Key findings are discussed below, with full results in App. I.3.

**Findings:** As shown in Fig. 10, combining our method's interpretable groupings with popularity scores reveals the visual elements driving virality (clusters with the highest $\bar{P}$) and the common traits of widely popular images (clusters with the highest $\bar{P}_w$). Interestingly, we observe that viral elements often sharply contrast with those of popular images, such as *Musical activities* vs. *Rest and relaxation*, or *High-intensity expressions* vs. *Neutral emotion*, suggesting that attention-grabbing visuals stand out due to their novelty or intensity, especially given today's short attention spans (McSpadden, 2015; Farid, 2024). Additionally, we unexpectedly found that some highly popular images in certain clusters contained *not safe for work (NSFW)* content, previously undiscovered in the SPID dataset. This underscores how provocative visuals can drive popularity and highlights the importance of thorough dataset inspection, where our framework proves valuable.

## 6  CONCLUSION

In this work, we introduce the task of semantic multiple clustering and propose TeDeSC, a system that automatically discovers grouping criteria in natural language from large image collections and uncovers interpretable data substructures based on these criteria. We rigorously evaluate various design choices of TeDeSC on four existing and two newly proposed benchmarks, and demonstrate its ability to reveal valuable insights that might otherwise go unnoticed in various real-world applications.

ETHICS STATEMENT

We do not anticipate any immediate negative societal impacts from our work. However, we encourage future researchers building on this work to remain vigilant, as we have, about the potential for TeDeSC, which integrates LLMs and MLLMs–particularly their human-like reasoning abilities– to be used both for good and for harm.

The motivation behind our studies on biases in existing datasets and text-to-image (T2I) generative model outputs, as well as our exploration of social media image popularity, is to *reveal and address* these biases and the presence of sensitive or not-safe-for-work (NSFW) content that objectively exist in the datasets and models. We emphasize that our aim is *to study and mitigate these issues*, and in doing so, we *do not create* any new biases or disturbing content. Specifically, in Sec. 5, we use well-established benchmarks, such as CelebA (Liu et al., 2015), for our study of dataset bias, and for bias discovery in T2I generative models, we select occupation-related subjects known to be associated with biases from prior studies (Bianchi et al., 2023; Bolukbasi et al., 2016). Furthermore, our framework reveals previously undisclosed sensitive and sexual content in the SPID dataset (Ortis et al., 2019). We responsibly present these findings in Sec. 5, applying significant blurring to disturbing content, with the intention of raising community awareness about the need to further scrutinize NSFW content in existing benchmarks. However, we acknowledge that our methodology and findings could potentially be misused by malicious actors to promote harmful narratives or discrimination against certain groups. We strongly oppose any such misuse or misrepresentation of our work. Our research is conducted with the aim of advancing technology while prioritizing public welfare and well-being.

For the creation of our two new benchmarks, COCO-4c and Food-4c, we sourced images exclusively from the COCO-val-2017 (Lin et al., 2014) and Food-101 (Bossard et al., 2014) datasets, strictly adhering to their licensing agreements. Additionally, we utilized voluntary human annotators for proposing valid grouping criteria and creating annotations along these criteria, rather than employing annotators from crowdsourcing platforms. This decision was made to ensure sustainability, fair compensation, and high-quality work, as well as to safeguard the psychological well-being of participants. Similarly, for our user study on T2I model bias evaluation, we recruited voluntary participants via questionnaires to collect human evaluation results. The user study was conducted entirely anonymously, with participants providing informed consent. Our project, including data annotation and the user study involving human subjects, was approved by the Ethical Review Board of our university.

Lastly, we emphasize that our proposed framework, TeDeSC, relies on open-source LLMs and MLLMs, allowing full deployment on local machines. We refrain from using APIs from industrial LLMs or MLLMs, both to ensure reproducibility and to protect data privacy.

REPRODUCIBILITY STATEMENT

We will *release all* essential resources required to reproduce the experimental results presented in this project, including source code, exact prompts, benchmarks with their data splits, and generated images, upon publication. Our proposed framework, TeDeSC, is built on *open-source*, *publicly accessible* models to ensure reproducibility. In Sec. 3, we provide a detailed description of how our framework is constructed. Additionally, App. C contains further implementation details, including exact prompts, to help practitioners easily reproduce our method. Details regarding the implementation of the compared methods are also provided in App. D. Moreover, we present extended numerical experimental results in App. E, alongside comprehensive findings for the application study in App. I. We believe that the thorough descriptions of our methodology, the extensive presentation of experimental results, and the open-source nature of our framework ensure that this work is highly reproducible, enabling future researchers and practitioners to readily apply our method to various domains.

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

APPENDIX

This appendix provides detailed supplementary information supporting the implementations, experiments, and findings presented in the paper. First, in App. A, we offer an extensive discussion of Related Work, covering tasks and methods pertinent to our study. In App. B, we describe the benchmarks used in our study, including the construction of the newly proposed COCO-4c and Food-4c datasets, along with the process for creating hard ground-truth criteria for evaluating different proposers. App. C outlines the prompts and implementation details for our proposed framework, including both Criteria Proposers and Semantic Groupers. App. D provides a deep dive into the implementation specifics of the compared methods used in our experiments. In App. E, we present additional quantitative results that supplement the findings reported in the main paper, covering evaluations of the Criteria Proposer, Semantic Grouper, and comparisons with other clustering methods. App. F extends our analysis of the proposed framework, including system sensitivity and studies on fine-grained image collections. App. G presents further qualitative results for the predicted clusters, while App. H offers a more detailed analysis of the failure cases encountered by our method. App. I provides additional findings, along with implementation details and user study results for the three applications explored in this work. App. J presents a deeper discussion of related settings pertinent to our research. App. K presents an analysis of the computational cost and runtime of the proposed TeDeSC framework. In App. L, we outline potential directions for future work. App. M offers an in-depth discussion on the impact of invalid criteria on system performance, while App. N addresses the limitations of this work. Additionally, App. O further investigates the effect of multi-granularity clustering output on image grouping. App. P provides insights into how LLMs can enhance image clustering, and App. Q includes a detailed discussion of the evaluation metrics employed in this work.

*We will release all essential resources for reproducing this work, including code, prompts, benchmarks, and annotations, upon publication.*

## A  RELATED WORK

**Deep Clustering.** Image clustering (Xu & Wunsch, 2005) discovers hidden grouping structures within large, unstructured, and unlabeled image collections, serving as a tool for various data-driven applications (Wazarkar & Keshavamurthy, 2018). To achieve this, deep clustering (DC) methods such as DEC (Xie et al., 2016) and SCAN (Van Gansbeke et al., 2020) focus on simultaneously learning feature representations and cluster assignments using deep neural networks via self-supervised techniques (Caron et al., 2020; Zhong et al., 2021; Ren et al., 2024). Furthermore, large-scale pre-trained feature representations like DINOv1 (Caron et al., 2021), DINOv2 (Oquab et al., 2023), and CLIP (Radford et al., 2021) have also been shown to be effective at clustering image collections in a zero-shot fashion with the help of KMeans (Vaze et al., 2022; Liu et al., 2024f; Han et al., 2023).

**Multiple Clustering.** However, it is well-known that "clusters are in the eye of the beholder" (Estivill-Castro, 2002); there often exist multiple ways to partition the same image collection into clusters, and what constitutes a cluster depends on the user's needs. This insight has led to the study of Multiple Clustering (MC) (Ren et al., 2022; Yao et al., 2023; Yu et al., 2024), which aims to simultaneously learn feature representations and cluster assignments from different perspectives to find various ways of grouping the same data, enabling alternative interpretations from different viewpoints. Early approaches primarily focused on discovering multiple clusterings directly within the original data space (Gondek & Hofmann, 2005). Building on these foundations, subsequent methods shifted toward uncovering multiple clusterings within subspaces (Qi & Davidson, 2009). Unlike traditional subspace clustering methods, which identify clusters in low-dimensional subspaces (Kriegel et al., 2009), subspace-based multiple clustering techniques explore distinct subspaces, each associated with unique, non-redundant clusterings (Wang et al., 2019). Although existing MC methods have achieved impressive results (Yao et al., 2023; 2024; Kwon et al., 2024) on some benchmarks, they share similar limitations with deep clustering approaches (Xie et al., 2016; Van Gansbeke et al., 2020). Their results require intensive *manual post-analysis*, and they also hold strong assumptions: users must specify *(i)* the number of potential clusterings and *(ii)* the number of clusters within each clustering. However, when dealing with millions of unstructured images, it is infeasible for users—who are trying to understand the data—to know this information *a priori*.

To address this challenge, recent works such as IC|TC (Kwon et al., 2024) and MMaP (Yao et al., 2024) propose a relaxed assumption: users may have certain criteria and corresponding cluster counts

in mind for grouping images. They leverage these user-provided priors as auxiliary information to generate multiple criteria-conditioned clusterings through the cooperation of large language models (*e.g.*, GPT-4 (OpenAI, 2023)), multimodal large language models (*e.g.*, LLaVA (Liu et al., 2024c)), or vision-language models (*e.g.*, CLIP (Radford et al., 2021)). However, as image collections—like those from social media platforms (Ortis et al., 2019)—continue to diversify, the complexity of the data structure grows. It is impractical to expect the user to specify grouping criteria for a large image collection that they are not familiar with. Moreover, relying solely on human-defined criteria limits our ability to discover novel patterns and insights that might otherwise remain unnoticed. Besides, data analysis tools such as REVISE (Wang et al., 2022) and Know Your Data (Google People + AI Research, 2021) also allow users to explore visual data through multiple dimensions. However, they require human annotations to function and are thus limited to existing annotated datasets only.

In stark contrast to prior work, we introduce and study the task of semantic multiple clustering (SMC). Instead of requiring users to specify the grouping criteria, SMC seeks to actively and automatically *discover* criteria expressed in natural language from large visual data and uncover the corresponding semantic substructures, *without* access to any of the aforementioned human priors. As demonstrated in Sec. 5, the flexibility provided by our proposed SMC framework adds significant value to various data-driven applications, unveiling novel insights about the data that might not have been noticed before.

**Topic Discovery.** The setting of semantic multiple clustering (SMC) is also related to the field of Topic Discovery (Blei et al., 2003; Wang et al., 2009; Eklund & Forsman, 2022) in natural language processing, which aims to identify textual themes from large *text corpora* (*e.g.*, documents). Our work shares motivational similarities with topic discovery because both tasks seek to find common, thematic concepts from large volumes of data. In contrast, our work focuses on discovering thematic criteria from large *visual content*. However, indeed, the core challenges of SMC and topic discovery are highly similar: they both require systems that can concurrently reason over large volumes of data. Nevertheless, SMC is an even more challenging task than topic discovery for two reasons: *i)* semantics are not explicitly expressed in images, whereas they are in text; *ii)* there is currently no vision model that can encode large sets of images and reliably reason over them. Thus, in this work, we translate images to text and use text as a proxy to elicit the large-scale reasoning capability of large language models (Meta, 2024b).

**Multimodal Large Language Model.** Recent advancements in multimodal large language models (MLLMs) have been driven by the availability of large-scale vision-language aligned training data. The typical paradigm (Liu et al., 2024c) involves using a pre-trained large language model (LLM) (Meta, 2024a; Chiang et al., 2023; Jiang et al., 2023; Meta, 2024b) alongside a pre-trained vision encoder (Radford et al., 2021). A projector is learned to align visual inputs with the LLM in the embedding space, which enhances visual understanding by utilizing the reasoning capabilities of LLMs. Several models have achieved significant success in zero-shot image captioning and visual question answering (VQA), including BLIP-2 (Li et al., 2023a), BLIP-3 (Xue et al., 2024), Kosmos-2 (Peng et al., 2023), and the LLaVA series (Liu et al., 2024c;b; Li et al., 2024). In our proposed TeDeSC framework, we employ MLLM primarily as a *zero-shot* image parser, converting visual information into text and using this text as a proxy to elicit LLMs for reasoning over large image collections and discovering grouping criteria. Additionally, we leverage the multi-image reasoning capability of LLaVA-NeXT-Interleave (Li et al., 2024) to establish a baseline image-based proposer for the SMC task, while utilizing BLIP-2 with customized prompts in a VQA style (Shao et al., 2023; Zhu et al., 2023) as the image-based grouper to form semantic clusters linked to specific visual content within the images.

**Large Language Model.** In the era of large language models (LLMs) advancement (Ouyang et al., 2022), modern LLMs, such as the Llama series (Touvron et al., 2023; Meta, 2024a;b), Vicuna (Chiang et al., 2023), Mistral-7B (Jiang et al., 2023), and the GPT series (Brown et al., 2020), have demonstrated remarkable zero-shot capabilities in tasks involving text analysis, completetion, generation, and summarization. With advanced prompting techniques like Chain-of-Thought (CoT) (Wei et al., 2022), the reasoning abilities of LLMs can be further enhanced. In the proposed TeDeSC framework, we design CoT prompts (see App. C) to harness the text generation and summarization capabilities of Llama-3.1 as a reasoning engine. This aids TeDeSC in several key areas: discovering grouping criteria from large sets of image captions, automatically prompting VQA models, generating criterion-specific tags, uncovering cluster semantics, and grouping images based on their captions.

Unlike prior works (Zhuge et al., 2023) that focus on set difference captioning (Dunlap et al., 2024), fine-grained concept discovery (Liu et al., 2024e), or video understanding (Wang et al., 2024b), we leverage LLMs to tackle the challenging semantic multiple clustering task. While IC|TC (Kwon et al., 2024) also uses the LLM (GPT-4 (OpenAI, 2023)) for grouping visual data, our proposed TeDeSC differs in two key aspects: *i)* TeDeSC does *not* require user-defined grouping criteria or the number of clusters, and *ii)* TeDeSC provides *multi-granularity* outputs to meet various user preferences.

**Text-Driven Image Retrieval.** Given a query text (*e.g.*,, "sofa" or "person wearing a blue T-shirt"), text-driven image retrieval methods (Karthik et al., 2024; Liu et al., 2023; Wu et al., 2023) aim to find images from an image collection that are relevant to the query. In other words, in the scenario we are considering, given the image collection and a list of text queries, one can organize images according to the text using text-driven image retrieval techniques. In this context, the query can be considered as a sort of "cluster name". However, this differs significantly from the proposed task of semantic multiple clustering (SMC), because SMC requires both discovering the textual criteria and the corresponding textual clusters. Thus, without knowing text queries as prior information, text-driven image retrieval methods are not able to accomplish SMC.

# B    BENCHMARK DETAILS

## B.1    BENCHMARK CONSTRUCTION OF COCO-4C AND FOOD-4C

Table 3: **Summary of number of classes for the basic criteria annotation across the six benchmarks.**

| Dataset | Number of Images | Basic Criterion | Number of Classes |
|---|---|---|---|
| COCO-4c | 5,000 | Activity | 64 |
| | | Location | 19 |
| | | Mood | 20 |
| | | Time of Day | 6 |
| Food-4c | 25,250 | Food Type | 101 |
| | | Cuisine | 15 |
| | | Course | 5 |
| | | Diet | 4 |
| Action-3c | 1,000 | Action | 40 |
| | | Location | 10 |
| | | Mood | 4 |
| Clevr-4c | 10,000 | Color | 10 |
| | | Texture | 10 |
| | | Shape | 10 |
| | | Count | 10 |
| Card-2c | 8,029 | Rank | 14 |
| | | Suit | 5 |
| Fruit-2c | 103 | Species | 34 |
| | | Color | 15 |

To create high-quality benchmarks for COCO-4c and Food-4c, we designed a four-step annotation pipeline:

**(1) Criteria Identification:** We first split COCO-val-2017 (Lin et al., 2014) and Food-101 (Bossard et al., 2014) images into batches of 100. Each batch was stitched into a 10×10 grid to form a single image. These grid images were then distributed to 5 human annotators, who were tasked with identifying grouping criteria. For each dataset, we selected the 4 most frequently occurring criteria, as shown in Tab. 3, to proceed with per-image annotation.

**(2) Label Candidate Generation:** To facilitate the annotation process, we used GPT-4V (OpenAI, 2023) to generate an initial list of candidate labels for each criterion. Specifically, for each criterion of COCO-4c and Food-4c, GPT-4V was prompted to assign a label that reflected the criterion for each image. This resulted in a list of criterion-specific label candidates for each dataset.

**(3) Image Annotation:** Next, 10 human annotators were tasked with assigning a label from the criterion-specific candidates to each image in COCO-4c and Food-4c for each criterion. The entire annotation process took 25 days to complete.

**(4) Label Merging:** Image annotation is inherently subjective, with annotators potentially assigning different labels for the same criterion. For example, one annotator might label the `Mood` criterion as "Happy", while another might label it as "Joyful" or "Delightful". To resolve such discrepancies, we used majority voting to determine the final label for each image. Specifically, the most frequently assigned label among the 10 annotators was chosen as the final label for each criterion.

Following these steps, we constructed COCO-4c and Food-4c. *Note that we used the official COCO-val-2017 (Lin et al., 2014) and Food-101 (Bossard et al., 2014) images for our benchmarks and did not collect any new images. We adhered strictly to the licenses of the datasets during their creation.* The exact number of classes is presented in Tab. 3. Additionally, the annotated class names for each criterion of COCO-4c are provided in Tab. 4, and for Food-4c in Tab. 5.

Table 4: **Full class names for COCO-4c across the four basic criteria.**

| Criterion | COCO-4c |
|---|---|
| Activity | "repairing a toilet", "playing volleyball", "playing guitar", "haircutting", "cutting a cigar", "kayaking", "applauding", "tying a tie", "playing basketball", "washing dishes", "gardening", "texting messages", "repairing a car", "peeing", "cleaning the floor", "writing on a book", "feeding a horse", "singing", "baking", "hiking", "smoking", "riding an elephant", "pouring liquid", "waving hands", "swimming", "meditating", "fixing a bike", "cutting vegetables", "walking a dog", "reading a book", "celebrating", "queuing", "cutting a cake", "brushing teeth", "playing soccer", "jumping", "snowboarding", "playing", "touching animals", "pushing a cart", "watching tv", "rowing a boat", "taking photos", "running", "flying a kite", "riding a horse", "playing video games", "holding up an umbrella", "throwing a frisbee", "lying down", "riding a bike", "drinking", "cooking", "phoning", "chatting", "skiing", "driving", "surfing", "skateboarding", "playing baseball", "playing tennis", "using a computer", "posing", "eating" |
| Location | "amusement or theme park", "healthcare facility", "virtual or digital space", "educational institution", "industrial area", "historical landmark", "public event or gathering", "store or market", "underground or enclosed space", "transportation hub", "zoo", "water body", "office or workplace", "park or recreational area", "restaurant or dining area", "sports facility", "natural environment", "urban area or city street", "residential area" |
| Mood | "anxious", "sombre", "contemplative", "suspenseful", "serene", "nostalgic", "inspired", "whimsical", "romantic", "mysterious", "melancholic", "chaotic", "humorous", "vibrant", "peaceful", "energetic", "focused", "joyful", "relaxed", "adventurous" |
| Time of Day | "evening", "afternoon", "night", "morning", "indoor lighting", "midday" |

## B.2    DETAILS ON HARD GROUPING CRITERIA ANNOTATION

In Tab. 6, we present the additional annotated **Hard** grouping criteria ground truth alongside the **Basic** criteria for each benchmark.

While we have established more rigorous and challenging benchmarks such as COCO-4c and Food-4c, which feature up to *four* distinct grouping criteria, these annotated criteria sets do not encompass all potential grouping criteria within the image collections. This is particularly true for more complex and realistic datasets like COCO-4c, Food-4c, and Action-3c. As a result, the performance differences between different criteria proposers on these basic criteria, as shown in Fig. 4, tend to be close to each other, limiting our understanding of each proposer's ability to generate comprehensive grouping criteria.

To address this limitation, we employed human annotators to further identify and propose grouping criteria across the six benchmarks, resulting in a more extensive ground-truth set for each benchmark. This provides a better basis for evaluating the comprehensiveness of the different proposers. We refer

Table 5: **Full class names for Food-4c across the four basic criteria.**

| Criterion | Food-4c |
|---|---|
| Food Type | "apple pie", "baby back ribs", "baklava", "beef carpaccio", "beef tartare", "beet salad", "beignets", "bibimbap", "bread pudding", "breakfast burrito", "bruschetta", "caesar salad", "cannoli", "caprese salad", "carrot cake", "ceviche", "cheesecake", "cheese plate", "chicken curry", "chicken quesadilla", "chicken wings", "chocolate cake", "chocolate mousse", "churros", "clam chowder", "club sandwich", "crab cakes", "creme brulee", "croque madame", "cup cakes", "deviled eggs", "donuts", "dumplings", "edamame", "eggs benedict", "escargots", "falafel", "filet mignon", "fish and chips", "foie gras", "french fries", "french onion soup", "french toast", "fried calamari", "fried rice", "frozen yogurt", "garlic bread", "gnocchi", "greek salad", "grilled cheese sandwich", "grilled salmon", "guacamole", "gyoza", "hamburger", "hot and sour soup", "hot dog", "huevos rancheros", "hummus", "ice cream", "lasagna", "lobster bisque", "lobster roll sandwich", "macaroni and cheese", "macarons", "miso soup", "mussels", "nachos", "omelette", "onion rings", "oysters", "pad thai", "paella", "pancakes", "panna cotta", "peking duck", "pho", "pizza", "pork chop", "poutine", "prime rib", "pulled pork sandwich", "ramen", "ravioli", "red velvet cake", "risotto", "samosa", "sashimi", "scallops", "seaweed salad", "shrimp and grits", "spaghetti bolognese", "spaghetti carbonara", "spring rolls", "steak", "strawberry shortcake", "sushi", "tacos", "takoyaki", "tiramisu", "tuna tartare", "waffles" |
| Cuisine | "japanese", "indian", "american", "greek", "spanish", "mexican", "italian", "vietnamese", "canadian", "korean", "chinese", "middle eastern", "french", "thai", "general" |
| Course | "appetizer", "main course", "side dish", "dessert", "breakfast" |
| Diet | "omnivore", "vegan", "vegetarian", "gluten free" |

to this set of larger annotation criteria as the **Hard** criteria, in contrast to the **Basic** criteria, which involve per-image annotations. Note that for the **Hard** criteria, per-image label annotation is not provided due to the high cost of annotation. The procedure for obtaining the **Hard** grouping criteria is as follows:

**(1) Criteria Discovery:** We divided each dataset into batches of 100 images, displaying each batch in a $10\times10$ grid. Five human annotators were assigned to each batch and instructed to identify as many valid grouping criteria as possible. The proposed criteria from each annotator were then combined to form a comprehensive set of grouping criteria for the dataset.

**(2) Criteria Merging:** After collecting the annotated criteria from all five annotators, we aggregated the criteria and manually cleaned the set by merging semantically similar criteria (*e.g.*, Location and Place) and discarding binary grouping criteria, as the inclusion of binary criteria can result in an unmanageable number of grouping criteria for complex datasets.

By following this process, we developed a more comprehensive grouping criteria set as the **Hard** ground-truth for each benchmark, as shown in Tab. 6. This resulted in sets containing 8 criteria for Fruit-2c, 4 criteria for card, 11 criteria for Action-3c, 7 criteria for Clevr-4c, 17 criteria for COCO-4c, and 11 criteria for Food-4c. These expanded ground-truth sets enable us to more effectively evaluate the capabilities of various criteria discovery methods, providing a clearer understanding of different criteria proposers.

## C  FURTHER IMPLEMENTATION DETAILS

In this section, we provide detailed descriptions of the exact prompts used in our framework, along with additional implementation details for the proposed Criteria Proposer in App. C.1 and the Semantic Grouper in App. C.2.

Table 6: **Annotated criteria for the six benchmarks.** The basic criteria are annotated on per-image level for each benchmark, while the hard criteria (those not in the basic criteria) are further exhaustively annotated by human annotators for further evaluating the performance of the rule proposer in SMC task.

| COCO-4c | | Food-4c | | Action-3c | |
|---|---|---|---|---|---|
| **Basic criteria** | **Hard criteria** | **Basic criteria** | **Hard criteria** | **Basic criteria** | **Hard criteria** |
| Total: 4 | Total: 17 | Total: 4 | Total: 11 | Total: 3 | Total: 11 |
| Activity | Activity | Food Type | Food Type | Action | Action |
| Location | Location | Cuisine | Cuisine | Mood | Mood |
| Mood | Mood | Course | Course | Location | Location |
| Time of Day | Time of Day | Diet | Diet | | Clothing Style |
| | Interaction | | Tableware Type | | Number of People Present |
| | Number of People Present | | Presentation Style | | Age or Age Composition |
| | Group Dynamics | | Color Palette | | Race or Race Composition |
| | Clothing Style | | Setting/Theme | | Occasion or Event Type |
| | Occasion or Event Type | | Primary Taste | | Group Dynamics |
| | Photo Style | | Primary Ingredient | | Lighting Condition |
| | Type of Animal Present | | Cooking Method | | Gender or Gender Composition |
| | Weather | | | | |
| | Type of Primary Object | | | | |
| | Continent | | | | |
| | Age or Age Composition | | | | |
| | Race or Race Composition | | | | |
| | Gender or Gender Composition | | | | |

| Clevr-4c | | Card-2c | | Fruit-2c | |
|---|---|---|---|---|---|
| **Basic criteria** | **Hard criteria** | **Basic criteria** | **Hard criteria** | **Basic criteria** | **Hard criteria** |
| Total: 4 | Total: 7 | Total: 2 | Total: 4 | Total: 2 | Total: 8 |
| Color | Color | Rank | Rank | Species | Species |
| Texture | Texture | Suit | Suit | Color | Color |
| Shape | Shape | | Color | | Size |
| Count | Count | | Illustration Style | | Seasonality |
| | Spatial Positioning | | | | Primary Taste |
| | Count of Surface | | | | Texture |
| | Complexity of Geometry | | | | Ripeness |
| | | | | | Fruit Quantity and Arrangement |

Table 7: **Prompts for the MLLM in the image-based proposer for criteria proposing.**

| Prompt purpose | Prompt |
|---|---|
| System Prompt | You are a helpful AI assistant |
| Input Explanation | This image contains 64 individual images arranged in 8 columns and 8 rows. |
| Goal Explanation | I am a machine learning researcher trying to identify all the possible clustering criteria or rules that could be used to group these images so I can better understand my data. |
| Task Instruction | Your job is to carefully analyze the entire set of the 64 images, and identify five distinct clustering criteria or rules that could be used to cluster or group these images. Please consider different characteristics. |
| Output Instruction | Please write a list of the 5 identified clustering criteria or rules (separated by bullet points "*"). |
| Task Reinforcement | Again, I want to identify all the possible clustering criteria or rules that could be used to group these images. List the 5 distinct clustering criteria or rules that you identified from the 64 images. Answer with a list (separated by bullet points "*"). Your response: |

## C.1 PROMPTS AND IMPLEMENTATION DETAILS OF CRITERIA PROPOSER

**Image-based Proposer:** In Tab. 7, we present the exact prompt used in the image-based proposer for querying the MLLM LLaVA-NeXT-Interleave-7B (Li et al., 2024). Given a target image set, we first randomly shuffle the images and divide them into disjoint subsets, each containing 64 images. Each subset is then stitched into an $8 \times 8$ image grid, treated as a single image, and fed into the MLLM.

Table 8: **Prompts for the LLM used in the tag-based proposer for criteria proposing.** We embed the exact image captions by replacing the placeholders "{TAGS}" in the prompt.

| Prompt purpose | Prompt |
|---|---|
| System Prompt | You are a helpful assistant. |
| Input Explanation | The following are the tagging results of a set of images in the format of "Image ID: tag 1, tag 2, ..., tag 10". These assigned tags reflect the visible semantic content of each image: |
| Tag Embedding | Image 1: "{TAGS}"
Image 2: "{TAGS}"
...
Image N: "{TAGS}" |
| Goal Explanation | I am a machine learning researcher trying to figure out the potential clustering or grouping criteria that exist in these images. So I can better understand my data and group them into different clusters based on different criteria. |
| Task Instruction | Please analyze these images by using their assigned tags. Come up with an array of distinct clustering criteria that exist in this set of images. |
| Output Instruction | Please write a list of clustering criteria (separated by bullet points "*"). |
| Task Reinforcement | Again, I want to figure out what are the potential clustering or grouping criteria that I can use to group these images into different clusters. List an array of clustering or grouping criteria that often exist in this set of images based on the tagging results. Answer with a list (separated by bullet points "*").
Your response: |

Table 9: **Prompts for the MLLM in the caption-based proposer for generating detailed descriptions of the images.**

| Prompt purpose | Prompt |
|---|---|
| System Prompt | You are a helpful AI assistant |
| Task Instruction | Describe the following image in detail. |

For each subset, the MLLM is prompted to propose 5 distinct grouping criteria for organizing the images within that subset, using the prompt shown in Tab. 7. After iterating through all subsets, we take the union of the criteria proposed for each subset as the discovered criteria for the target image set. Finally, we deduplicate the discovered criteria and accumulate them into the criteria pool.

**Tag-based Proposer:** In Tab. 8, we present the exact prompt used in the tag-based proposer for querying the LLM Llama-3.1-8B (Meta, 2024b). For a given target image set, we first utilize an open-vocabulary tagger, CLIP ViT-L/14 (Radford et al., 2021), to assign 10 related natural language tags to each image. These tags are selected from the WordNet (Miller, 1995) vocabulary, which contains 118k English synsets, and represent the semantic content of the images. We employ the standard prompt "A photo of {concept}" provided by CLIP for image tagging. Next, we embed the assigned tags into the prompt shown in Tab. 8 to carry the semantics of the entire image set and query the LLM to propose grouping criteria. The criteria proposed by the LLM are then added to the criteria pool. Note that in this case, we embed the tags for the entire dataset into a single prompt for criteria proposal, without reaching the LLM context length limits (*e.g.*,, 128k for Llama-3.1-8B) for the datasets used in our experiments. However, for larger datasets, it may be necessary to split the dataset into subsets, prompt the LLM for each subset, and use the union of the proposed criteria as the final output.

**Caption-based Proposer:** We present the prompt used in the caption-based proposer for the MLLM LLaVA-NeXT-7B (Liu et al., 2024b) in Tab. 9, and the prompt for the LLM Llama-3.1-8B (Meta, 2024b) in Tab. 10. Specifically, we first use the MLLM with a general prompt to generate detailed descriptions for each image in the target dataset, effectively translating the visual information into natural language. The generated captions are then *randomly shuffled* and split into disjoint subsets, each containing 400 captions. Next, we embed the captions from each subset into the prompt shown

Table 10: **Prompts for the LLM used in the caption-based proposer for criteria proposing.** We embed the exact image captions by replacing the placeholders "{CAPTION}" in the prompt.

| Prompt purpose | Prompt |
|---|---|
| System Prompt | You are a helpful assistant. |
| Input Explanation | The following are the result of captioning a set of images: |
| Caption Embedding | Image 1: "{CAPTION}" 
 Image 2: "{CAPTION}" 
 ... 
 Image N: "{CAPTION}" |
| Goal Explanation | I am a machine learning researcher trying to figure out the potential clustering or grouping criteria that exist in these images. So I can better understand my data and group them into different clusters based on different criteria. |
| Task Instruction | Come up with ten distinct clustering criteria that exist in this set of images. |
| Output Instruction | Please write a list of clustering criteria (separated by bullet points "*"). |
| Task Reinforcement | Again I want to figure out what are the potential clustering/grouping criteria that I can use to group these images into different clusters. List ten clustering or grouping criteria that often exist in this set of images based on the captioning results. Answer with a list (separated by bullet points "*"). 
 Your response: |

Table 11: **Prompts for the LLM used in Proposed Criteria Refinement step** We embed the exact initially discovered criteria by replacing the placeholders "{CRITERION}" in the prompt.

| Prompt purpose | Prompt |
|---|---|
| System Prompt | You are a helpful assistant. |
| Input Explanation | I am a machine learning researcher working with a set of images. I aim to cluster this set of images based on the various clustering criteria present within them. Below is a preliminary list of clustering criteria that I've discovered to group these images: |
| Criteria Embedding: | * Criterion 1: "{CRITERION}" 
 * Criterion 2: "{CRITERION}" 
 ... 
 * Criterion L: "{CRITERION}" |
| Goal Explanation | My goal is to refine this list by merging similar criteria and rephrasing them using more precise and informative terms. This will help create a set of distinct, optimized clustering criteria. |
| Task Instruction | Your task is to first review and understand the initial list of clustering criteria provided. Then, assist me in refining this list by: 
 * Merging similar criteria. 
 * Expressing each criterion more clearly and informatively. |
| Output Instruction | Please respond with the cleaned and optimized list of clustering criteria, formatted as bullet points (using "*"). 
 Your response: |

in Tab. 10 and use it to query the LLM to propose grouping criteria for the images represented by the captions. After iterating through all subsets, we take the union of the proposed criteria across subsets as the discovered criteria for the target image set. Finally, we deduplicate these criteria and add them to the criteria pool. Due to the context window limitations of LLMs, embedding all captions into a single prompt is infeasible. To address this, we limit each subset to 400 captions, which results in approximately 115k tokens per subset. This strategy allows us to remain within the context length limits of modern LLMs (*e.g.*,, 128k tokens for both Llama-3.1 and GPT-4o) while maximizing the number of samples per query to effectively propose clustering criteria.

**Criteria Pool Refinement:** In Tab. 11, we present the exact prompt used for criteria pool refinement when querying the LLM Llama-3.1-8B (Meta, 2024b). Since the accumulated criteria pool $\widetilde{\mathcal{R}}$ may contain highly similar or noisy clustering criteria, we embed the criteria from the pool into the prompt

Table 12: **Prompts for the LLM used in the image-based grouper for automatic criterion-specific VQA question generation.** We embed the exact discovered criterion by replacing the placeholder "{CRITERION}" in the prompt.

| Prompt purpose | Prompt |
|---|---|
| System Prompt | You are a helpful assistant. |
| Goal Explanation | Hello! I am a machine learning researcher focusing on image categorization based on the aspect of "{CRITERION}" depicted in images. |
| Task Instruction | Therefore, I need your assistance in designing a prompt for the Visual Question Answering (VQA) model to help it identify the "{CRITERION}" category in a given image at three different granularity. Please help me design and generate this prompt using the following template: "Question: [Generated VQA Prompt Question] Answer (reply with an abstract, a common, and a specific category name, respectively):". The generated prompt should be simple and straightforward. |
| Output Instruction | Please respond with only the generated prompt using the following format "* Answer *". 
 Your response: |

Table 13: **Prompts for the LLM used in the tag-based grouper for generating middle-grained criterion-specific tags.** We embed the exact discovered criterion by replacing the placeholder "{CRITERION}" in the prompt.

| Prompt purpose | Prompt |
|---|---|
| System Prompt | You are a helpful assistant. |
| Goal Explanation | Hello! I am a machine learning researcher focusing on image categorization of a certain aspect. I'm interested in generating a list of tags specifically for categorizing the types of "{CRITERION}" depicted in images. |
| Task Instruction | Please provide a list of potential "{CRITERION}" category names. Please generate diverse category names. Do not include too general or specific category names such as "Sports". |
| Output Instruction | Please respond with the list of category names. Each category should be formatted as follows: "* Category Name". 
 Your response: |

shown in Tab. 11 and ask the LLM to merge similar criteria and rephrase their names to enhance clarity. This process yields a refined set of grouping criteria, which is then passed to the next stage for image grouping.

## C.2 PROMPTS AND IMPLEMENTATION DETAILS OF SEMANTIC GROUPER

**Image-based Grouper:** In Tab. 12, we present the prompt used to query the LLM Llama-3.1-8B (Meta, 2024b) for automatically generating criterion-specific VQA questions for the image-based grouper. The objective at this stage is to condition the VQA model BLIP-2 Flan-T5$_{XXL}$ (Li et al., 2023a) to label each image across three different semantic granularity levels based on a specific criterion. To guide the VQA model effectively, a criterion-specific question is required.

Rather than manually creating these questions, we embed the target criterion into the prompt shown in Tab. 12 and query the LLM to automatically generate high-quality, criterion-specific questions. These questions are then used to direct the VQA model, enabling it to accurately label each image according to the visual content relevant to the target criterion.

**Tag-based Grouper:** We present the prompts used in the tag-based grouper for querying the LLM Llama-3.1-8B. The prompt for generating criterion-specific tags is shown in Tab. 13, while the prompts for generating coarse-grained and fine-grained tags are shown in Tab. 14 and Tab. 15, respectively.

In the tag-based grouper, we begin by embedding the target criterion into the prompt from Tab. 13 to generate criterion-specific tags at a middle granularity. To enhance the diversity and coverage of

Table 14: **Prompts for the LLM used in the tag-based grouper for generating coarse-grained criterion-specific tags.** We embed the exact discovered criterion and middle-grained category by replacing the placeholder "{CRITERION}" and "{MIDDLE-GRAINED CATEGORY NAME}" in the prompt, respectively.

| Prompt purpose | Prompt |
|---|---|
| System Prompt | You are a helpful assistant. |
| Task Instruction | Generate a list of three more abstract or general "{CRITERION}" super-categories that the following "{CRITERION}" category belongs to and output the list separated by "&" (without numbers): "{MIDDLE-GRAINED CATEGORY NAME}" |
| Output Instruction | Your response: |

Table 15: **Prompts for the LLM used in the tag-based grouper for generating fine-grained criterion-specific tags.** We embed the exact discovered criterion and middle-grained category by replacing the placeholder "{CRITERION}" and "{MIDDLE-GRAINED CATEGORY NAME}" in the prompt, respectively.

| Prompt purpose | Prompt |
|---|---|
| System Prompt | You are a helpful assistant. |
| Task Instruction | Generate a list of ten more detailed or specific "{CRITERION}" sub-categories of the following "{CRITERION}" category and output the list separated by "&" (without numbers): "{MIDDLE-GRAINED CATEGORY NAME}" |
| Output Instruction | Your response: |

Table 16: **Prompts for the MLLM used in the caption-based grouper for generating criterion-specific captions.** We embed the exact discovered criterion by replacing the placeholder "{CRITERION}" in the prompt.

| Prompt purpose | Prompt |
|---|---|
| System Prompt | You are a helpful AI assistant. |
| Task Instruction | Analyze the image focusing specifically on the "{CRITERION}". Provide a detailed description of the "{CRITERION}" depicted in the image. Highlight key elements and interactions relevant to the "{CRITERION}" that enhance the understanding of the scene. |
| Output Instruction | Your response: |

the tags, we query the LLM 10 times and take the union of the generated tags after deduplication as candidates. Following the SHiNe framework (Liu et al., 2024d), for each middle-grained tag, we further embed it into the prompts from Tab. 14 and Tab. 15 to generate 3 super-categories (coarse-grained) and 10 sub-categories (fine-grained) for each tag.

After generating coarse and fine-grained categories for all middle-grained tags, we take the union of the super-categories as the coarse-grained tag candidates and the union of the sub-categories as the fine-grained tag candidates. Lastly, we use the open-vocabulary tagger CLIP ViT-L/14 to assign the most relevant tags to each image based on cosine similarity, using candidates from each granularity level. After tagging all the images, we group those sharing the same tag into clusters, yielding the clustering result. Note that we do not utilize lexical databases such as WordNet (Miller, 1995) or ConceptNet (Speer et al., 2017) for tag generation, as they do not support free-form input and may not capture certain discovered criteria.

**Caption-based Grouper:** We first present the MLLM prompt used for LLaVA-NeXT-7B (Liu et al., 2024b) to generate criterion-specific captions in Tab. 16. Following this, we present the LLM Llama-3.1-8B prompts used in the caption-based grouper for the *Initial Naming* step in Tab. 17, the *Multi-granularity Cluster Refinement* step in Tab. 18, and the *Final Assignment* step in Tab. 19.

Specifically, we begin by generating criterion-specific captions for each image using LLaVA-NeXT-7B with the prompt shown in Tab. 16. For each image, we then embed its criterion-specific caption and the relevant criterion into the LLM prompt shown in Tab. 17, querying the LLM to assign an initial name based on the target criterion. Once the initial names for all images in the dataset are obtained, we embed these names along with the target criterion into the prompt in Tab. 18 to query

Table 17: **Prompts for the LLM used in the caption-based grouper at the *Initial Naming* step for initially assigning a criterion-based category name to the image based on its criterion-specific caption.** We embed the exact discovered criterion and the corresponding criterion-specific caption by replacing the placeholder "{CRITERION}" and "{CRITERION-SPECIFIC CAPTION}" in the prompt, respectively.

| Prompt purpose | Prompt |
|---|---|
| System Prompt | You are a helpful assistant. |
| Input Explanation | The following is the description about the "{CRITERION}" of an image: |
| Caption Embedding | "{CRITERION-SPECIFIC CAPTION}" |
| Goal Explanation | I am a machine learning researcher trying to assign a label to this image based on what is the "{CRITERION}" depicted in this image. |
| Task Instruction | Understand the provided description carefully and assign a label to this image based on what is the "{CRITERION}" depicted in this image. |
| Output Instruction | Please respond in the following format within five words: "*Answer*". Do not talk about the description and do not respond long sentences. The answer should be within five words. |
| Task Reinforcement | Again, your job is to understand the description and assign a label to this image based on what is the "{CRITERION}" shown in this image. Your response: |

the LLM for cluster name refinement across three semantic granularity levels: coarse, middle, and fine.

Finally, for each image, we embed the target criterion, its criterion-specific caption, and cluster candidates from each granularity level into the prompt shown in Tab. 19, and use this to query the LLM for final cluster assignment at each granularity level.

# D   FURTHER IMPLEMENTATION DETAILS OF THE COMPARED METHODS

In this section, we provide the implementation details of the compared methods, IC|TC (Kwon et al., 2024) and MMaP (Yao et al., 2024).

**Implementation details of IC|TC** (Kwon et al., 2024): In the original implementation of IC|TC, LLaVA-1.5 (Liu et al., 2024c) was used as the MLLM, and GPT-4-2023-03-15-preview (OpenAI, 2023) as the LLM. However, since the GPT-4-2023-03-15-preview API has been deprecated, we re-implemented IC|TC using the state-of-the-art MLLM LLaVA-NeXT-7B (Liu et al., 2024b) and the latest version of GPT-turbo-2024-04-09 as the LLM, while strictly adhering to the original IC|TC prompt design in our experiments to ensure a fair comparison.

**Implementation details of MMaP** (Yao et al., 2024): We closely followed the training configuration outlined in the original MMaP paper. Specifically, GPT-turbo-2024-04-09 was used as the LLM to generate reference words for each dataset. We then prompt-tuned CLIP-ViT/B32 using Adam with a momentum of 0.9, training the model for 1,000 epochs for each criterion across all datasets. Hyperparameters were optimized according to the loss score of MMaP, with the learning rate searched in {0.1, 0.05, 0.01, 0.005, 0.001, 0.0005}, weight decay in {0.0005, 0.0001, 0.00005, 0.00001, 0}, $\alpha$ and $\beta$ in {0.0, 0.1, 0.2, ..., 1.0}, and $\lambda$ fixed at 1 for all experiments. After training, KMeans, with the ground-truth number of clusters, was applied for each criterion and dataset to perform clustering.

# E   SUPPLEMENTARY RESULTS OF THE QUANTITATIVE EXPERIMENTS

In this section, we present additional numerical experiment results to supplement the figures in the main paper. In Sec. E.1, we provide supplementary results for the evaluation of the Criteria Proposer in our framework. In Sec. E.2, we present additional results for the evaluation of the Semantic Grouper across various criteria on the six tested benchmarks. Furthermore, we include expanded results comparing our framework to prior criteria-conditioned clustering methods. Lastly, we present detailed results from the ablation study of the multi-granularity refinement component in Sec. E.4.

Table 18: **Prompts for the LLM used in the caption-based grouper at the *Multi-granularity Cluster Generation* step for refining the initially assigned names to a structured three granularity levels.** We embed the exact discovered criterion and the initially assigned name categories by replacing the placeholder "{CRITERION}" and "{MIDDLE-GRAINED CATEGORY NAME}" in the prompt, respectively.

| Prompt purpose | Prompt |
|---|---|
| System Prompt | You are a helpful assistant. |
| Input Explanation | The following is an initial list of "{CRITERION}" categories. These categories might not be at the same semantic granularity level. For example, category 1 could be "cutting vegetables", while category 2 is simply "cutting". In this case, category 1 is more specific than category 2. |
| Category Embedding | * "{MIDDLE-GRAINED CATEGORY NAME}" 
 * "{MIDDLE-GRAINED CATEGORY NAME}" 
 ... 
 * "{MIDDLE-GRAINED CATEGORY NAME}" |
| Task Instruction | These categories might not be at the same semantic granularity level. For example, category 1 could be "cutting vegetables", while category 2 is simply "cutting". In this case, category 1 is more specific than category 2. Your job is to generate a three-level class hierarchy (class taxonomy, where the first level contains more abstract or general coarse-grained classes, the third level contains more specific fine-grained classes, and the second level contains intermediate mid-grained classes) of "{CRITERION}" based on the provided list of "{CRITERION}" categories. Follow these steps to generate the hierarchy. |
| Sub-task Instruction | Follow these steps to generate the hierarchy: 
 Step 1 - Understand the provided initial list of "{CRITERION}" categories. The following three-level class hierarchy generation steps are all based on the provided initial list. 
 Step 2 - Generate a list of abstract or general "{CRITERION}" categories as the first level of the class hierarchy, covering all the concepts present in the initial list. 
 Step 3 - Generate a list of middle-grained "{CRITERION}" categories as the second level of the class hierarchy, in which the middle-grained categories are the subcategories of the categories in the first level. The categories in the second-level are more specific than the first level but should still cover and reflect all the concepts present in the initial list. 
 Step 4 - Generate a list of more specific fine-grained "{CRITERION}" categories as the third level of the class hierarchy, in which the categories should reflect more specific "{CRITERION}" concepts that you can infer from the initial list. The categories in the third-level are subcategories of the second-level. 
 Step 5 - Output the generated three-level class hierarchy as a JSON object where the keys are the level numbers and the values are a flat list of generated categories at each level, structured like: 
 { 
 "level 1": ["categories"], 
 "level 2": ["categories"], 
 "level 3": ["categories"] 
 } |
| Output Instruction | Please only output the JSON object in your response and simply use a flat list to store the generated categories at each level. 
 Your response: |

## E.1 SUPPLEMENTARY RESULTS FOR CRITERIA PROPOSER EVALUATION

We provide detailed numerical results corresponding to Fig. 4(a) in Tab. 20 and Fig. 4(c) in Tab. 21 for the six tested benchmarks.

Although captions generated by the MLLM may exhibit some information loss (*e.g.*,, ignoring small objects or attributes) (He et al., 2024) and hallucinations (*e.g.*,, introducing objects not present in the images) Liu et al. (2024a), these issues generally occur at the object or fine-grained attribute level. However, when reasoning about grouping criteria for SMC task, the focus is on identifying general

Table 19: **Prompts for the LLM used in the caption-based grouper at the *Final Assignment* step.** We embed the exact discovered criterion and the refined category names from each granularity level, by replacing the placeholder "{CRITERION}" and "{CANDIDATE CATEGORY NAME}" in the prompt, respectively.

| Prompt purpose | Prompt |
|---|---|
| System Prompt | You are a helpful assistant. |
| Input Explanation | The following is a detailed description about the "{CRITERION}" of an image. |
| Caption Embedding | "{CRITERION-SPECIFIC CAPTION}" |
| Task Instruction | Based on the content and details provided in the description, classify the image into one of the specified "{CRITERION}" categories listed below: |
| Candidate Category Embedding | "{CRITERION}" categories:
* "{CANDIDATE CATEGORY NAME}"
* "{CANDIDATE CATEGORY NAME}"
...
* "{CANDIDATE CATEGORY NAME}" |
| Output Instruction | Ensure that your classification adheres to the details mentioned in the image description. Respond with the classification result in the following format: "*category name*".
Your response: |

thematic elements shared across the image set. As a result, these minor inconsistencies in the captions do not hinder the LLM in our framework from effectively reasoning about grouping criteria, helping the Caption-based Proposer to achieve the best performance among all the studied design choices.

Table 20: **Comparison of True Positive Rate (TPR) (%) for criteria proposers across the six SMC benchmarks**. TPR performance is reported for both Basic and Hard ground-truth criteria. The best performance is highlighted in bold.

| | COCO-4c | | Food-4c | | Action-3c | | Clevr-4c | | Card-2c | | Fruit-2c | | Average | |
|---|---|---|---|---|---|---|---|---|---|---|---|---|---|---|
| | Basic | Hard | Basic | Hard | Basic | Hard | Basic | Hard | Basic | Hard | Basic | Hard | Basic | Hard |
| Image-based | **100.0** | 52.9 | 25.0 | 36.4 | 66.7 | 54.6 | 50.0 | 28.6 | 50.0 | 25.0 | 50.0 | 20.0 | 56.9 | 36.2 |
| Tag-based | 50.0 | 35.3 | **100.0** | 72.7 | 66.7 | 36.4 | 75.0 | 42.9 | 50.0 | 50.0 | 50.0 | 20.0 | 65.3 | 42.9 |
| Caption-based | **100.0** | **64.7** | **100.0** | **81.8** | **100.0** | **72.7** | **100.0** | **71.4** | **100.0** | **100.0** | **100.0** | **60.0** | **100.0** | **75.1** |

Table 21: **Study of the impact of data scale on criteria discovery.** The Caption-based Proposer is used for criteria discovery, and TPR performance (%) is reported on the *Hard* ground-truth criteria sets across the six SMC benchmarks for different data scales. The best performance is highlighted in bold.

| Data scales | COCO-4c | Food-4c | Action-3c | Clevr-4c | Card-2c | Fruit-2c | Average |
|---|---|---|---|---|---|---|---|
| 100% | **64.7** | **81.8** | **72.7** | 71.4 | **100.0** | **60.0** | **75.1** |
| 80% | 47.1 | 72.7 | 54.6 | 71.4 | 75.0 | 30.0 | 58.5 |
| 60% | 52.9 | 63.6 | 54.6 | 71.4 | **100.0** | 50.0 | 65.4 |
| 40% | 41.2 | 45.5 | 45.5 | **85.7** | **100.0** | 40.0 | 59.6 |
| 20% | 35.3 | 45.5 | 36.4 | 42.9 | **100.0** | 40.0 | 50.0 |
| 1 img | 23.5 | 36.4 | 27.3 | 57.1 | 75.0 | 50.0 | 44.9 |

## E.2 SUPPLEMENTARY RESULTS FOR SEMANTIC GROUPER EVALUATION

In this section, we present the expanded numerical results comparing different semantic groupers to supplement the summary in Fig. 6. Specifically, we provide detailed results for the evaluation of the six tested datasets as follows:

- COCO-4c (Fig. 6(a)) in Tab. 22
- Card-2c (Fig. 6(b)) in Tab. 23
- Action-3c (Fig. 6(c)) in Tab. 24
- Food-4c (Fig. 6(d)) in Tab. 25

Table 22: **Comparison of Semantic Groupers on COCO-4c.** We report Clustering Accuracy (CAcc), Semantic Accuracy (SAcc), and their Harmonic Mean (HM) in percentages (%). These results are plotted in Fig. 6(a).

| Methods | Activity | | | Location | | | Mood | | | Time of Day | | |
|---|---|---|---|---|---|---|---|---|---|---|---|---|
| | CAcc | SAcc | HM | CAcc | SAcc | HM | CAcc | SAcc | HM | CAcc | SAcc | HM |
| CLIP Zero-shot | 62.6 | 73.5 | 67.6 | 34.3 | 51.5 | 41.1 | 22.4 | 43.3 | 29.5 | 40.6 | 74.1 | 52.4 |
| KMeans CLIP | 34.4 | - | - | 32.7 | - | - | 18.9 | - | - | 38.6 | - | - |
| KMeans DINOv1 | 34.8 | - | - | 37.5 | - | - | 17.9 | - | - | 36.5 | - | - |
| KMeans DINOv2 | 38.2 | - | - | 37.9 | - | - | 22.5 | - | - | 43.8 | - | - |
| Img-based BLIP-2 | 48.7 | 64.1 | **55.3** | 39.6 | 48.0 | 43.4 | 30.2 | 37.5 | 33.4 | 40.7 | 60.3 | 48.6 |
| Img-based LLaVA | 46.5 | 61.8 | 53.1 | 34.0 | 46.3 | 39.2 | 28.0 | 24.7 | 26.3 | 39.4 | 51.7 | 44.7 |
| Tag-based | 43.2 | 51.5 | 47.0 | 28.6 | 46.6 | 35.5 | 13.0 | 25.6 | 17.2 | 19.3 | 48.8 | 27.7 |
| Caption-based | 44.1 | 48.9 | 46.4 | 55.2 | 55.6 | **55.4** | 38.1 | 32.6 | **35.2** | 67.6 | 56.7 | **61.7** |

Table 23: **Comparison of Semantic Groupers on Card-2c.** We report Clustering Accuracy (CAcc), Semantic Accuracy (SAcc), and their Harmonic Mean (HM) in percentages (%). These results are plotted in Fig. 6(b).

| Methods | Suit | | | Rank | | |
|---|---|---|---|---|---|---|
| | CAcc | SAcc | HM | CAcc | SAcc | HM |
| CLIP Zero-shot | 47.9 | 69.5 | 56.7 | 35.0 | 64.2 | 45.3 |
| KMeans CLIP | 45.0 | - | - | 28.6 | - | - |
| KMeans DINOv1 | 38.5 | - | - | 20.7 | - | - |
| KMeans DINOv2 | 36.7 | - | - | 22.3 | - | - |
| Img-based BLIP-2 | 66.7 | 77.7 | **71.8** | 47.5 | 54.4 | 50.7 |
| Img-based LLaVA | 36.8 | 65.8 | 47.2 | 24.6 | 49.8 | 32.9 |
| Tag-based | 39.2 | 32.9 | 35.8 | 22.3 | 39.1 | 28.4 |
| Caption-based | 54.5 | 73.6 | 62.6 | 92.1 | 95.1 | **93.6** |

Table 24: **Comparison of Semantic Groupers on Action-3c.** We report Clustering Accuracy (CAcc), Semantic Accuracy (SAcc), and their Harmonic Mean (HM) in percentages (%). These results are plotted in Fig. 6(c).

| Methods | Action | | | Location | | | Mood | | |
|---|---|---|---|---|---|---|---|---|---|
| | CAcc | SAcc | HM | CAcc | SAcc | HM | CAcc | SAcc | HM |
| CLIP Zero-shot | 97.1 | 99.2 | 98.1 | 66.7 | 67.1 | 66.9 | 75.5 | 80.7 | 78.0 |
| KMeans CLIP | 62.3 | - | - | 58.3 | - | - | - | - | - |
| KMeans DINOv1 | 49.3 | - | - | 61.4 | - | - | - | - | - |
| KMeans DINOv2 | 75.7 | - | - | 67.6 | - | - | - | - | - |
| Img-based BLIP-2 | 79.7 | 80.9 | 80.3 | 43.3 | 42.4 | 42.8 | 43.1 | 43.8 | 43.4 |
| Img-based LLaVA | 70.1 | 60.5 | 65.0 | 45.8 | 42.8 | 44.2 | 32.0 | 38.0 | 34.7 |
| Tag-based | 70.2 | 55.0 | 61.6 | 36.8 | 48.1 | 41.7 | 50.7 | 47.6 | 49.1 |
| Caption-based | 82.8 | 82.8 | **82.8** | 69.8 | 55.2 | **61.6** | 52.3 | 50.2 | **51.2** |

- Fruit-2c (Fig. 6(e)) in Tab. 26

- Clevr-4c (Fig. 6(f)) in Tab. 27

In addition, we present the statistics of the predicted clusters at each granularity level in Tab. 28.

## E.3 SUPPLEMENTARY RESULTS FOR COMPARISON CRITERIA-CONDITIONED CLUSTERING METHODS

We provide expanded results in Tab. 29 for each criterion and benchmark, detailing the comparison of criteria-conditioned clustering methods presented in Tab. 2 in the main paper.

Table 25: **Comparison of Semantic Groupers on Food-4c.** We report Clustering Accuracy (CAcc), Semantic Accuracy (SAcc), and their Harmonic Mean (HM) in percentages (%). These results are plotted in Fig. 6(d).

| Methods | Food Type | | | Cuisine | | | Course | | | Diet | | |
|---------|------|------|------|------|------|------|------|------|------|------|------|------|
| | CAcc | SAcc | HM | CAcc | SAcc | HM | CAcc | SAcc | HM | CAcc | SAcc | HM |
| CLIP Zero-shot | 90.6 | 94.6 | 92.6 | 54.9 | 81.4 | 65.6 | 63.5 | 84.7 | 72.6 | 47.6 | 59.9 | 53.0 |
| KMeans CLIP | 66.1 | - | - | 29.8 | - | - | 49.5 | - | - | 36.9 | - | - |
| KMeans DINOv1 | 33.6 | - | - | 15.3 | - | - | 38.1 | - | - | 41.4 | - | - |
| KMeans DINOv2 | 72.7 | - | - | 22.5 | - | - | 47.6 | - | - | 43.4 | - | - |
| Img-based BLIP-2 | 54.2 | 71.4 | **61.6** | 54.8 | 73.3 | **62.7** | 42.3 | 71.0 | 53.0 | 34.2 | 53.8 | 41.9 |
| Img-based LLaVA | 42.2 | 64.0 | 50.9 | 33.7 | 57.6 | 42.6 | 46.9 | 73.1 | 57.1 | 27.0 | 40.5 | 32.4 |
| Tag-based | 45.0 | 63.3 | 52.6 | 48.8 | 42.1 | 45.2 | 42.7 | 70.1 | 53.1 | 25.2 | 34.1 | 29.0 |
| Caption-based | 34.6 | 54.2 | 42.2 | 47.0 | 65.9 | 54.9 | 69.1 | 85.7 | **76.5** | 41.5 | 54.0 | **46.9** |

Table 26: **Comparison of Semantic Groupers on Fruit-2c.** We report Clustering Accuracy (CAcc), Semantic Accuracy (SAcc), and their Harmonic Mean (HM) in percentages (%). These results are plotted in Fig. 6(e).

| Methods | Species | | | Color | | |
|---------|------|------|------|------|------|------|
| | CAcc | SAcc | HM | CAcc | SAcc | HM |
| CLIP Zero-shot | 84.0 | 93.1 | 88.3 | 54.8 | 83.5 | 66.1 |
| KMeans CLIP | 67.1 | - | - | 39.6 | - | - |
| KMeans DINOv1 | 53.8 | - | - | 36.0 | - | - |
| KMeans DINOv2 | 71.2 | - | - | 36.7 | - | - |
| Img-based BLIP-2 | 70.7 | 68.3 | 69.5 | 40.9 | 70.6 | 51.8 |
| Img-based LLaVA | 63.9 | 67.8 | 65.8 | 51.0 | 83.2 | **63.2** |
| Tag-based | 64.0 | 67.1 | 65.5 | 54.1 | 44.1 | 48.6 |
| Caption-based | 76.9 | 70.7 | **73.7** | 53.3 | 51.5 | 52.4 |

Table 27: **Comparison of Semantic Groupers on Clevr-4c.** We report Clustering Accuracy (CAcc), Semantic Accuracy (SAcc), and their Harmonic Mean (HM) in percentages (%). These results are plotted in Fig. 6(f).

| Methods | Color | | | Texture | | | Count | | | Shape | | |
|---------|------|------|------|------|------|------|------|------|------|------|------|------|
| | CAcc | SAcc | HM | CAcc | SAcc | HM | CAcc | SAcc | HM | CAcc | SAcc | HM |
| CLIP Zero-shot | 77.7 | 94.0 | 85.1 | 34.1 | 41.9 | 37.6 | 43.7 | 81.5 | 56.9 | 71.1 | 72.7 | 71.9 |
| KMeans CLIP | 48.8 | - | - | 61.4 | - | - | 44.2 | - | - | 56.1 | - | - |
| KMeans DINOv1 | 53.0 | - | - | 58.4 | - | - | 47.5 | - | - | 67.0 | - | - |
| KMeans DINOv2 | 44.1 | - | - | 46.9 | - | - | 52.5 | - | - | 87.0 | - | - |
| Img-based BLIP-2 | 69.3 | 76.5 | **72.7** | 57.8 | 34.4 | 43.1 | 25.7 | 55.9 | 35.2 | 69.1 | 62.6 | 65.7 |
| Img-based LLaVA | 56.5 | 63.5 | 59.8 | 51.9 | 26.9 | 35.4 | 53.7 | 39.4 | 45.4 | 64.3 | 71.3 | **67.6** |
| Tag-based | 66.6 | 55.3 | 60.4 | 57.2 | 40.2 | 47.3 | 47.4 | 8.3 | 14.1 | 62.7 | 36.5 | 46.2 |
| Caption-based | 70.3 | 63.4 | 66.7 | 65.3 | 42.1 | **51.2** | 65.7 | 73.3 | **69.3** | 58.4 | 38.5 | 46.4 |

### E.4 SUPPLEMENTARY RESULTS FOR STUDYING THE NECESSITY OF MULTI-GRANULARITY CLUSTER GENERATION.

We present expanded results in Tab. 30 for the ablation study on multi-granularity refinement, providing a detailed breakdown of the summary shown in Fig. 7 in the main paper.

## F FURTHER STUDIES OF THE PROPOSED FRAMEWORK

In this section, we provide additional studies on our proposed framework, using the main configuration (Caption-based Proposer and Caption-based Grouper) for the analysis. In Sec. F.1, we conduct control experiments to examine the system sensitivity of our framework to different multimodal large language models (MLLMs) and large language models (LLMs). In Sec. F.2, we demonstrate how

Table 28: **Summary of cluster counts across six benchmarks for the comparison of semantic groupers.** The results yield by the main Caption-based Grouper is reported. Specifically, we report: *i)* GT: the number of ground-truth clusters; *ii)* Pred-Init: predicted clusters from initial names; *iii)* Pred-Coarse: predicted coarse-grained clusters after multi-granularity refinement; *iv)* Pred-Middle: predicted middle-grained clusters after multi-granularity refinement; and *v)* Pred-Fine: predicted fine-grained clusters after multi-granularity refinement.

| Dataset | Criteria | GT | Pred-Init | Pred-Corase | Pred-Middle | Pred-Fine |
|---------|----------|----|-----------|-------------|-------------|-----------|
| COCO-4c | Activity | 64 | 203 | 12 | 23 | 52 |
|         | Location | 19 | 145 | 7 | 14 | 28 |
|         | Mood | 20 | 122 | 15 | 25 | 30 |
|         | Time of Day | 6 | 96 | 2 | 8 | 31 |
| Food-4c | Food Type | 101 | 301 | 7 | 37 | 127 |
|         | Cuisine | 15 | 141 | 9 | 18 | 53 |
|         | Course | 5 | 97 | 4 | 12 | 78 |
|         | Diet | 4 | 139 | 5 | 8 | 64 |
| Action-3c | Action | 40 | 71 | 8 | 15 | 51 |
|         | Location | 10 | 82 | 5 | 10 | 67 |
|         | Mood | 4 | 95 | 6 | 18 | 55 |
| Clevr-4c | Color | 10 | 25 | 6 | 12 | 17 |
|         | Texture | 10 | 23 | 2 | 5 | 12 |
|         | Shape | 10 | 22 | 5 | 11 | 14 |
|         | Count | 10 | 11 | 2 | 4 | 11 |
| Card-2c | Rank | 14 | 147 | 4 | 7 | 16 |
|         | Suit | 5 | 56 | 4 | 7 | 30 |
| Fruit-2c | Species | 34 | 54 | 8 | 25 | 38 |
|         | Color | 15 | 66 | 5 | 15 | 39 |

incorporating advanced prompting strategies can further enhance the framework's performance on fine-grained criteria.

## F.1 FURTHER SYSTEM SENSITIVITY ANALYSIS OF VARIOUS MLLMS AND LLMS

In Fig. 11, we perform a system-level sensitivity analysis using our default system configuration (caption-based proposer and caption-based grouper) to examine the impact of different MLLMs and LLMs on the system performance. Since all variants successfully propose the basic criteria in each benchmark, we report the average clustering accuracy (CAcc) and semantic accuracy (SAcc) across various criteria for comparative analysis.

Specifically, in Fig. 11(a), we first fix the LLM in our system to Llama-3.1-8B (Meta, 2024b) and assess the influence of various MLLMs: GPT-4V (OpenAI, 2023), BLIP-3-4B (Xue et al., 2024), and LLaVA-NeXT-7B (Liu et al., 2024b). Next, in Fig. 11(b), we set the MLLM to LLaVA-NeXT-7B and evaluate different LLMs: GPT-4-turbo (OpenAI, 2023), GPT-4o (OpenAI, 2024), Llama-3-8B (Meta, 2024a), and Llama-3.1-8B.

Findings in Fig. 11(a) indicate a direct correlation between the size of the MLLM and the ability of our system to uncover substructures, highlighting the significant role of MLLMs in translating visual information into natural language. On the other hand, this scalability demonstrates that our system can enhance performance with more robust MLLMs, thanks to its training-free design, which ensures compatibility with any MLLM. Despite this, we use LLaVA-NeXT-7B as our default MLLM due to its *reproducibility*, being open-source and unaffected by API changes, and its capacity for local deployment, which *upholds privacy* by not exposing sensitive image data to external entities.

As for the LLMs, as depicted in Fig. 11(b), despite GPT-4-turbo showing marginally superior performance, the open-source Llama-3.1-8B achieves similar results across benchmarks, making it our default LLM. Notably, except for the Card-2c dataset, system performance remains largely consistent regardless of the power of the LLM. This consistency suggests that the reasoning task for SMC, given the capabilities of modern LLMs to tackle complex problems (Street et al., 2024), is relatively straightforward.

Table 29: **Comparison with criteria-conditioned clustering methods on the six SMC benchmarks.** We report Clustering Accuracy (CAcc) and Semantic Accuracy (SAcc)as percentages (%). Average (*Avg.*) CAcc and SAcc across different criteria on each dataset is also provided. For reference, we include the pseudo upper-bound (UB) performance of CLIP ViT-L/14 in zero-shot transfer, using ground-truth criteria and class names. Note that both IC|TC and MMaP utilize ground-truth criteria and the number of clusters for clustering. These expanded results correspond to Tab. 2.

| Benchmark | Criterion | UB | | IC\|TC | | MMaP | | Ours | |
|---|---|---|---|---|---|---|---|---|---|
| | | CAcc | SAcc | CAcc | SAcc | CAcc | SAcc | CAcc | SAcc |
| COCO-4c | Activity | 62.6 | 73.5 | 51.3 | 53.2 | 33.8 | - | 44.1 | 48.9 |
| | Location | 34.3 | 51.5 | 58.5 | 54.0 | 35.3 | - | 55.2 | 55.6 |
| | Mood | 22.4 | 43.3 | 23.2 | 40.4 | 20.9 | - | 38.1 | 32.6 |
| | Time of Day | 40.6 | 74.1 | 62.8 | 65.2 | 45.7 | - | 67.6 | 56.7 |
| | *Avg.* | 40.1 | 60.6 | 48.9 | **53.2** | 33.9 | - | **51.2** | 48.4 |
| Food-4c | Food Type | 90.6 | 94.6 | 36.0 | 52.6 | 48.9 | - | 34.6 | 54.2 |
| | Cuisine | 54.9 | 81.4 | 46.8 | 42.4 | 31.7 | - | 47.0 | 65.9 |
| | Course | 63.5 | 84.7 | 70.5 | 89.5 | 48.6 | - | 69.1 | 85.7 |
| | Diet | 47.6 | 59.9 | 48.5 | 62.1 | 45.9 | - | 41.5 | 54.0 |
| | *Avg.* | 64.1 | 80.2 | **50.5** | 61.7 | 43.8 | - | 48.1 | **64.9** |
| Clevr-4c | Color | 77.7 | 94.0 | 51.2 | 43.2 | 75.3 | - | 70.3 | 63.4 |
| | Texture | 34.1 | 41.9 | 64.9 | 26.4 | 56.5 | - | 65.3 | 42.1 |
| | Count | 43.7 | 81.5 | 46.9 | 39.0 | 53.9 | - | 65.7 | 73.3 |
| | Shape | 71.1 | 72.7 | 70.0 | 38.7 | 65.5 | - | 58.4 | 38.5 |
| | *Avg.* | 56.7 | 72.5 | 58.3 | 36.8 | 62.8 | - | **64.9** | **54.3** |
| Action-3c | Action | 97.1 | 99.2 | 86.4 | 58.7 | 51.3 | - | 82.8 | 76.3 |
| | Location | 66.7 | 67.1 | 82.0 | 52.9 | 59.4 | - | 69.8 | 55.2 |
| | Mood | 75.5 | 80.7 | 60.8 | 57.4 | 71.0 | - | 52.3 | 50.2 |
| | *Avg.* | 79.8 | 82.3 | **76.4** | 56.3 | 60.6 | - | 68.3 | **60.6** |
| Card-2c | Suit | 47.9 | 69.5 | 54.9 | 65.6 | 41.3 | - | 54.5 | 73.6 |
| | Rank | 35.0 | 64.2 | 94.6 | 96.8 | 32.6 | - | 92.1 | 95.1 |
| | *Avg.* | 41.4 | 66.9 | **74.8** | 81.2 | 36.9 | - | 73.3 | **84.3** |
| Fruit-2c | Species | 84.0 | 93.1 | 69.3 | 66.9 | 58.8 | - | 76.9 | 70.7 |
| | Color | 54.8 | 83.5 | 57.2 | 43.3 | 43.3 | - | 53.3 | 51.5 |
| | *Avg.* | 69.4 | 88.3 | 63.3 | 55.1 | 51.0 | - | **65.1** | **61.1** |

## F.2 FURTHER STUDY ON FINE-GRAINED IMAGE COLLECTIONS

Image collections may include fine-grained grouping criteria, such as `Bird species` in bird photography. Fine-grained criteria pose unique challenges for substructure discovery due to small inter-class differences and large intra-class variations (Zhang et al., 2014; Vedaldi et al., 2014; He & Peng, 2017). This requires the model to detect subtle visual distinctions to accurately infer cluster names and guide the grouping process. The straightforward captioning process in our current framework may not fully capture these subtle visual nuances. However, the modular design of our framework allows for seamless integration of advanced cross-modal chain-of-thought (CoT) prompting strategies to address this issue.

We demonstrate this by enhancing our Caption-based Grouper with FineR (Liu et al., 2024e), a cross-modal CoT prompt method specifically designed for fine-grained visual recognition. When the proposer identifies fine-grained criteria, such as `Bird species`, the framework switches to a FineR-enhanced captioning strategy that provides more detailed attribute descriptions, such as "Wing color: Blue-grey," to enrich the captions and capture per-attribute visual characteristics to better support the subsequent substructure uncovering process.

We evaluate this on two image collections containing fine-grained criteria: CUB200 (Wah et al., 2011) and Stanford Cars196 (Khosla et al., 2011). Our framework successfully discovers the fine-grained criteria `Bird species` for CUB200 and `Car model` for Cars196. As shown in Tab. 31, when uncovering fine-grained substructures, integrating the FineR prompting strategy significantly improves performance by up to +15.0% CAcc and +12.2% SAcc, achieving results comparable to

Table 30: **Ablation study of multi-granularity refinement on the six SMC benchmarks.** We compare three ways of constructing cluster names: Initial Names (IN), Flat Refinement (FR), Multi-granularity Refinement (MR). We report Clustering Accuracy (CAcc) and Semantic Accuracy (SAcc)as percentages (%). Average (*Avg.*) CAcc and SAcc across different criteria on each dataset is also provided. These expanded results correspond to the plotting shown in Fig. 7.

| Benchmark | Criterion | IN | | FR | | MR | |
|---|---|---|---|---|---|---|---|
| | | CAcc | SAcc | CAcc | SAcc | CAcc | SAcc |
| COCO-4c | Activity | 14.1 | 48.5 | 34.5 | 40.5 | 44.1 | 48.9 |
| | Location | 30.0 | 51.9 | 41.4 | 56.0 | 55.2 | 55.6 |
| | Mood | 6.6 | 34.7 | 21.9 | 32.1 | 38.1 | 32.6 |
| | Time of Day | 24.4 | 50.5 | 28.2 | 54.4 | 67.6 | 56.7 |
| | *Avg.* | 18.8 | 46.4 | 31.5 | 45.8 | **51.2** | **48.4** |
| Food-4c | Food Type | 33.9 | 52.4 | 35.5 | 54.3 | 34.6 | 54.2 |
| | Cuisine | 30.6 | 39.7 | 27.6 | 36.5 | 47.0 | 65.9 |
| | Course | 52.9 | 81.1 | 62.8 | 83.0 | 69.1 | 85.7 |
| | Diet | 14.0 | 46.6 | 36.8 | 58.2 | 41.5 | 54.0 |
| | *Avg.* | 32.9 | 55.0 | 40.7 | 58.0 | **48.1** | **64.9** |
| Clevr-4c | Color | 56.5 | 49.7 | 60.9 | 53.0 | 70.3 | 63.4 |
| | Texture | 56.5 | 26.0 | 60.9 | 33.0 | 65.3 | 42.1 |
| | Count | 56.5 | 39.6 | 56.5 | 40.8 | 65.7 | 73.3 |
| | Shape | 47.8 | 33.6 | 47.8 | 41.8 | 58.4 | 38.5 |
| | *Avg.* | 54.3 | 37.2 | 56.5 | 42.2 | **64.9** | **54.3** |
| Action-3c | Action | 72.2 | 63.6 | 90.5 | 63.0 | 82.8 | 76.3 |
| | Location | 46.0 | 50.4 | 65.9 | 59.3 | 69.8 | 55.2 |
| | Mood | 20.6 | 41.9 | 46.0 | 51.0 | 52.3 | 50.2 |
| | *Avg.* | 46.3 | 52.0 | 67.5 | 57.8 | **68.3** | **60.6** |
| Card-2c | Suit | 40.9 | 50.1 | 45.7 | 45.7 | 54.5 | 73.6 |
| | Rank | 43.0 | 55.1 | 47.7 | 54.6 | 92.1 | 95.1 |
| | *Avg.* | 42.0 | 52.6 | 46.7 | 50.2 | **73.3** | **84.3** |
| Fruit-2c | Species | 59.2 | 68.6 | 64.1 | 67.0 | 76.9 | 70.7 |
| | Color | 41.8 | 56.7 | 44.7 | 42.3 | 53.3 | 51.5 |
| | *Avg.* | 50.5 | 62.7 | 54.4 | 54.7 | **65.1** | **61.1** |

Table 31: **Study of substructure discovery for fine-grained criteria.** We report clustering accuracy (CAcc) and semantic accuracy (SAcc) as percentages (%). The pseudo upper-bound (UB) performance is obtained using CLIP (Radford et al., 2021) ViT-L/14 in a zero-shot transfer setting with the ground-truth class names. †: We compare with FineR (Liu et al., 2024e) without its post-class name refinement step to ensure a fair comparison.

| | CUB200 | | Car196 | |
|---|---|---|---|---|
| | CAcc | SAcc | CAcc | SAcc |
| UB | 57.4 | 80.5 | 63.1 | 66.3 |
| FineR† | 44.8 | 64.5 | **33.8** | **52.9** |
| Ours | 30.1 | 56.7 | 21.3 | 35.9 |
| Ours + FineR | **45.1** | **68.9** | 31.1 | 47.3 |

FineR itself. This demonstrates the flexibility of our system, allowing future adaptations to specific application needs, such as fine-grained image collections.

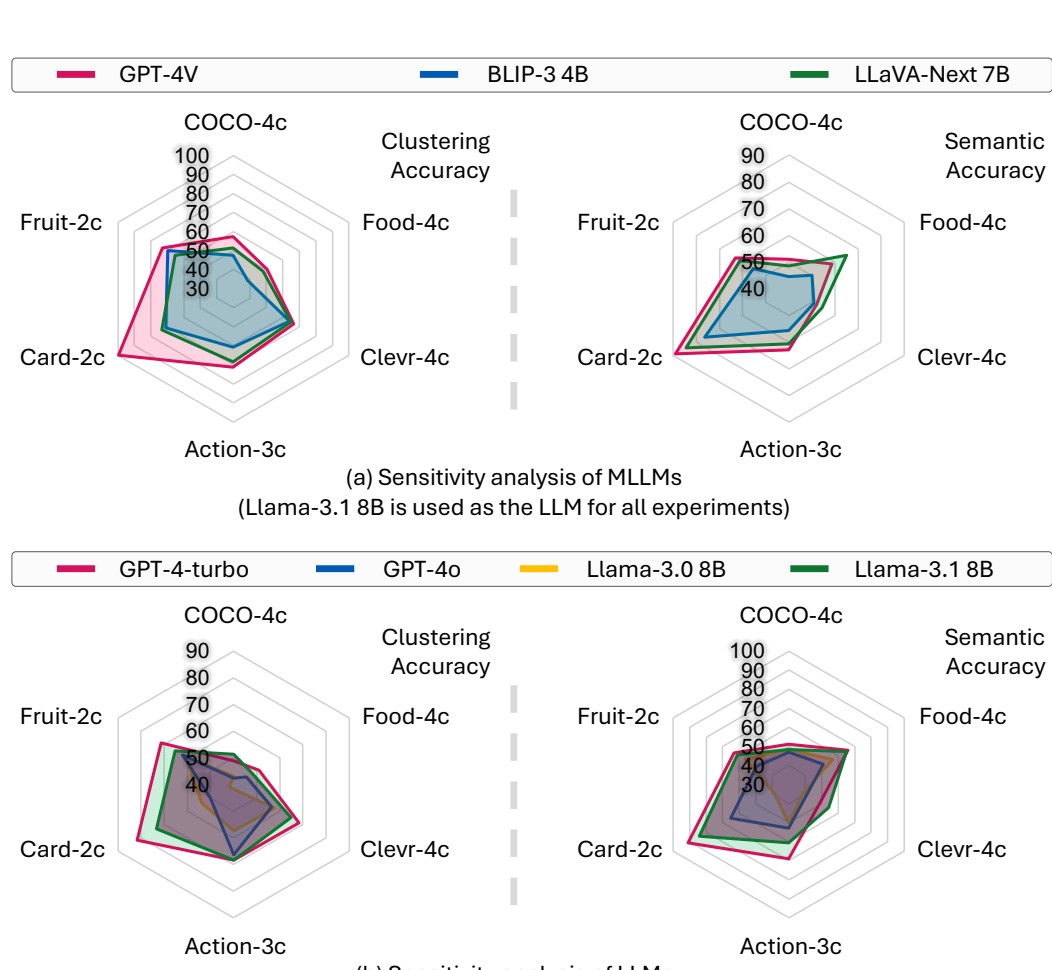

(a) Sensitivity analysis of MLLMs
(Llama-3.1 8B is used as the LLM for all experiments)

(b) Sensitivity analysis of LLMs
(LLaVA-Next 7B is used as the MLLM for all experiments)

Figure 11: **Sensitivity analysis of different MLLMs and LLMs on the six SMC benchmarks. Top (a):** We fix the LLM to Llama-3.1-8B and study the impact of different MLLMs. **Bottom (b):** We fix the MLLM to LLaVA-NeXT-7B and study the impact of different LLMs. The average clustering accuracy(%) across different criteria is reported on the **left**, while the average semantic accuracy(%) is reported on the **right**.

## G    FURTHER QUALITATIVE RESULTS

In this section, we visualize the grouping results predicted by the best configuration of our proposed framework (Caption-based Proposer and Caption-based Grouper). Specifically, we present example clustering results across different criteria for COCO-4c in Fig. 12, Food-4c in Fig. 13, Action-3c in Fig. 14, Clevr-4c in Fig. 15, and Card-2c in Fig. 16. Additionally, we showcase example clustering results at different predicted granularity levels for COCO-4c in Fig. 17.

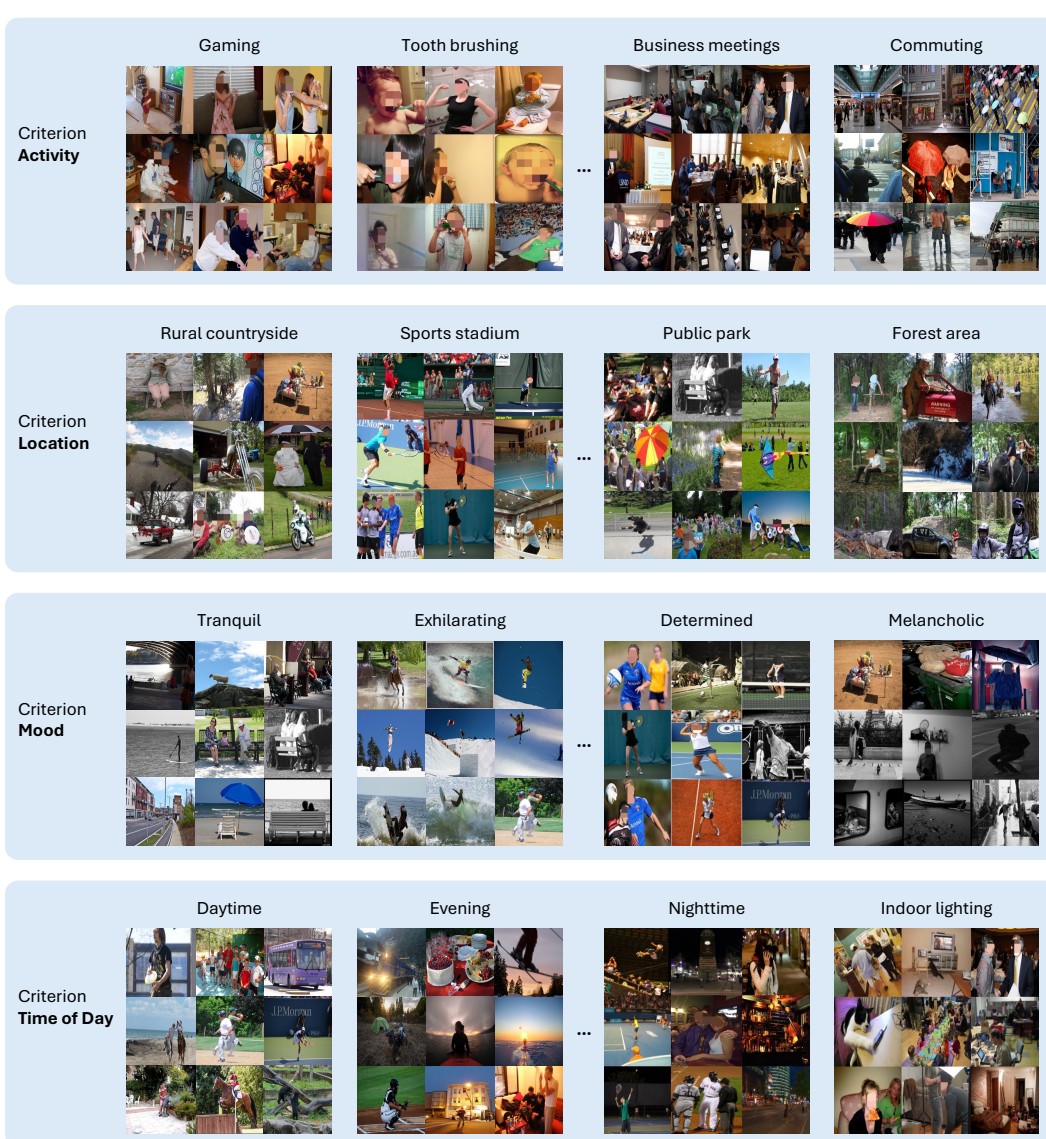

Figure 12: **Example predicted clusters of COCO-4c.**

## H    FURTHER FAILURE CASE ANALYSIS

In Fig. 18, we present several failure cases from the best configuration of our proposed framework (Caption-based Proposer and Caption-based Grouper). As observed, our method frequently mis-assigns "Surfing" to the "Kayaking" cluster under the `Activity` criterion. Upon examining the intermediate criterion captions generated by the MLLM, we found that this error is largely due to the MLLM incorrectly describing a "Surfboard" as a "Kayak". This highlights the importance of the MLLM's ability to accurately describe images, as it is critical for the performance of our system.

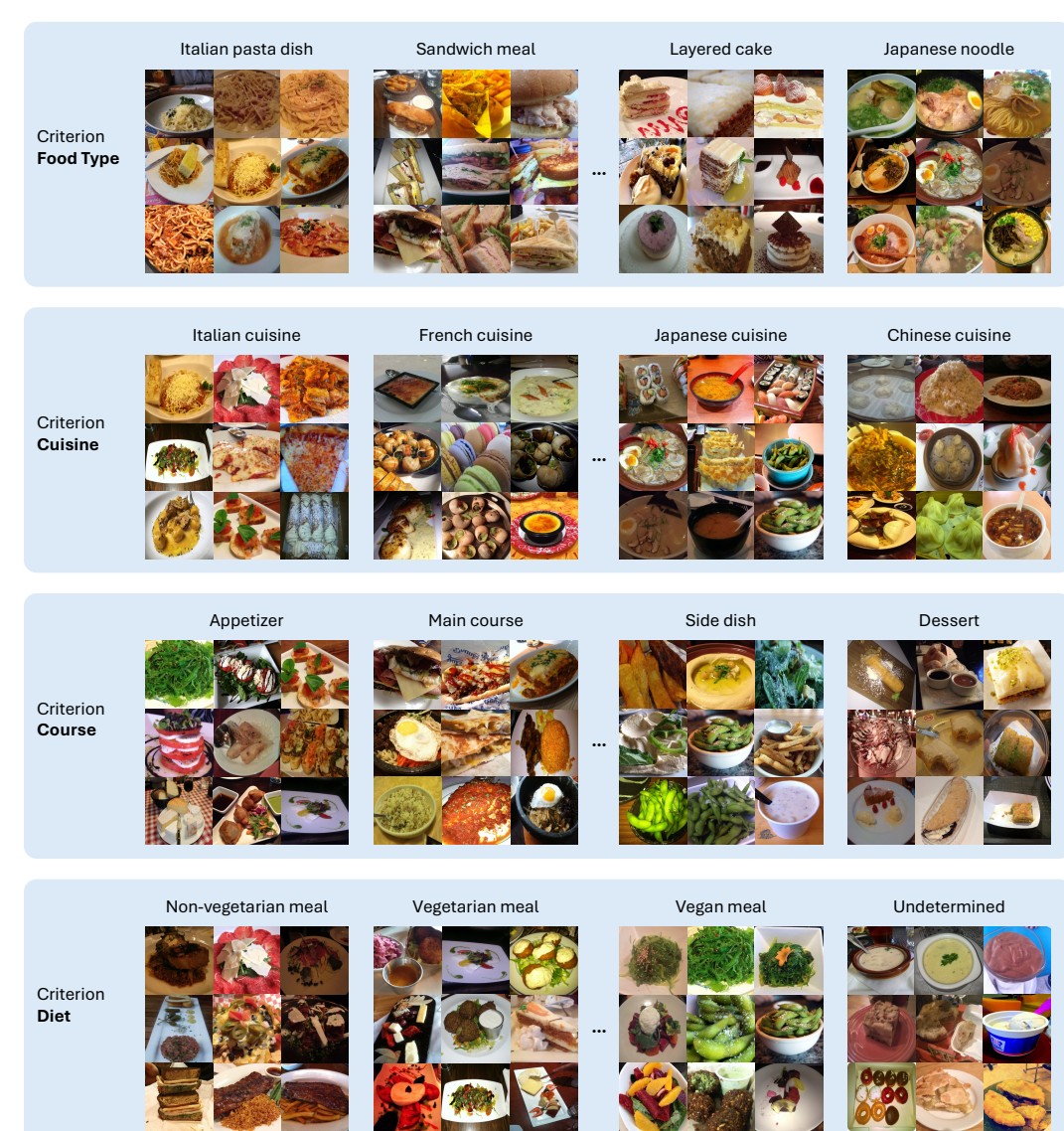

Figure 13: **Example predicted clusters of Food-4c.**

Potential improvements could include majority voting or model ensembling using different MLLM models.

Another issue arises in crowded scenes. When multiple people are present in an image, the model consistently assigns the Mood label "Communal" to the images. We speculate that this occurs because, in the presence of multiple people, the model struggles to accurately determine the mood of one key individual.

Finally, we observed that our method sometimes fails to distinguish subtle, fine-grained differences between images, leading to incorrect labels. For example, as shown in Fig. 18, "Edamame" or "Pho" are typical dishes from China, Vietnam, and Japan, but they may be presented differently depending on the cuisine. The "Edamame" shown in Fig. 18 is presented in a traditional Japanese style, yet our model incorrectly predicted it as Chinese cuisine. This oversight of fine-grained details could be improved by employing a more advanced prompting strategy (Liu et al., 2024e).

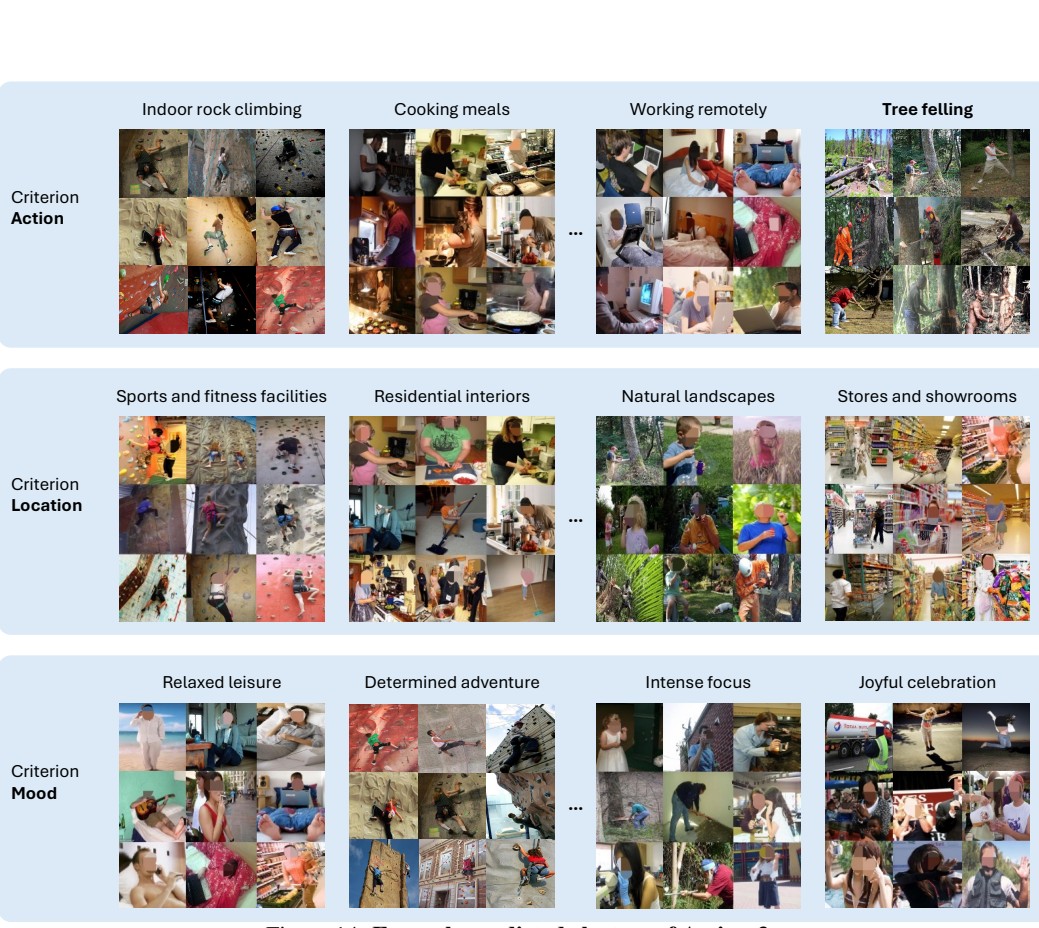

Figure 14: **Example predicted clusters of Action-3c.**

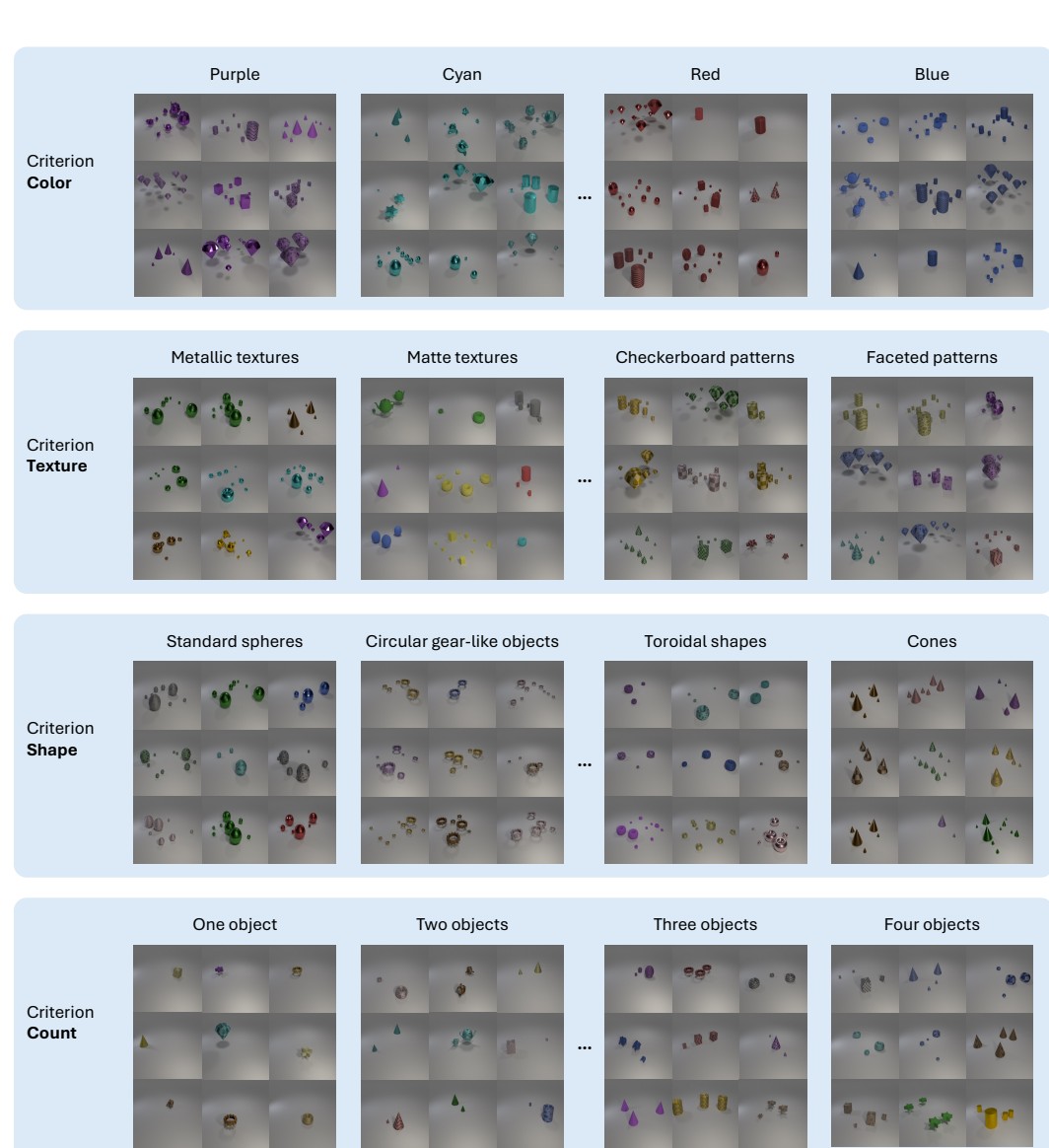

Figure 15: **Example predicted clusters of Clevr-4c.**

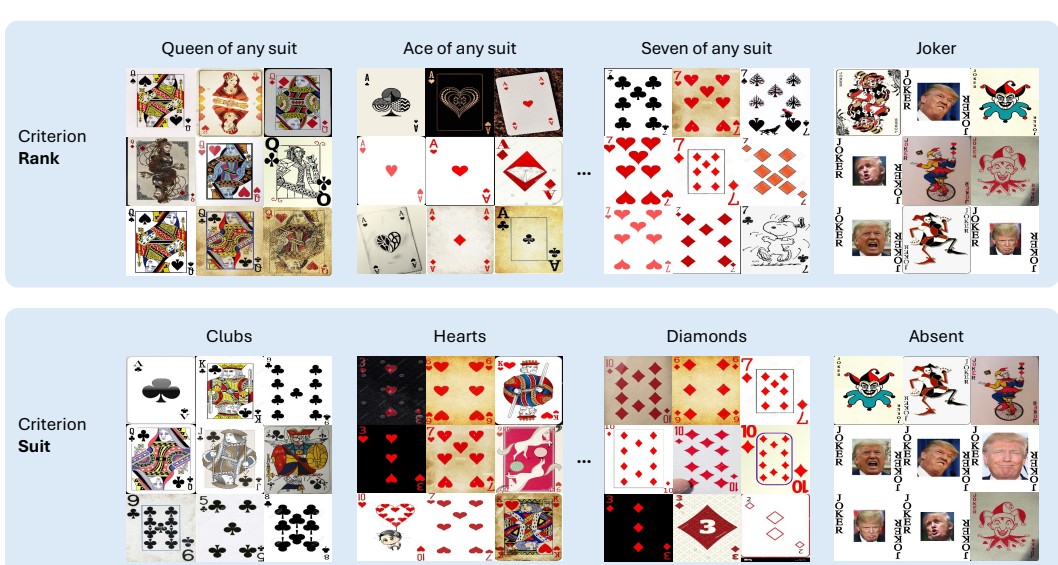

Figure 16: **Example predicted clusters of Card-2c.**

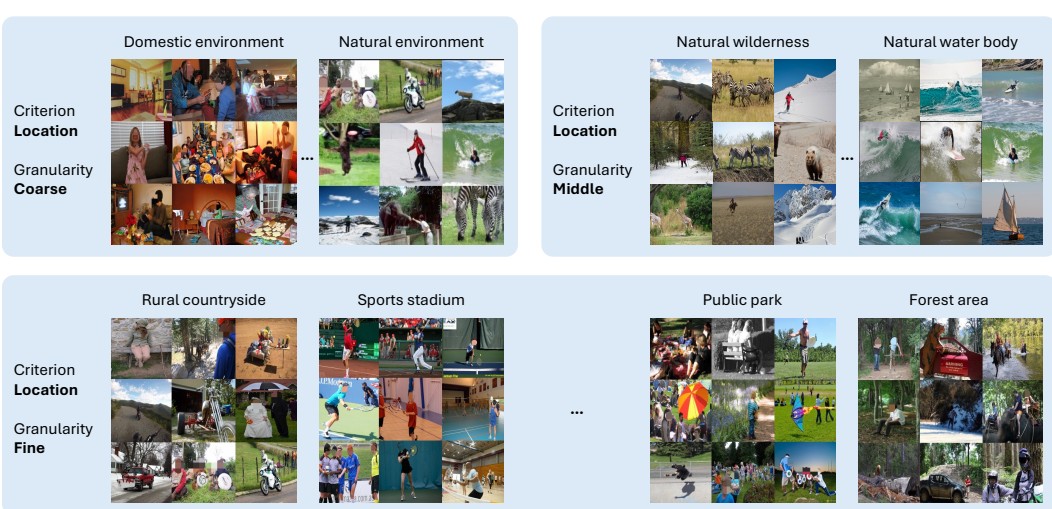

Figure 17: **Example predicted clusters of COCO-4c at different granularities.**

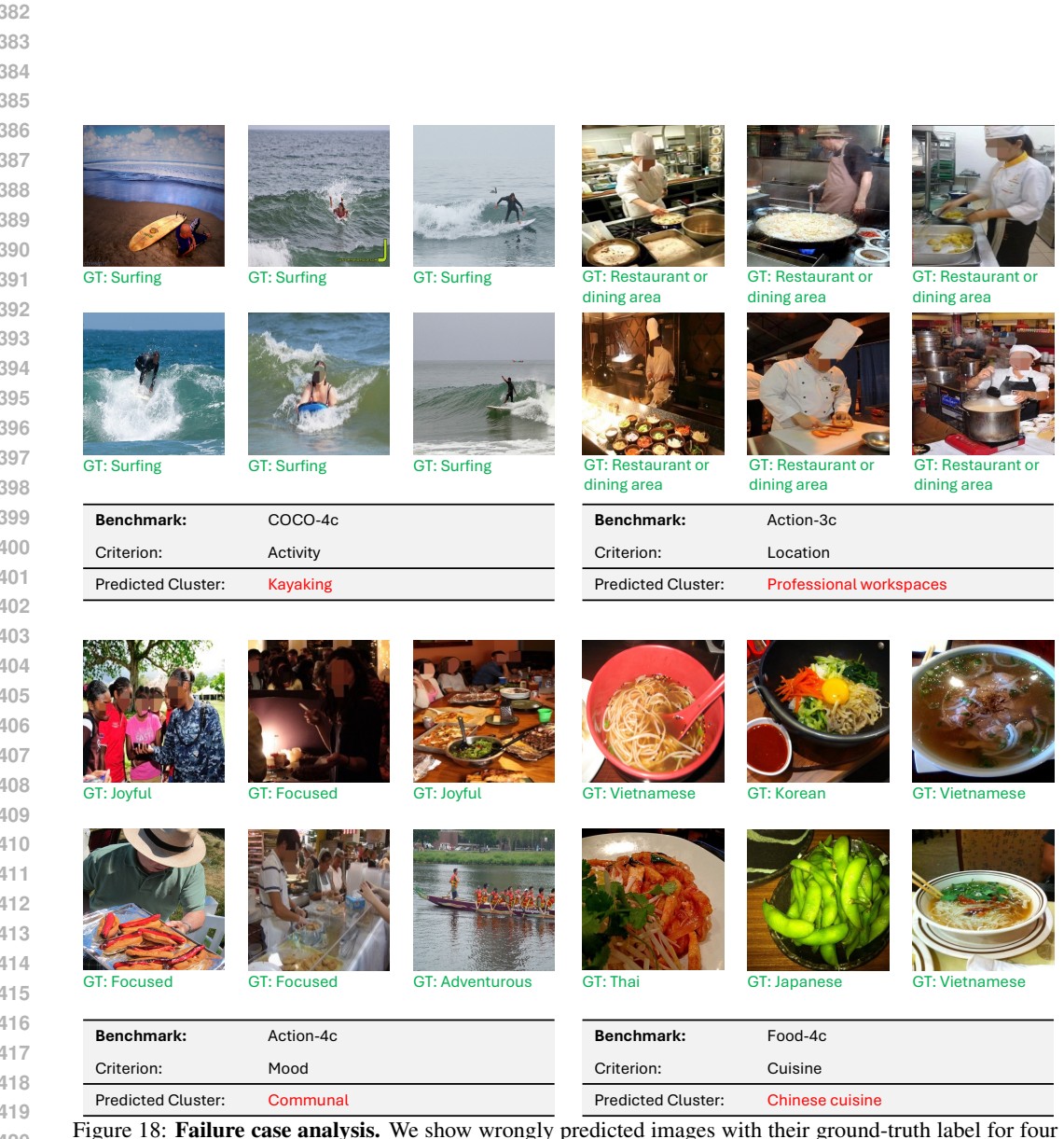

Figure 18: **Failure case analysis.** We show wrongly predicted images with their ground-truth label for four clusters.

## I    FURTHER DETAILS ON APPLICATION STUDY

In this section, we present additional implementation details, evaluation results, and findings for the application study discussed in Sec. 5 of the main paper. Specifically, Sec. I.1 offers further evaluation results and implementation details on using our predicted distribution to train a debiased model with GroupDRO (Sagawa et al., 2020). Sec. I.2 outlines the implementation of the user study that assesses the alignment between predicted biases and human judgments, along with comprehensive findings for all studied occupations and identified criteria. Finally, Sec. I.3 provides additional insights from the analysis of social media image popularity.

### I.1    FURTHER DETAILS ON DISCOVERING AND MITIGATING DATASET BIAS

In this section, we provide additional evaluation results and implementation details for the application study presented in Sec. 5 of the main paper.

**Additional Evaluation:** To further evaluate the prediction quality of our method for hair color and gender, we used the ground-truth labels from the CelebA dataset (Liu et al., 2015) to assess the classification accuracy of them. Our method achieved an impressive classification accuracy of 99.1% for gender and 87.4% for hair color on the 162,770 training images, demonstrating its effectiveness for uncovering gender and hair color substructures within the training set.

In addition, we quantified the *spurious correlation* between hair color and gender using the metric proposed by Yang et al. (2023). Specifically, given the correlated gender attribute distribution $A$ and the target hair color distribution $Y$, we computed the normalized mutual information between $A$ and $Y$ to quantify the spurious correlation as:

$$I(A;Y) = \frac{2I(A;Y)}{H(A) + H(Y)} \tag{1}$$

where $H(A)$ and $H(Y)$ represent the normalized entropy of the gender and hair color distributions, respectively. A value of $H(A)$ or $H(Y)$ equal to 1 indicates a uniform distribution (*i.e.*, no class imbalance). We then used the ground-truth distribution from the dataset's labels and our predicted distribution to estimate the spurious correlation intensity using the score from Eq. 1. For gender and hair color, our method's predictions yielded a score of $I_{Pred} = 0.10$, which is nearly identical to the ground-truth score of $I_{GT} = 0.11$. This demonstrates that our method effectively identifies and confirms the bias directly from the training set.

**Implementation Details of Training GroupDRO:** To conduct debiased training using Group-DRO (Sagawa et al., 2020), we first used our predicted distribution to define four distinct training groups, rather than relying on the ground-truth distribution. We closely followed the training protocol outlined in B2T (Kim et al., 2024) and GroupDRO (Sagawa et al., 2020). Specifically, we fine-tuned a ResNet-50 (He et al., 2016) model pre-trained on ImageNet (Deng et al., 2009), using the training split of the CelebA dataset (Liu et al., 2015). The training was performed using the SGD optimizer (Ruder, 2016) with a momentum of 0.9, a batch size of 64, and a learning rate of $1 \times 10^{-5}$. We applied a weight decay of 0.1 and set the group adjustment parameter to zero. The model was trained over 50 epochs. For evaluation, we reported both the average and worst-group test accuracies, selecting the model from the epoch that achieves the highest worst-group accuracy on the validation set. The final evaluation and comparison results are provided in Fig. 8.

### I.2    FURTHER DETAILS ON DISCOVERING NOVEL BIAS IN TEXT-TO-IMAGE DIFFUSION MODELS

**Image Generation for the Subject Occupation:** Following prior studies (Bianchi et al., 2023; Bolukbasi et al., 2016), we selected nine occupations for our study: three stereotypically biased towards females (Nurse, Cleaning staff, Call center employee), three biased towards males (CEO, Firefighter, Basketball player), and three considered gender-neutral (Teacher, Computer user, Marketing coordinator). We used two state-of-the-art T2I diffusion model, DALL·E3 (Betker et al., 2023) and Stable Diffusion (SDXL) (Podell et al., 2024) to generate 100 images for each occupation for our study. This resulted in a total of 1,800 images. For each occupation, we provide some examples of images generated by DALL·E3 in Fig. 21, while provide some examples of images generated by

Figure 19: **Bias quantification results and human evaluation** for each occupation and criterion across the two studied T2I models, DALL·E3 and SDXL. The bias intensity score is reported.

SDXL in Fig. 22. We only used the simple prompt "A portrait photo of a <OCCUPATION>" for image generation for all occupations and did not include any potential biases in the prompt.

**Bias Discovery and Quantification:** We applied our method to 1,800 generated images and automatically identified 10 grouping criteria (bias dimensions) along with their predicted distributions for each occupation image set. For this study, we utilized the mid-granularity output of our system. To evaluate the biases, we first identified the dominant cluster for each criterion—the cluster containing the largest number of images—as the *bias direction*. We then calculated the normalized entropy of the distribution for each criterion of the occupation's images to determine the *bias intensity* score, following the method proposed by D'Incà et al. (2024):

$$\mathcal{H}_{bias}^l = 1 + \frac{\sum_{c^l \in \mathcal{C}^l} \log(p(c^l | \mathcal{C}^l, \mathcal{D}_{\text{Occupation}}))}{\log(|\mathcal{C}^l|)} \tag{2}$$

where $\mathcal{D}_{\text{Occupation}}$ represents the generated images for each occupation, $\mathcal{C}^l$ denotes the clusters discovered under the $l$-th criterion, and $p(c^l | \mathcal{C}^l)$ is the probability of each cluster under the current distribution. The resulting score is bounded between $\mathcal{H}_{bias}^l \in [0, 1]$, where 0 indicates no bias towards a specific cluster (concept) under the evaluated criterion, and 1 indicates that the images are completely biased towards a particular cluster (concept) (*e.g.*, "Grey" hair color) within the current bias dimension (e.g., Hair color). We used the score defined in Eq. 2 to quantify the biases for each occupation across the 10 discovered grouping criteria. We report the bias intensity score for each occupation and each model across the 10 discovered grouping criteria in Fig. 19.

**Human Evaluation Study Details:** To assess the alignment between our method's predictions and human judgments on bias detection, we conducted a user study to gather human evaluation results for the generated images. As shown in the questionnaire example in Fig. 23, participants were presented with images generated by DALL-E3 and SDXL for each occupation and were asked to identify the bias direction (dominant class) for each of the 10 discovered criteria and rate the bias intensity on a scale from 0 to 10. We collected responses from 54 anonymous participants, resulting in 6 human evaluations for each occupation and each criterion.

The Absolute Mean Error (AME) between the bias intensity scores predicted by our system and those rated by humans (scaled to 0 to 1) was 0.1396. Additionally, our system's predicted bias directions aligned with human evaluations 72.3% of the time, with most discrepancies occurring in the criteria of "Age group," "Skin tone," and "Accessories worn." These findings indicate a strong correlation between our system's predictions and human judgments, validating the effectiveness of our approach. Detailed user study results are provided in Sec. I.2. We believe the discrepancies in certain criteria may be due to the influence of personal subjective cognition on respondents' answers. In Fig. 19, we present the human evaluation results, averaged across all participants for each model, occupation, and criterion, with the human ratings scaled from 0 to 1.

**Complete Results and Additional Findings:** In Fig. 19, we present the detailed bias detection results for each model, occupation, and criterion, alongside human evaluation scores for reference. A particularly interesting phenomenon emerges: *While DALL·E3 significantly outperforms SDXL on the **well-known** bias dimensions* (*e.g.*,, Gender, Race, Age, and Skin tone), *both DALL·E3 and SDXL exhibit moderate to strong biases along the **novel** bias dimensions* (*e.g.*,, Hair color, Mood, Attire, and Accessories).

We speculate that DALL·E3's superior performance in mitigating well-known biases may be attributed to its "guardrails" (OpenAI, 2022b), designed as part of its industrial deployment to avoid amplifying social biases via its easily accessible APIs. However, these guardrails do not prevent it from exhibiting biases along the novel dimensions discovered by our method, as these dimensions remain understudied. This observation highlights the importance of studying novel biases that could potentially exist in widely used T2I generative models to prevent further bias amplification.

## I.3 FURTHER DETAILS ON ANALYZING SOCIAL MEDIA IMAGE POPULARITY

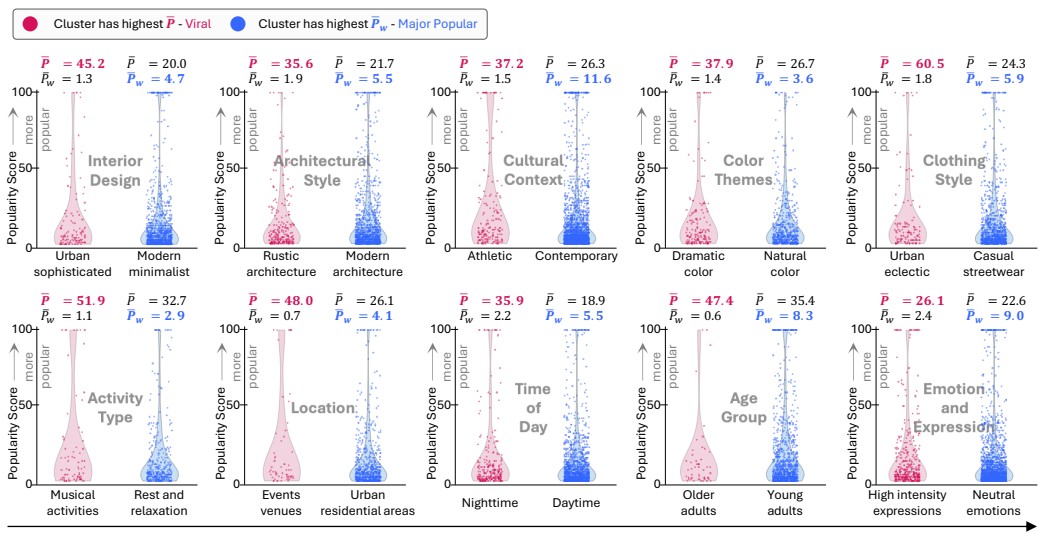

Figure 20: **Complete analysis of social media photo popularity on the SPID dataset.** We display the *viral* and *major popular* clusters, along with the popularity distribution of data points within these clusters across all *ten* discovered criteria (in Grey).

With the rise of image-centric content on social media platforms like Instagram, Flickr, and TikTok, understanding what makes an image popular has become crucial for applications such as marketing, content curation, and recommendation systems. Traditional research often approaches image popularity as a regression problem (Ortis et al., 2019; Cheng et al., 2024), utilizing metadata like hashtags, titles, or follower counts. However, the specific semantic visual elements that contribute to an image's popularity remain largely unexplored. In this study, we applied our proposed method to automatically categorize social media images based on semantic visual elements across different dimensions (criteria). By analyzing these interpretable results alongside image popularity metrics (*e.g.*,, number of views), we gained insights into the factors contributing to virality and identified common visual traits among popular images. These insights can provide valuable guidance for content creators and advertisers, enhancing productivity and informing strategic decision-making.

To expand on the discussion in Sec. 5 of the main paper, we present the complete findings across all ten discovered criteria in Fig. 20. Notably, we consistently observed a sharp semantic contrast between the visual elements in viral images and those in the majority of popular images across all ten criteria. For instance, there is a contrast between *Urban sophisticated* and *Modern minimalist* under Interior Design, *Rustic architecture* and *Modern architecture* under Architecture Style, and *Event venues* versus *Urban residential areas* under Location.

This recurring observation reinforces the idea that viral content tends to capture more attention, likely because it features novel, surprising, or striking visual elements. Humans are inherently attracted to

stimuli that deviate from the norm (Priester et al., 2004; Bruni et al., 2012; Palmer & Gore, 2014). On the other hand, widely popular yet "neutral" content is shared more often due to its familiarity and broad appeal, though it is less likely to provoke the strong emotional responses that fuel virality. We believe the insights generated by our method could offer valuable guidance to social media platform practitioners, helping them tailor their content more effectively to target audiences and gain a deeper understanding of social media image trends from various perspectives.

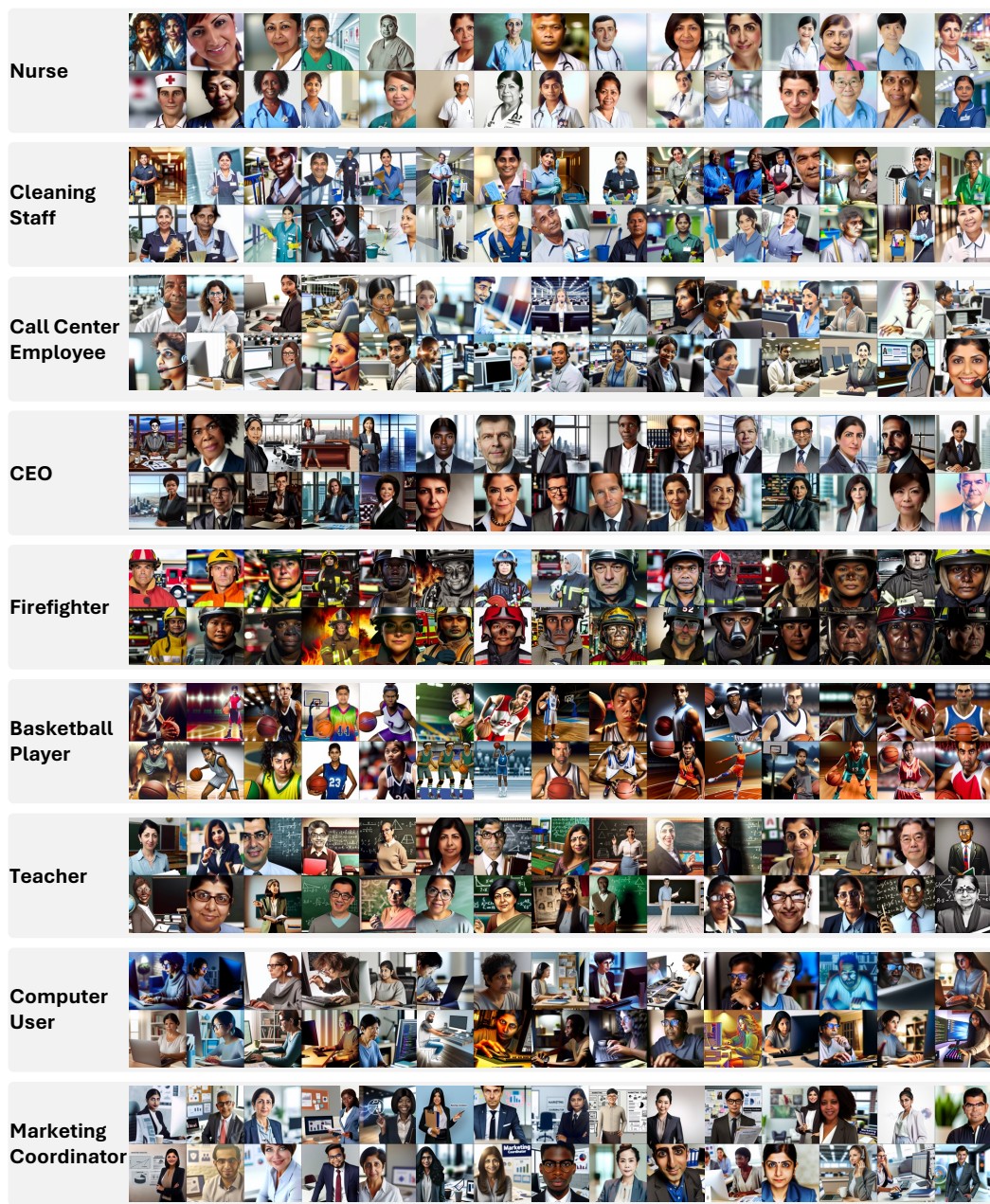

Figure 21: **Samples of DALL·E3 generated images.** For each occupation, the simple prompt "A portrait photo of a <OCCUPATION>", that does not reference any potential bias dimensions such as gender, race or hair color, is fed to DALL·E3 to generate 100 images. We present a random sample of 30 generated images.

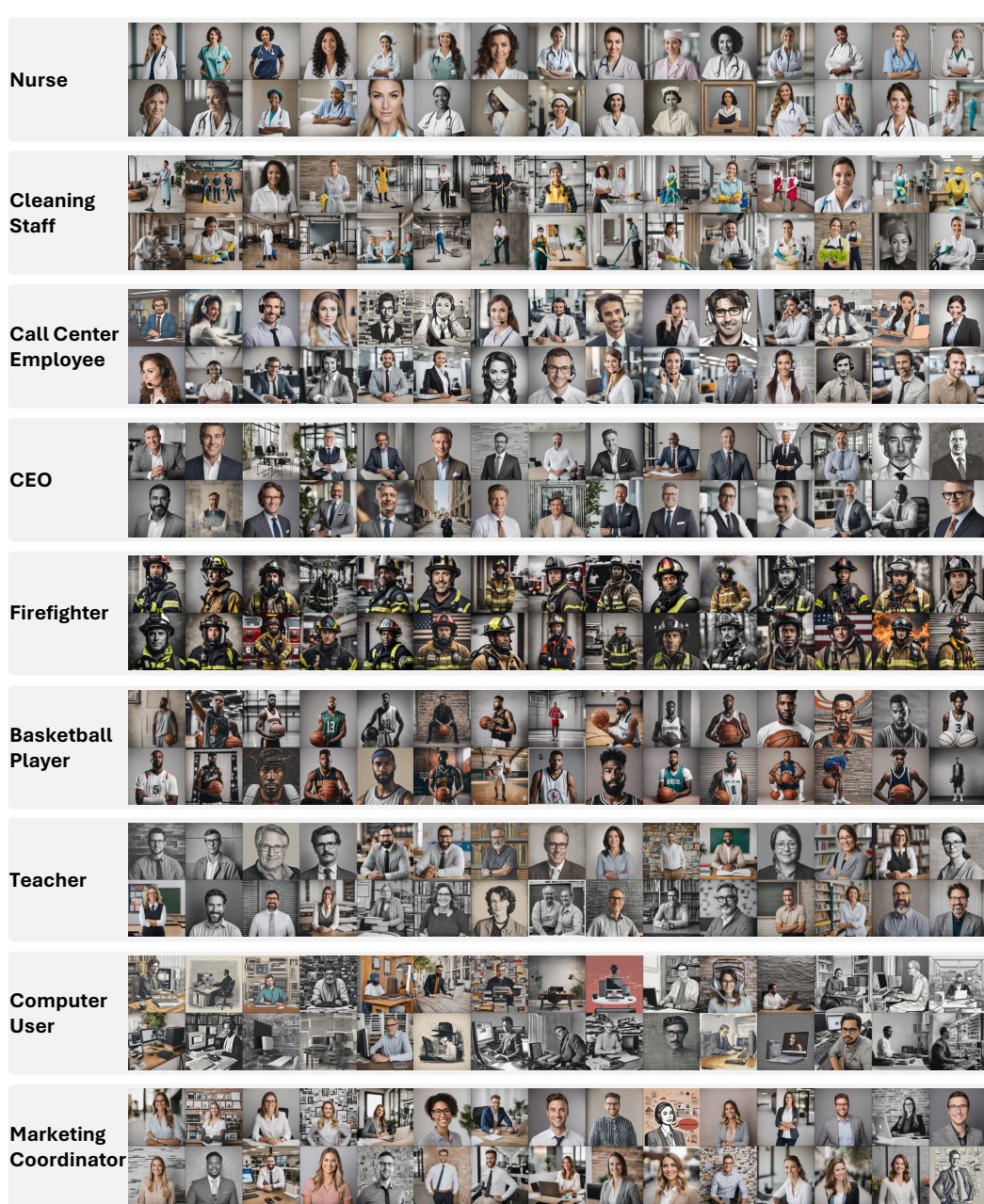

Figure 22: **Samples of SDXL generated images.** For each occupation, the simple prompt "A portrait photo of a <OCCUPATION>", that does not reference any potential bias dimensions such as gender, race or hair color, is fed to SDXL to generate 100 images. We present a random sample of 30 generated images.

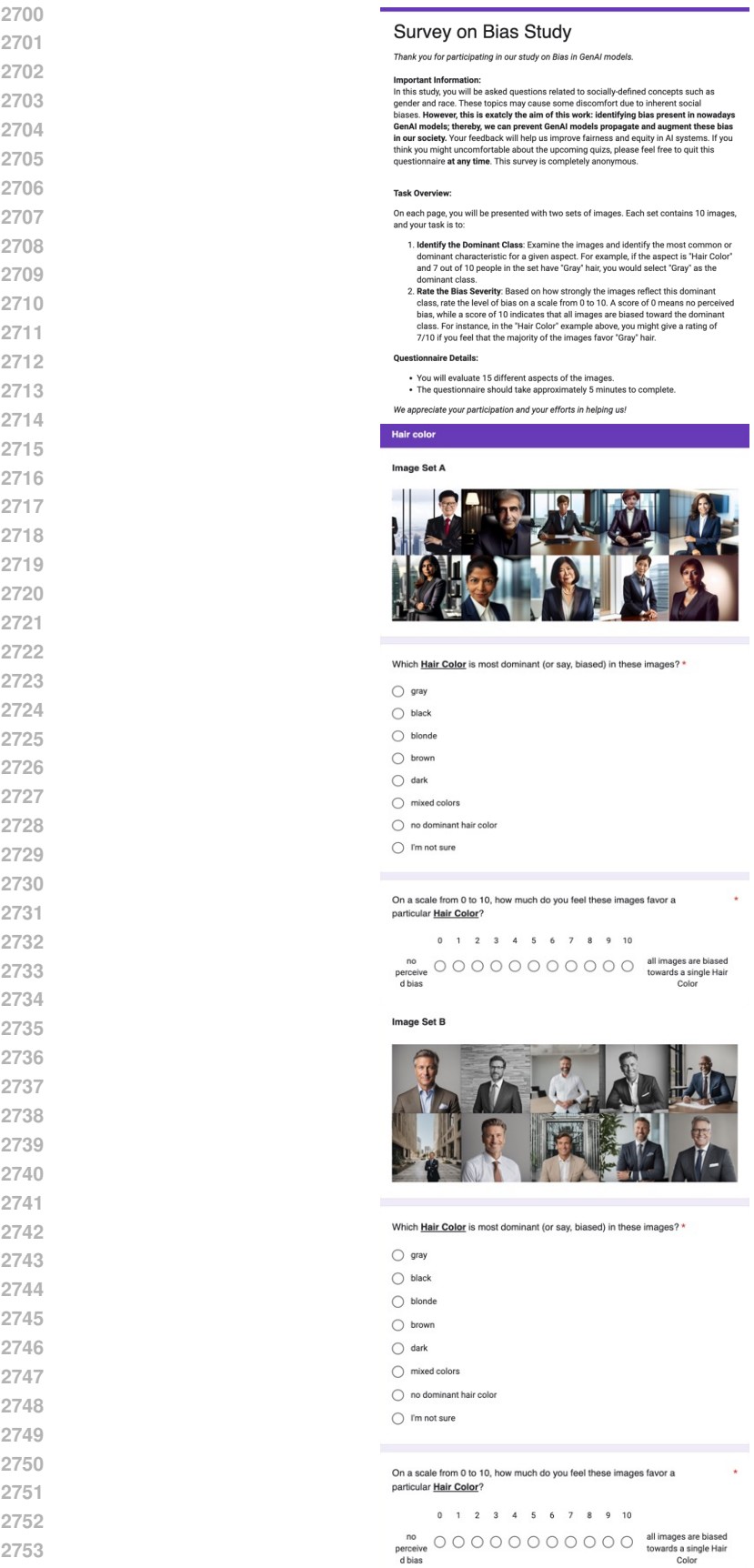

Figure 23: **Example of the questionnaire for human evaluation study.**

## J IN-DEPTH DISCUSSION OF RELATED SETTINGS

**Distinction from Multiple Clustering:** Multiple clustering involves finding diverse clusterings from the same dataset in either a semi-supervised (Bae & Bailey, 2006; Ren et al., 2022) or unsupervised (Caruana et al., 2006; Mautz et al., 2020; Yao et al., 2023) manner. However, multiple clustering methods *require* users to specify the number of clusterings and clusters (or rely on ground-truth annotations for evaluation) (Yu et al., 2024). In other words, $L$ (the number of clusterings) and $K_l$ (the number of clusters within each clustering) are assumed as prior knowledge in multiple clustering settings. This strong assumption creates a *chicken-or-egg dilemma* and is often impractical in real-world applications: *Users want to employ multiple clustering methods to understand their data, but how can they predefine the number of valid clusterings and clusters without already having a deep understanding of the entire dataset?* Although some strategies exist to determine the number of clusters, most of these methods (Monti et al., 2003; Zhang et al., 2017; Liu et al., 2024e) only work for single clustering. In contrast, SMC requires the model to automatically *discover both* the number of clusterings and the clusters within them.

Additionally, while multiple clustering methods can reveal diverse sample division patterns and underlying data structures, they do *not* provide interpretations of the results—specifically, what rules the output clustering follows and the semantic meaning of the clusters. As a result, users often need to manually investigate the clustering outcomes. In stark contrast, SMC methods provide interpretability by describing both the clustering rules and the semantic meanings of the clusters in natural language. This not only offers a more comprehensive understanding of the data but also allows users to combine different clusterings to gain deeper insights into the data distribution.

Although recent work such as IC|TC Kwon et al. (2024) and MMaP (Yao et al., 2024) allows users to specify clustering criteria and propose text-conditioned clustering based on user-defined criteria and cluster numbers, these approaches do not resolve the *dilemma* of traditional multiple clustering. They still require users to have prior knowledge of large image collections, which is impractical in many real-world scenarios. In contrast, our method *automatically* discovers clustering criteria, expressed in natural language, from unstructured image collections and provides interpretable results, allowing users to freely explore their data.

**Distinction from Multi-label Zero-shot Classification:** SMC differs from multi-label zero-shot classification in that the latter (Lee et al., 2018; Huang et al., 2020; Ali & Khan, 2023; Gupta et al., 2023) requires predefined sets of classes under different rules, with the goal being to assign each image to multiple classes from these sets. In contrast, SMC requires *discovering* the class names (or cluster semantics). In fact, multi-label zero-shot classification can be viewed as a specific instance of SMC when the user explicitly and precisely defines all the clusters and their semantic meanings under different clustering rules.

## K FURTHER COMPUTATIONAL COST ANALYSIS

The proposed TeDeSC framework is training-free, requiring only inference processes. Specifically, our main framework (Caption-based) requires up to 31 GB of GPU memory to operate. All experiments reported in the paper were conducted on 4 Nvidia A100 40GB GPUs. In Tab. 32, we provide a detailed analysis of the computational efficiency of our main TeDeSC framework (Caption-based Proposer and Caption-based Grouper) on the COCO-4c benchmark (5,000 images with four criteria) across various hardware configurations. For these experiments, we used LLaVA-NeXT-7B (Liu et al., 2024b) as the MLLM and Llama-3.1-8B (Meta, 2024b) as the LLM.

As shown in Tab. 32, organizing 5,000 images based on all four discovered criteria can be completed by TeDeSC in 29.1 hours on a single A100 GPU or 16.7 hours on a single H100 GPU. More importantly, most steps in our framework, such as per-image captioning and per-caption cluster assignment, are parallelizable across multiple GPUs, significantly accelerating the process. Therefore, when parallelizing the framework on 4 A100 or H100 GPUs, we achieve approximately a $4\times$ speedup, reducing computational time to 7.6 hours on 4 A100 GPUs and 4.3 hours on 4 H100 GPUs.

Table 32: **Computational cost analysis on the COCO-4c benchmark (5,000 images with four criteria).** We report the average and total time costs on various machines. The time costs were calculated for organizing all 5,000 images according to all the 4 criteria. Our main caption-based TeDeSC framework is used in this experiment.

| Method | Hardware | Average time cost (sec/img) ↓ | Total time cost (hrs) ↓ |
|--------|----------|-------------------------------|--------------------------|
| TeDeSC | 1 Nvidia A100-40GB | 20.9 | 29.1 |
| | 4 Nvidia A100-40GB | 5.5 | 7.6 |
| | 1 Nvidia H100-80GB | 12.0 | 16.7 |
| | 4 Nvidia H100-80GB | 3.1 | 4.3 |

## L  FUTURE WORK

**Closed-Loop Optimization.** In this work, we designed our prompts following the Iterative Prompt Engineering methodology (DeepLearning.AI, 2024) introduced by Isa Fulford and Andrew Ng. In App. C, we provide the exact LLM and MLLM prompts used in our framework and break down each prompt to explain the objectives and purposes behind each design choice. These explanations cover elements such as system prompts, input formatting, task and sub-task instructions, and output instructions. Our focus in this work is on creating a highly generalizable framework, TeDeSC, and we do not perform any closed-loop, dataset-specific prompt optimizations. However, in future work or application scenarios where a labeled training/validation dataset is available, practitioners could build upon our design objectives. By leveraging our proposed evaluation metrics (see Sec. 4) for each step, it would be possible to develop a Semantic Multiple Clustering (SMC) system with a closed-loop optimization pipeline to achieve improved performance.

**TeDeSC on Other Data Types.** The core idea of our proposed framework, TeDeSC, is to *use text as a proxy (or medium)* for reasoning over large volumes of unstructured data, generating human-interpretable insights at scale. As such, TeDeSC can be directly applied to textual data (*e.g.,*, documents). Moreover, since natural language is a highly versatile and widely-used medium of representation, TeDeSC can be extended to other data types by converting these data into text (by replacing the captioning module with suitable tools) in future work, such as:

- **Audio Data:** Speech-to-Text models like Whisper (Radford et al., 2023) can convert audio data into text, enabling subsequent analysis with TeDeSC.

- **Tabular Data:** Table-to-Text models, such as TabT5 (Andrejczuk et al., 2022), can translate tabular data into text, making it compatible with TeDeSC. For tables containing figures, modern MLLMs like LLaVA-Next, which support both OCR and image-to-text capabilities, can handle these elements to create a unified textual representation for TeDeSC.

- **Protein Structures:** Protein structure-to-text models, such as ProtChatGPT (Wang et al., 2024a), can convert protein sequences into textual descriptions for analysis with TeDeSC.

- **Point Cloud Data:** 3D captioning models, like Cap3D (Luo et al., 2024), can transform point cloud data or rendered 3D models into text, enabling their analysis using TeDeSC.

We believe the versatile nature of TeDeSC has the potential to open up a broad range of applications across diverse data modalities, fostering new directions in future research.

## M  DISCUSSION ON HANDLING INVALID CRITERIA

At the criteria refinement step, *invalid* grouping criteria (False Positives) may be proposed due to hallucinations from large language models (LLMs). While we did not observe hallucinated criteria being introduced during our experiments across six datasets and three application studies, it is important to further investigate the potential impact of such invalid criteria on the proposed TeDeSC system.

To this end, we design and conduct a control experiment using the Fruit-2c dataset (Muresan & Oltean, 2018), where we *artificially* introduced two "hallucinated" invalid grouping criteria (False Positives), `Action` and `Clothing Style`, into the refined criteria pool. These invalid criteria were

then used in the subsequent grouping process to evaluate their effect on our system. We apply the main Caption-based Grouper to group fruit images based on these "hallucinated" criteria.

The grouping results for the two invalid criteria are presented in Tab. 33. As observed, when processing invalid "hallucinated" criteria, nearly all images are assigned to a cluster named "Not visible" by our framework. This occurs because, in the absence of relevant visual content in the images, the MLLM-generated captions do not include descriptors corresponding to the invalid criteria. Consequently, the LLM creates a "Not visible" cluster and assigns the images to it. Since the system provides interpretable outputs, users can easily identify and disregard such invalid groupings. This control experiment highlights the robustness of our system against hallucination in practical scenarios.

Table 33: **Study of the Influence of Invalid Grouping Criteria (False Positives) on the Fruit-2c Dataset.** We report the distribution of predicted groupings under the two "hallucinated" invalid grouping criteria. The main Caption-based Semantic Grouper is used for this experiment. †: For simplicity, all other minority clusters are grouped as "Others".

| Predicted Clusters | Action (%) | Clothing Style (%) |
|---|---|---|
| Not visible | 98.3 | 96.7 |
| Others† | 1.7 | 3.3 |

## N    LIMITATION

**Model hallucination.** LLM hallucination (Wang et al., 2024c) typically occurs when LLMs are tasked with complex queries requiring world knowledge or factual information—for instance, answering a question like "Who was the 70th president of the United States?" might lead to a fabricated response. However, in our system, the use of LLMs is fully grounded in the visual descriptions (tags or captions) of the images. Consequently, the LLM output is strongly constrained to analyzing these visual descriptions, significantly reducing the likelihood of hallucination. That said, LLM hallucination can still have mild effects on clustering results. For example, as discussed in the failure case analysis in Sec. H, the LLM incorrectly grouped "Korean bibimbap" and "Vietnamese rice noodles" under "Chinese cuisine" (see Fig. 18). MLLMs also play a crucial role in our system, as they are responsible for translating images into text for subsequent processing steps. MLLM hallucination (Wang et al., 2024c) typically involves incorrectly identifying the existence of objects, attributes, or spatial relationships within an image. However, since our proposed system operates at the *dataset level* rather than on a per-image basis, it is largely insensitive to such hallucinations, especially at the fine-grained visual detail level. Moreover, as our system is training-free, it can be further enhanced with LLM or MLLM hallucination mitigation techniques, such as the Visual Fact Checker (Ge et al., 2024), which we leave as a direction for future work.

**Model Bias.** Foundation models such as LLMs and MLLMs are known to inherit biases from their training data (Bommasani et al., 2021). In our system, we addressed potential biases using Hard Positive Prompting techniques: *i) MLLM Bias Mitigation:* The MLLM is further prompted to generate criterion-specific captions that focus solely on describing the criterion-related content in each image. This approach constrains the MLLM from generating irrelevant content influenced by inherent biases; *ii) LLM Bias Mitigation:* Similarly, when prompting the LLM to assign image captions to clusters, we condition it to concentrate exclusively on the Criterion depicted in each image (see Tab. 17).

To validate the effectiveness of these bias mitigation techniques, we conducted a fair clustering experiment. Specifically, following Kwon et al. (2024), we sampled images for four occupations (Craftsman, Laborer, Dancer, and Gardener) from the FACET (Gustafson et al., 2023) dataset, which contains images from 52 occupations. For each occupation, we selected 10 images of men and 10 images of women, totaling 80 images, ensuring a ground-truth gender proportion disparity of 0% for each occupation. Using our main TeDeSC system, we grouped these images based on the criterion `Occupation` using three bias mitigation strategies: *i) No mitigation:* using general descriptions from the MLLM for LLM grouping; *ii) Our default hard positive prompting strategy:* using criterion-specific captions from the MLLM for LLM grouping; and *iii) Our default strategy with additional*

*negative prompt:* adding a simple negative prompt, "Do not consider gender," to both the MLLM captioning and LLM grouping prompts.

In this experiment, non-biased result is defined as achieving equal gender proportions within each cluster. Tab. 34 presents the average gender ratios of the clustering results for each method across the four occupations. As observed, without bias mitigation, TeDeSC exhibits noticeable gender bias in the studied occupations, with a gender disparity of 19.4%. However, our default bias mitigation techniques effectively reduce this disparity to 4.9%, achieving performance comparable to the addition of a manual negative prompt. This experiment demonstrates the effectiveness of our bias mitigation strategy and highlights the potential for further reducing model bias in our framework using more advanced techniques.

Table 34: **Average gender ratio and disparity** across the four studied occupations (Craftsman, Laborer, Dancer, and Gardener) from the FACET dataset. Images sampled from each occupation have an equal proportion of genders. Results from different bias mitigation strategies are reported.

| Bias Mitigation Strategy | Male (%) | Female (%) | Gender Disparity (%) |
|---|---|---|---|
| Ground-truth | 50.0 | 50.0 | 0.0 |
| No mitigation | 40.3 | 59.7 | 19.4 |
| Ours (default) | 47.6 | 52.5 | 4.9 |
| Ours w. Negative prompt | 48.4 | 51.6 | 3.2 |

## O  FURTHER STUDY ON MULTI-GRANULARITY CLUSTERING

In this section, we provide a detailed study on how different levels of multi-granularity output from our TeDeSC framework impact grouping results. Specifically, for the Action-3c dataset, we employed human annotators to label two additional granularity levels for the criteria `Action` and `Location`. For the `Action` criterion, we consider the original annotation as fine-grained (L3) and tasked annotators to name the action in the image using more abstract and general coarse-grained (L1) and middle-grained (L2) labels. For the `Location` criterion, we consider the original annotation as middle-grained (L2) and tasked annotators to provide both more abstract coarse-grained (L1) labels and more specific fine-grained (L3) labels. This process resulted in expanded ground-truth annotations at three distinct semantic granularity levels for both the `Action` and `Location` criteria of the Action-3c dataset.

Next, we quantitatively evaluated the multi-granularity grouping results at each predicted clustering granularity level against each ground-truth annotation granularity level by measuring clustering accuracy (CAcc) and semantic accuracy (SAcc). The main caption-based TeDeSC framework was used for this experiment. In Fig. 24, we report the Harmonic Mean of CAcc and SAcc for the `Action` and `Location` criteria of Action-3c, across each predicted clustering granularity level evaluated against each ground-truth annotation level. As clearly shown, the highest grouping performance consistently appears along the diagonal. This indicates that the best grouping performance is achieved when the predicted granularity *matches* the annotation granularity.

These experimental results not only highlight the importance of the multi-granularity output of our framework but also validate the effectiveness of our multi-granularity design in aligning with user-preferred granularities that is reflected by the annotations in these experiments.

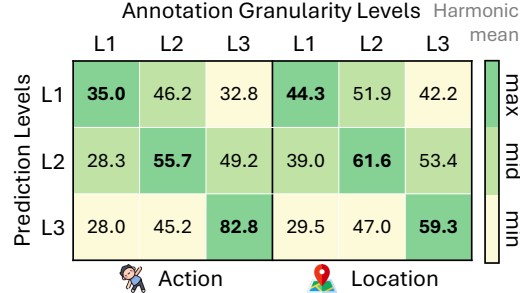

Figure 24: **Further study on the influence of multi-granularity clustering output.** We evaluate the CAcc and SAcc of the multi-granularity grouping results at each predicted clustering granularity level against each ground-truth annotation granularity level for the `Action` and `Location` criteria of the Action-3c dataset. The Harmonic Mean of CAcc and SAcc is reported for each granularity pair. L1, L2, and L3 represent the coarse-grained, middle-grained, and fine-grained levels, respectively, for both predictions and annotations.

## P  WHY LLMs IMPROVE IMAGE CLUSTERING?

The most compelling aspect of this work lies in our TeDeSC framework's ability to transform large volumes of unstructured images into natural language and leverage the advanced text understanding and summarization capabilities of LLMs to tackle the challenging Semantic Multiple Clustering (SMC) task. This approach draws inspiration from the use of LLMs in the Topic Discovery task within the NLP domain (Eklund & Forsman, 2022). Our core motivation is: "If LLMs can discover topics from documents and organize them, then by converting images into text, we can similarly use LLMs to organize unstructured images."

Traditional clustering methods (Estivill-Castro, 2002; Caron et al., 2018; Van Gansbeke et al., 2020; Li et al., 2023b; Yu et al., 2024) often depend on pre-defined criteria, pre-determined numbers of clusters, fixed feature representations (which require training), and are typically not interpretable. These limitations hinder their applicability to diverse datasets in open-world scenarios, as they demand significant human priors and retraining for each new dataset.

In contrast, LLMs (OpenAI, 2022a; 2023; Touvron et al., 2023; Meta, 2024a;b) excel at understanding, summarizing, and reasoning over high-level semantics expressed in natural language across diverse domains (*e.g.*,, everyday content, cultural knowledge, or medical content). Operating in a zero-shot (Kojima et al., 2022), interpretable manner, LLMs are uniquely suited to the SMC task, which aims to discover meaningful and interpretable clustering criteria without requiring prior knowledge or training. By integrating LLMs with MLLMs (Liu et al., 2024b) into the carefully designed TeDeSC framework, we enable the discovery and refinement of clustering criteria directly from the dataset's content, followed by automatic grouping of the dataset. This design allows our framework to overcome the rigid assumptions of traditional clustering methods, making it automatic, generalizable, and training-free. Our approach provides a novel perspective, demonstrating how clustering tasks can evolve beyond traditional paradigms.

**Challenges of employing LLMs to facilitate the SMC task.** The main challenge of employing LLMs for the SMC task lies in accurately translating visual content from images into natural language that LLMs can effectively reason with. This is evident from the sensitivity analysis results in App. F.1: TeDeSC's performance improves with larger or more powerful MLLMs (see Fig. 11 (a)), while it remains relatively insensitive to the specific choice of LLM (see Fig. 11 (b)). In other words, the quality of image captions generated by MLLMs is critical for the effective use of LLMs in the SMC task. Specifically, in the first stage of TeDeSC (criteria proposal), captions need to be as comprehensive as possible to provide *rich* information for LLMs to discover grouping criteria. In the second stage (semantic grouping), criterion-specific captions should precisely capture relevant visual content to provide *accurate* information for assigning images to clusters.

To enhance caption quality, techniques such as MLLM model ensembling, prompt ensembling (Liu et al., 2024c), or stronger models like GPT-4V (OpenAI, 2023) can improve comprehensiveness. For better precision, advanced prompting methods like CoT (Wei et al., 2022) or FineR (Liu et al., 2024e) can capture nuanced details, while hallucination mitigation tools like Visual Fact Checker (Ge et al., 2024) can reduce noise caused by hallucinations. However, these techniques increase computational costs and framework complexity. In this work, we choose to keep TeDeSC simple yet effective, and we outline these potential improvements for future practitioners.

## Q  FURTHER EVALUATION DETAILS

**Further Discussion on Clustering Accuracy (CAcc).** Clustering Accuracy (CAcc) (Han et al., 2021) is evaluated by applying the Hungarian algorithm (Kuhn, 1955) to determine the optimal assignment between the predicted cluster indices and ground-truth labels. As extensively discussed in the GCD (Vaze et al., 2022) literature, if the number of predicted clusters (groups) exceeds the total number of ground-truth classes (groups), the extra clusters (not matched by the Hungarian algorithm) are assigned to a null set, and all instances in those clusters are considered incorrect during evaluation. On the other hand, if the number of predicted clusters is lower than the number of ground-truth classes, the extra classes are assigned to a null set, and all instances with those ground-truth labels are similarly considered incorrect. Thus, CAcc is maximized only when the number of predicted clusters matches the number of ground-truth clusters.

In the Semantic Multiple Clustering (SMC) task newly proposed in this work, we do not assume access to the ground-truth number of clusters as prior input. Consequently, our proposed method TeDeSC does not rely on the ground-truth number of clusters to achieve an "optimal" CAcc with respect to the testing dataset. All clusters are automatically predicted by the TeDeSC system. In stark contrast, in the comparison with criterion-conditioned clustering methods shown in Tab. 2, both IC|TC (Kwon et al., 2024) and MMaP (Yao et al., 2024) use the ground-truth number of clusters as prior input.

