# OpenReview forum: "Organizing Unstructured Image Collections using Natural Language"
_ICLR.cc/2025/Conference — Submitted to ICLR 2025_

### Official Review · Reviewer_rRnH · 2024-10-30

**Soundness:** 4
**Presentation:** 3
**Contribution:** 4
**Rating:** 6
**Confidence:** 2

**Summary:**

This paper introduces the semantic multiple clustering task and proposes a two-stage framework named TeDeSC, which consists of Criteria Proposer and Semantic Grouper. This framework utilizes text as a proxy to automatically discovers grouping criteria in natural language from large image collections and uncovers interpretable data substructures. Extensive experiments conducted on 6 benchmarks and various real-world applications demonstrate its utility to reveal valuable insights.

**Strengths:**

1.	This work introduces a new task, i.e., Semantic Multiple Clustering (SMC), which is interesting and important for real-world applications.

2.	The framework can automatically discover both the criteria and the corresponding clusters from a image set from various granularities, which reduces the cost of manual annotation.

3.	The experimental part is quite comprehensive and presented with a rich and diverse manner.

**Weaknesses:**

1.	The framework proposed by the authors relies heavily on a variety of pre-trained LLMs and MLLMs, it is important to discuss the costs of time and resource required.

2.	The “ill-posed nature” in the Introduction needs further explanations. In addition, there is a lack of clear explanation of the meaning of the values in Figure 4 (b).

**Questions:**

Please refer to the Weakness.

In sum, I think this paper is an interesting paper. I would like to raise my score after further discussions.

---

> ### Author Response · Authors · 2024-11-20
> **Response to Reviewer rRnH (1/4)**
>
> Thank you for your positive and attentive feedback. We are happy that you found our paper interesting. Hope you enjoyed reading it. We are greatly encouraged that you found our proposed new task interesting, recognized the usefulness of our automated approach, and appreciated its significant value for real-world applications. This is exactly what we aimed to demonstrate through the three application experiments. Also, we are glad that you acknowledged our experiments as comprehensive and presented in a rich and diverse manner.  We address your questions point by point below.

---

> ### Author Response · Authors · 2024-11-20
> **Response to Reviewer rRnH (2/4)**
>
> > **Q1:** “The framework proposed by the authors relies heavily on a variety of pre-trained LLMs and MLLMs, it is important to discuss the costs of time and resource required.”
>
> > **A1:** Great suggestion! Following your suggestion, we have now conducted a detailed analysis of the computational efficiency of our main method (Caption-based Proposer + Caption-based Grouper) on the COCO-4c benchmark (5,000 images with 4 criteria) on various machines, reporting the average and total time costs in the `Table below`. The time costs were calculated for organizing all 5,000 images according to all the 4 criteria. For these experiments, we used standard LLaVA-NeXT-7B as the MLLM and Llama-3.1-8B as the LLM in the caption-based system. **We have included this analysis in Appendix K of the revised paper.**
> >
> > Specifically, our main framework (caption-based) requires **up to 31 GB GPU memory**. All experiments reported in the paper were conducted on 4 Nvidia A100 40GB GPUs. As shown in the `Table below`, organizing 5,000 images based on all the 4 discovered criteria can be completed in 29.1 hours on a single A100 machine or 16.7 hours on a single H100 machine. More importantly, most steps in our system, such as per-image captioning and per-caption cluster assignment, can be parallelized across multiple GPUs to accelerate the process. As demonstrated in the `Table below`, with 4 A100 or H100 GPUs, we can achieve approximately a 4x speedup (7.6 hours on 4 A100 and 4.3 hours on 4 H100) in computational time. We will release our code with support for multi-GPU parallelization.
> >
> > Since the intended application scenarios for our method involve image collection analysis rather than real-time systems, we believe this computational time cost is applicable for its potential applications. In the application study of Dataset Bias Discovery on CelebA, we successfully applied our system to analyze the large-scale CelebA dataset, which contains 162,000 images. In addition, as you noted, one of the primary functions of our proposed system is to reduce manual annotation costs. To illustrate this, we included the human annotation time cost for COCO-4c as a reference in the `Table below`. As shown in the comparison, our proposed method significantly reduces human annotation time (from 360 hours to 16.7 hours on one H100), demonstrating that our method can serve as an efficient automatic tool for organizing unstructured image collections.
>
> > | Method   | Resources| Average time cost (sec/img) ↓ | Total time cost (hrs) ↓|
> > | --------                 | --------               | -------- | -------- |
> > | Human annotator†         | 5 annotators           | 259.2    | 360.0    |
> > | Caption-based TeDeSC     | 1 Nvidia A100-40GB     | 20.9     | 29.1     |
> > | Caption-based TeDeSC     | 4 Nvidia A100-40GB     | 5.5      | 7.6      |
> > | Caption-based TeDeSC     | 1 Nvidia H100-80GB     | 12.0     | 16.7     |
> > | Caption-based TeDeSC     | 4 Nvidia H100-80GB     | 3.1      | 4.3      |
> >
> > †: In our project, annotating the newly proposed benchmark required 15 working days, with 5 human annotators working 8 hours per day.

---

> ### Author Response · Authors · 2024-11-20
> **Response to Reviewer rRnH (3/4)**
>
> > **Q2:** “The “ill-posed nature” in the Introduction needs further explanations.”
>
> > **A2:** Thank you for the suggestion to improve our writing. We have added further clarification on this point on **Page 1, Line#49-52** (in green) in the revised paper. The term “ill-posed nature” in the Introduction refers to the issue where “Given an image collection, Deep Clustering (DC) methods assume only a single clustering (grouping) structure. However, it is well-known that clusters are subjective—i.e., there may be multiple clustering structures. What forms a clustering depends on user needs.”
>
> ---
>
> > **Q3:** “In addition, there is a lack of clear explanation of the meaning of the values in Figure 4 (b).”
>
> > **A3:** The meaning and calculation of the diversity scores shown in `Figure 4 (b)` are explained in the "Evaluation Metrics for Criteria Discovery" section (**Page 5, Line#268-272**) of the main paper, as these scores serve as a metric for assessing the quality of criteria discovery. We have improved the figure caption to help the readers to interpret the figure.

---

> ### Author Response · Authors · 2024-11-20
> **Response to Reviewer rRnH (4/4)**
>
> We hope our responses have adequately addressed your questions, and that might encourage you to reconsider your evaluation of our work.  We have carefully incorporated your suggestions in the revised paper.
>
> Should you have any additional questions, please don't hesitate to let us know. We would be more than happy to engage in further discussion with you. Thank you once again for your valuable feedback, which has helped strengthen our paper.
>
> Authors of Submission #1890

---

> ### Author Response · Authors · 2024-11-23
> **Looking Forward to Your Feedback – Thank You from the Authors**
>
> Dear Reviewer,
>
> Thank you once again for your valuable feedback on our work. We deeply appreciate the opportunity to engage in a meaningful scientific discussion about this work with you.
>
> As the Author-Reviewer Discussion Phase nears its conclusion, we kindly ask if our rebuttal sufficiently addresses the concerns and questions raised in your initial review? Your insights are highly valuable to us, and we are eager to clarify any remaining points or provide additional explanations if needed.
>
> This work represents seven months of dedicated effort, and with this submission, we aspire to share its exploration, methodology, results, and applications with the community in a comprehensive way. We will also open-source the code.
>
> Your insights are  essential to us, and we wholeheartedly hope for the chance to address any remaining concerns you may have.
>
> Thank you for your time and consideration. We look forward to your reply!
>
> Sincerely,
> Authors of Submission #1890

---

> > ### Comment · Reviewer_rRnH · 2024-11-29
> >
> > Thanks for your responses. I have also read the comments from other reviewers. I prefer tp hold my score.

---

> > > ### Author Response · Authors · 2024-11-29
> > > **Response to Reviewer rRnH - Thank you!**
> > >
> > > Dear Reviewer rRnH,
> > >
> > > Thank you for maintaining your initial positive score! We sincerely appreciate your response and acknowledgment.
> > >
> > > Incorporating the computational cost analysis based on your suggestion has undoubtedly strengthened our work. This not only enhances the robustness of our study but also highlights its potential and efficiency in assisting   manual annotation.
> > >
> > > We hope our rebuttal and the revised paper now have sufficiently addressed your concerns. If so, we would be grateful if you would reconsider your score for our work on the novel, interesting Semantic Multiple Clustering task, as you recognised in your initial comments: *”In sum, I think this paper is an interesting paper. I would like to raise my score after further discussions."* If you have any further questions, we would be happy to engage in further discussion.
> > >
> > > Thank you once again for your time and consideration.
> > >
> > > Sincerely,
> > >
> > > Authors of Submission #1890

---

### Official Review · Reviewer_Npux · 2024-11-04

**Soundness:** 2
**Presentation:** 3
**Contribution:** 2
**Rating:** 5
**Confidence:** 3

**Summary:**

The paper addresses the challenge of organizing large, unstructured image collections by introducing the task of Semantic Multiple Clustering (SMC). SMC aims to automatically group images into semantically meaningful clusters based on multiple, unknown criteria without requiring prior human input or predefined categories. To tackle this problem, the authors propose TeDeSC (Text Driven Semantic Multiple Clustering). This framework leverages large language models (LLMs) and multimodal LLMs (MLLMs) to discover clustering criteria and organize images accordingly. By translating visual data into textual form, TeDeSC enables the identification of hidden patterns within image collections. The authors evaluate their method on two new benchmarks they contribute—COCO-4c and Food-4c—and demonstrate its applicability in tasks such as discovering biases in text-to-image generative models and analyzing social media image popularity.

**Strengths:**

This paper tackles the challenge of organizing unstructured image collections without prior knowledge by introducing a new task called Semantic Multiple Clustering (SMC). This significant contribution fills a clear gap in the existing literature. The authors present a TeDeSC framework, which cleverly leverages large language models (LLMs) and multimodal LLMs (MLLMs) to automatically discover clustering criteria by turning visual data into text. This approach is innovative and showcases originality in methodology.

They conduct thorough experiments and introduce two new challenging datasets, COCO-4c and Food-4c, which better reflect real-world complexity. By comparing various design choices using appropriate metrics like True Positive Rate (TPR), Semantic Accuracy (SAcc), and Clustering Accuracy (CAcc), they provide a comprehensive evaluation of their method.

The paper is well-written, with a clear problem statement and a structured presentation. The effective use of figures and tables enhances understanding. Practical applications—such as uncovering biases in text-to-image generative models and analyzing social media image popularity—underscore the work's significance across different domains.

**Weaknesses:**

1. The explanation of the multi-granularity clustering in the Semantic Grouper isn't detailed enough, making it challenging to understand and reproduce the method thoroughly. Additionally, the paper doesn't discuss the computational costs or how well TeDeSC scales with large datasets, which is essential given that LLMs and MLLMs can be quite resource-intensive.

2. The paper doesn't sufficiently explore the potential limitations of TeDeSC, such as the risk of introducing biases from LLMs and MLLMs into the clustering results. It would benefit from a deeper discussion on these biases, how to mitigate them, and the ethical implications—especially in sensitive areas like bias detection. Also, there's no discussion on how TeDeSC might be applied to other unstructured data types beyond images.

3. Using True Positive Rate (TPR) to evaluate criteria discovery without considering false positives might not give the complete picture. Including additional metrics or discussing how false positives could impact the results would provide a more comprehensive assessment. Moreover, the comparative analysis could be strengthened by adding more baseline methods or conducting ablation studies to isolate the contributions of each component within TeDeSC.

4. The user study on bias discovery in text-to-image generative models doesn't offer enough methodological detail. Providing more information about how the study was designed, who the participants were, what questions were asked, and the statistical methods used would enhance the credibility of the findings and help readers better assess the conclusions.

**Questions:**

First, I find the concept of multi-granularity clustering in your Semantic Grouper quite interesting. Still, I'm a bit unclear on how you determine the different granularity levels (coarse, middle, fine). Could you provide more details on this process? An illustrative example showing how these levels impact the clustering results would be beneficial.

Second, given that TeDeSC heavily relies on large language models (LLMs) and multimodal LLMs (MLLMs), I'm concerned about potential biases in these models affecting your clustering outcomes—especially in sensitive applications like bias detection. How have you addressed or mitigated these biases within your framework to ensure fair and unbiased results?

Third, in your evaluation of criteria discovery, you use True Positive Rate (TPR) but don't seem to account for false positives. Could you explain the reasoning behind this choice? How might false positives impact your system's performance, and do you think incorporating metrics that consider false positives would provide a more comprehensive assessment?

Finally, in the application section on bias discovery in text-to-image generative models, you mention a user study but provide limited details. Could you elaborate on the study design? Information such as the number of participants, their demographics, the questions they were asked, and the statistical methods used for analysis would enhance the credibility of your findings and help readers better understand the significance of your results.

---

> ### Author Response · Authors · 2024-11-20
> **Response to Reviewer Npux (1/7)**
>
> Thank you for your positive and insightful feedback. We are glad that you consider our proposed Semantic Multiple Clustering (SMC) task as a **significant contribution** to the field and appreciate its role in **filling a clear gap** in the existing literature. We are also pleased that you found our method **innovative and methodologically original**. Additionally, we greatly value your acknowledgment of how our presentation—through figures and tables—enhanced understanding, and how our **application studies demonstrated the significance of our work across various domains**-This is exactly what we most want readers to see for SMC task. Below, we carefully address your questions and concerns point by point.

---

> ### Author Response · Authors · 2024-11-20
> **Response to Reviewer Npux (2/7)**
>
> > **Q1.1:** “How do you determine the different granularity levels (coarse, middle, fine)? Could you provide more details on this process?”
>
> > **A1.1:** Yes, we provided the implementation details of our multi-granularity clustering design in **Appendix C.2** for each type of Semantic Grouper, due to space constraints in the main paper. To improve clarity, we have added additional notes in **Section 3.2** to guide readers to these details.
> >
> > To determine different granularity levels, our core idea is to leverage the *common sense* knowledge embedded in the weights of MLLMs and LLMs, allowing the models to automatically identify and generate varying granularity levels. This approach was chosen because: (1) Semantic granularity is inherently subjective and ambiguous (even WordNet hierarchies present ambiguity). Manual definitions would make our framework less flexible; and (2) we aimed to keep our framework fully automatic. Therefore, specifically, given an image dataset and a target criterion, the three proposed Semantic Groupers achieve multi-granularity clustering as follows:
> > * **Caption-based Grouper (main)** (described in Page 30, Lines#1612-1642): We first query the LLM to assign an initial criterion-specific name to each image based on its caption, resulting in a *flat set of initial names* after de-duplication. These initial names, along with the target criterion, are embedded into the prompt shown in `Table 18`. Using this prompt, the LLM generates three levels of cluster names (or say class taxonomy) at different semantic granularities: coarse, middle, and fine.
> > * **Tag-based Grouper** (described in Page 26, Lines#1565-1610): First, the LLM generates a list of criterion-specific tags (class names), which we consider as *middle-grained* tags. Then, for each middle-grained tag, the LLM is prompted to generate 3 super-categories (coarse-grained) and 10 sub-categories (fine-grained) using the prompts in `Tables 14 and 15`. The coarse-grained candidates are derived by taking the union of the generated super-categories, and the fine-grained candidates are obtained from the union of the sub-categories.
> > * **Image-based Grouper** (described in Page 25, Lines#1551-1555): We directly prompt the MLLM (using `Table 12`) to assign three class names for each image: an abstract one (coarse-grained), a common one (middle-grained), and a specific one (fine-grained).
> >
> > By relying on the foundation models' inherent knowledge and automatic processes, we ensure that our framework is both flexible and scalable for various datasets and criteria to achieve multi-granularity outputs.

---

> ### Author Response · Authors · 2024-11-20
> **Response to Reviewer Npux (3/7)**
>
> > **Q1.2:** “An illustrative example showing how these levels impact the clustering results would be beneficial.”
>
> > **A1.2:** Thank you for the suggestion. In the ablation study presented in **Figure 7**, we quantitatively analyzed how multi-granularity clustering improves system performance compared to single-granularity baselines. Additionally, we provided a visualization of clustering results across different granularities in **Figure 17**, which **illustrates the actual multi-granularity outputs**.
> >
> > Furthermore, in response to your suggestion, we provide an experiment we conducted during the early exploration and prototyping phase of this project, which further demonstrates the impact of multi-granularity clustering. Specifically, we employed human annotators to label two additional granularity levels for the `Action` criterion (coarse and middle-grained labels) and the `Location` criterion (coarse and fine-grained labels) for the `Action-3c` dataset (1,000 images). This process resulted in ground-truth annotations at three different granularity levels for the `Action` and `Location` criteria.
> >
> > We then quantitatively measured the grouping results of **each predicted clustering granularity level against each ground-truth annotation granularity level**, by comparing the clustering accuracy (CAcc) and semantic accuracy (SAcc). We report the Harmonic mean of the CAcc and SAcc in the `Tables below`. As we can clearly observe,** the highest values always appear along the diagonal**. **This represents that the best grouping performance is always achieved when the predicted granularity *matches* the annotation granularity**. These experimental results not only highlights the importance of multi-granularity output of our framework, but also validate the effectiveness of our multi-granularity design in aligning with user-preferred granularities (that is the annotations in experiments). We have now included this additional study in **Appendix O** of the revised paper.
>
> > **`Tables below`: Analysis of the multi-granularity output of the Caption-based Grouper across different annotated granularities on the Action-3c dataset.** The Harmonic Mean of CAcc and SAcc is reported in the table below, with results provided for the criteria Action and Location.
>
> > | Criterion=`Action` | GT L1 (coarse) | GT L2 (middle) | GT L3 (fine. original annotation) |
> > | ------------------ | -------------- | -------------- | -------------- |
> > | Pred L1 (coarse)   | **35.0**       | 46.2           | 32.8           |
> > | Pred L2 (middle)   | 28.3           | **55.7**       | 49.2           |
> > | Pred L3 (fine)     | 28.0           | 45.2           | **82.8**       |
>
> > | Criterion=`Location` | GT L1 (coarse) | GT L2 (middle. original annotation) | GT L3 (fine) |
> > | ------------------ | -------------- | -------------- | -------------- |
> > | Pred L1 (coarse)   | **44.3**       | 51.9           | 42.2           |
> > | Pred L2 (middle)   | 39.0           | **61.6**       | 53.4           |
> > | Pred L3 (fine)     | 29.5           | 47.0           | **59.3**       |

---

> ### Author Response · Authors · 2024-11-20
> **Response to Reviewer Npux (4/7)**
>
> > **Q2.1:** “How have you addressed or mitigated these biases within your framework to ensure fair and unbiased results?”
>
> > **A2.1:** We agree that potential biases from MLLMs and LLMs might lead to unfair clustering results, as foundation models are known to inherit biases from their training data [1]. In practice, potential risks of bias in our TeDeSC framework can be mitigated by integrating techniques such as 1) Model ensembling, 2) Majority voting, or 3) Negative prompting at the MLLM and LLM querying stages. However, to keep our method simple and automatic, we chose Hard Positive Prompting to address these biases directly in our work:
> > * **MLLM bias mitigation**: We further prompt (condition) the MLLM to generate criterion-specific captions that **describe only the criterion-related content in each image**. This constraints the MLLM from generating non-relevant content influenced by inherent biases.
> > * **LLM bias mitigation**: Similarly, when prompting the LLM to assign image captions to clusters, we condition it to focus solely on the `Criterion` depicted in each image (see `Table 17`).
> >
> > This bias mitigation strategy allowed TeDeSC to achieve a high classification accuracy of **87.4%** for `Hair color` and **99.1%** for `Gender` on the 162,770 images in the CelebA dataset for dataset bias detection, as discussed in Appendix I.1, demonstrating TeDeSC's robustness in sensitive applications.
>
> ---
>
> > **Q2.2:** “It would benefit from a deeper discussion on these biases, how to mitigate them, and the ethical implications—especially in sensitive areas like bias detection.”
>
> > **A2.2:** Nice suggestion! Following your suggestion, we have conducted a fair clustering experiment to further evaluate our bias mitigation strategy. We sampled images for 4 occupations (Craftsman, Laborer, Dancer, and Gardener) from the FACET [2] dataset, which contains images of 52 occupations. For each occupation, we sampled 10 images of men and 10 of women, totaling 80 images, ensuring a ground-truth gender proportion disparity of 0% per occupation. Using TeDeSC, we then grouped these images based on the criterion `Occupation` with three bias mitigation strategies: (1) No mitigation: using general captions and LLM prompts for grouping; (2) Our default mitigation strategy: using criterion-specific captions and LLM prompts; and (3) Our default mitigation strategy with a Negative Prompt: adding a simple negative prompt, `Do not consider gender`, in both the MLLM captioning and LLM grouping prompts. In this experiment, we consider not-biased results to be achieved when the gender proportions within each cluster are equal.
> >
> > The `Table below` presents the average gender ratio of clustering results for each method across the four occupations. As observed, without bias mitigation, TeDeSC exhibits a certain gender bias toward occupations. However, this bias is effectively mitigated with our default bias mitigation techniques, showing comparable performance to adding a manual negative prompt. **Thank you for the suggestion! We have included this detailed discussion in the Limitation section in `Appendix N` of the revised paper.**
>
>
> > |   | Methods  | Male Proportion    | Female Propotion |Gender Disparity |
> > | -------- | -------- | -------- | -------- | -------- |
> > |     | Ground-truth                            | 50%       | 50%       | 0%    |
> > | (1) | TeDeSC w.o. Our bias mitigation strategy    | 40.3%     | 59.7%     | 19.4% |
> > | (2) | Ours w. Our bias mitigation strategy   | 47.6%     | 52.5%     | 4.9%  |
> > | (3) | Ours w. Our bias mitigation strategy + explicit negative prompt  | 48.4%     | 51.6%     | 3.2%  |
>
>
> [1] Bommasani, R., Hudson, D. A., Adeli, E., Altman, R., Arora, S., von Arx, S., ... & Liang, P. On the opportunities and risks of foundation models. arXiv, 2021.
>
> [2] Gustafson, L., Rolland, C., Ravi, N., Duval, Q., Adcock, A., Fu, C. Y., ... & Ross, C. Facet: Fairness in computer vision evaluation benchmark. In ICCV, 2023.

---

> ### Author Response · Authors · 2024-11-20
> **Response to Reviewer Npux (5/7)**
>
> > **Q3.1:** Justification for using True Positive Rate (TPR) as the metric for Criteria Proposal evaluation.
>
> > **A3.1:** As emphasized on **Page 5, Line#273-274**, the number of valid grouping criteria (True Positives) in an unstructured image collection is inherently unlimited because our Semantic Multiple Clustering (SMC) task is open-ended. This is particularly true for complex real-world image datasets, such as COCO-4c, where the diversity of scenes and objects introduces countless possible grouping criteria. Given a large image dataset, we cannot assume that human-annotated criteria cover all potential valid groupings. Even in a simple case, the range of criteria can be vast:
> >
> > ```text
> > Consider a set of 4 images captured in different gyms. How many valid grouping criteria might exist for these images? Potential criteria could include: Interior design style, Lighting color, Floor material, Density of sports equipment, Brand of equipment, Number of windows, Number of people, Age composition, Activity type, or even Number of people using phones, and so forth. The possibilities are nearly endless, even for this toy example.
> > ```
> >
> > Given this, the discovered criteria that are not included in the annotated set cannot simply be considered False Positives (FPs), as defining what constitutes an FP (or "invalid grouping criterion") is subjective and extremely challenging—even for humans. Therefore, we employ True Positive Rate (TPR) as a metric for Criteria Proposal evaluation in this open-ended task. Additionally, as noted on **Page 6, Line#313-318**, we made a substantial effort to expand the annotated criteria with human annotators, creating a **Hard ground-truth set** to make TPR a more strict metric.
>
> ---
>
> > **Q3.2:** Influence of False Positive discovered criteria on system performance.
>
> > **A3.2:** Following your suggestion, we designed and conducted an interesting control experiment on the object-centric dataset Fruit-2c, where we could clearly define False Positive grouping criteria. Specifically, we artificially introduced two invalid (FP) grouping criteria, `Action` and `Clothing Style`, into the discovered criteria. We then applied our Caption-based Grouper to group images based on these invalid criteria, with the grouping statistics reported in the `Table below`.
> >
> > The results show that, when processing invalid grouping criteria, nearly all images are grouped into a cluster named `Not visible`. This occurs because, in the absence of relevant visual content in the images, the MLLM-generated captions do not contain related descriptors. Consequently, the LLM creates a `Not visible` cluster and assigns the images to it. This control experiment highlights the robustness of our system against false positives. Since the system outputs interpretable results, users can easily identify and disregard invalid groupings.
> >
> > Thank you for your suggestion. We have included this experiment and discussion in Appendix M.
>
> > | Assigned Clusters  | Action   | Clothing Style |
> > | --------           | -------- | --------       |
> > | `Not visible`      | 98.3%    | 96.7%          |
> > | `Others`†          | 1.7%     | 3.3%           |
> > †: For simplicity, all other minority clusters are grouped as `Others`.

---

> ### Author Response · Authors · 2024-11-20
> **Response to Reviewer Npux (6/7)**
>
> > **Q4:** Additional discussion on computational costs and how well TeDeSC scales with large datasets.
>
> > **A4:** The proposed TeDeSC framework is training-free; thus, it involves only inference process. Specifically, our main framework (caption-based) requires **up to 31 GB GPU memory** to run. All experiments reported in the paper were conducted on 4 Nvidia A100 40GB GPUs. Following your suggestion, we have now conducted a detailed analysis of the computational efficiency of our main method (Caption-based Proposer + Caption-based Grouper) on the COCO-4c benchmark (5,000 images with 4 criteria) on various machines, reporting the average and total time costs in the `Table below`. The time costs were calculated for organizing all 5,000 images according to all the 4 criteria. For these experiments, we used standard LLaVA-NeXT-7B as the MLLM and Llama-3.1-8B as the LLM in the caption-based system. **We have included this analysis in Appendix K of the revised paper.**
> >
> > As shown in the `Table below`, organizing 5,000 images based on all the 4 discovered criteria can be completed in 29.1 hours on a single A100 machine or 16.7 hours on a single H100 machine. More importantly, most steps in our system, such as per-image captioning and per-caption cluster assignment, can be parallelized across multiple GPUs to accelerate the process. As demonstrated in the `Table below`, with 4 A100 or H100 GPUs, we can achieve approximately a 4x speedup (7.6 hours on 4 A100 and 4.3 hours on 4 H100) in computational time. We will release our code with support for multi-GPU parallelization.
> >
> > Regarding the scalability of TeDeSC with large datasets, in our application study on Dataset Bias Discovery with CelebA, we successfully applied (scaled) our system to analyze the large-scale CelebA dataset, which contains **162,000 images**. Given that the intended application scenarios for our method involve image collection analysis rather than real-time systems, we believe this computational time cost is appropriate for its intended applications.
>
> > | Method   | Resources| Average time cost (sec/img) ↓ | Total time cost (hrs) ↓|
> > | --------                 | --------               | -------- | -------- |
> > | Human annotator†         | 5 annotators           | 259.2    | 360.0    |
> > | Caption-based TeDeSC     | 1 Nvidia A100-40GB     | 20.9     | 29.1     |
> > | Caption-based TeDeSC     | 4 Nvidia A100-40GB     | 5.5      | 7.6      |
> > | Caption-based TeDeSC     | 1 Nvidia H100-80GB     | 12.0     | 16.7     |
> > | Caption-based TeDeSC     | 4 Nvidia H100-80GB     | 3.1      | 4.3      |
> >
> > †: In our project, annotating the newly proposed benchmark required 15 working days, with 5 human annotators working 8 hours per day.
>
> ---
>
> > **Q5:** Additional details on the user study design for bias discovery in text-to-image generative models.
>
> > **A5:** Indeed, due to space constraints in the main paper, as noted on **Page 10, Line#487**, detailed information about the user study is provided in **Appendix I.2 (Page 47, Lines#2523-2537)**, which includes the study's methodology, participant count, and example questions (see `Figure 23`). Additionally, `Figure 19` presents comprehensive comparative results between human evaluation scores and TeDeSC-estimated scores for each model, occupation, and criterion. In our study, responses were collected from 54 anonymous participants, with 6 evaluations conducted for each occupation and criterion. The study was approved by our university's Ethical Review Board. To ensure and respect participant privacy, the interviews were fully anonymous, and no demographic information (e.g., identity, gender, race, age) was collected.

---

> ### Author Response · Authors · 2024-11-20
> **Response to Reviewer Npux (7/7)**
>
> > **Q6:** “Moreover, the comparative analysis could be strengthened by adding more baseline methods or conducting ablation studies to isolate the contributions of each component within TeDeSC.”
>
> > **A6:** Thank you for your suggestions. Since the Semantic Multiple Clustering (SMC) task proposed in our work is new, to the best of our knowledge, no existing methods directly address it. Therefore, we introduced a general two-stage framework, TeDeSC, and explored three types of Criteria Proposers and three types of Semantic Groupers by varying components within the framework to investigate possible approaches to the SMC task. As these three variants follow the same framework, the comparisons between them serve as our ablation studies.
> >
> > Additionally, in evaluating the semantic groupers, we compared TeDeSC with four baseline methods that use strong representations (CLIP zero-shot, and k-means with CLIP, DINOv1, DINOv2) for CAcc, as well as with two recent text-conditioned clustering methods (IC|TC and MMaP) for both CAcc and SAcc. We believe these comparisons offer sufficient insights and experience for future researchers aiming to develop SMC systems. If you have additional baseline suggestions, please let us know—we would be more than happy to compare them and include the results in our paper.
>
> ---
>
> > **Q7:** “How TeDeSC might be applied to other unstructured data types beyond images.”
>
> > **A7:** Great suggestion! The core idea of TeDeSC is to *use text as a proxy (or medium)* for reasoning over unstructured data, mass-producing human-interpretable insights about the data. As such, TeDeSC can be directly applied to textual data (e.g., documents). Furthermore, since natural language is a versatile and one of the most common medium of representation, TeDeSC could be extended to other data types by converting these data into text (by replacing the captioning module with alternative tools), such as:
> > * **Audio Data**: Using Speech-to-Text models like Whisper [1], audio data can be converted to text for subsequent analysis with TeDeSC.
> > * **Tabular Data**: Table-to-Text models, such as TabT5 [2], can translate tables into text, allowing TeDeSC to be applied. For tables that include figures, modern MLLMs like LLaVA-Next, which support both OCR and image-to-text capabilities, could handle these elements to create a unified text representation for TeDeSC.
> > * **Protein Structure**: Protein structure to text models, such as ProtChatGPT [3], can translate protein sequence to text descriptions for subsequent analysis with TeDeSC.
> > * **Point Cloud Data**: 3D captioning models, like Cap3D [4], can transform point cloud data (or rendered 3D models) into text, allowing TeDeSC to be applied.
> >
> > **We have included this discussion in the Future Work section in `Appendix L` of the revised paper**, providing directions for future practitioners to explore these possibilities further.
>
> [1] Radford, A., Kim, J. W., Xu, T., Brockman, G., McLeavey, C., & Sutskever, I. Robust speech recognition via large-scale weak supervision. In *ICML*, 2023.
>
> [2] Andrejczuk, E., Eisenschlos, J. M., Piccinno, F., Krichene, S., & Altun, Y. Table-to-text generation and pre-training with tabt5. In *Findings of EMNLP*, 2022.
>
> [3] Wang, C., Fan, H., Quan, R., & Yang, Y. (2024). Protchatgpt: Towards understanding proteins with large language models. *arXiv*, 2024.
>
> [4] Luo, T., Rockwell, C., Lee, H., & Johnson, J. Scalable 3d captioning with pretrained models. In *NeurIPS*, 2024.
>
> ---
>
> We hope our responses have thoroughly addressed your questions and clarified any uncertainties, and that encourage you to reconsider your evaluation of our work. We have carefully incorporated your suggestions in the revised paper.
>
> If you have any further questions or require additional clarification, please feel free to reach out. We would be delighted to engage in further discussion. Thank you once again for your insightful feedback, which has significantly contributed to improving our paper.
>
> Authors of Submission #1890

---

> ### Author Response · Authors · 2024-11-23
> **Looking Forward to Your Feedback – Thank You from the Authors**
>
> Dear Reviewer Npux,
>
> Thank you once again for your valuable feedback on our work. We deeply appreciate the opportunity to engage in a meaningful scientific discussion about this work with you.
>
> As the Author-Reviewer Discussion Phase nears its conclusion, we kindly ask if our rebuttal sufficiently addresses the concerns and questions raised in your initial review? Your insights are highly valuable to us, and we are eager to clarify any remaining points or provide additional explanations if needed.
>
> This work represents seven months of dedicated effort, and with this submission, we aspire to share its exploration, methodology, results, and applications with the community in a comprehensive way. We will also open-source the code.
>
> Your insights are  essential to us, and we wholeheartedly hope for the chance to address any remaining concerns you may have.
>
> Thank you for your time and consideration. We look forward to your reply!
>
> Sincerely,
> Authors of Submission #1890

---

### Official Review · Reviewer_VuXq · 2024-11-04

**Soundness:** 3
**Presentation:** 2
**Contribution:** 3
**Rating:** 8
**Confidence:** 4

**Summary:**

The paper presents semantic multiple clustering of images based on text generated through LLMs or MLLMs through prompting. The proposed method is unsupervised and produces multiple clusterings based on the criteria given in natural language without specifying number of clusters. The method automatically discovers multiple criteria for clustering and produces clustering at multiple granularities.  The paper also introduces two new benchmarks COCO-4c and Food-4c sourced from Food-101 and COCO-val with each having four clustering criteria. TeDeSC is applied to a variety of applications, including discovering biases in real-world datasets and text-to-image generative models, and analyzing the popularity of social media images.

**Strengths:**

The proposed method implements many approaches to discover the clustering criteria and compared the results.  For each proposer approach, a grouper approach is implemented which combines the similar criteria and assigns a name to it.
The results are analysed against the hard ground truth (annotated by human oracle)
The proposed method is applied on different applications achieved good results.

**Weaknesses:**

The entire method is based on use of LLMs or MLLMs through prompting. What is the effect of hallucination and biases present in these models?

The comparison with existing models is limited. Only two methods IC/TC and MMaP are considered.

In image-based proposer, an image grid (8x8) is constructed and passed to a MLLM for criteria generation. What is the significance of such a grid? The grid would have different type of images and will confuse the MLLM.

Criteria pool refinement process takes the model generated text and work on it. It may lead to hallucination. How would you justify it?

Typos
Page 1 last line …remove the
Table 28 – Highlighted values for Action-3c … both CACC

**Questions:**

Same as in weaknesses

---

> ### Author Response · Authors · 2024-11-20
> **Response to Reviewer VuXq (1/5)**
>
> Thank you for your constructive feedback. We are glad that you recognize the effectiveness of our proposed framework and its variants for the newly introduced Semantic Multiple Clustering task, as well as its good performance across various applications. Below, we address your questions and concerns point by point.

---

> ### Author Response · Authors · 2024-11-20
> **Response to Reviewer VuXq (2/5)**
>
> > **Q1:** “The entire method is based on use of LLMs or MLLMs through prompting. What is the effect of hallucination and biases present in these models?”
>
> > **A1:** Thanks for raising a valid issue of hallucination and biases of LLMs and MLLMs that impacts any system built around LLMs and MLLMs, which is an open research topic. Next, we address your concern about hallucination and biases respectively, and explain how our proposed system places necessary guardrails against hallucination and biases. Moreover, based on your suggestions and the subsequent discussions, we have included a discussion on foundation model hallucination and biases in the Limitation section (`Appendix N`) of the revised paper.
>
> > **A1.1 About hallucination:** First, as discussed in [1], LLM hallucination often occurs when it is tasked with complex tasks that requires world knowledge and information, i.e., for the question of “Who was the 70th president of the United States?”, the LLM might generates a fabricated name. However, the use of LLM in our system is fully grounded to visual descriptions (tags or captions) of the images. Therefore, the LLM output is strongly constrained to analyze the visual descriptions. As such, the hallucination from LLMs in our system would be mild due to the strong grounding. Nevertheless, LLM hallucination indeed has certain effects on the clustering results. As we discussed in our Failure Case Analysis (see Appendix H, `Figure 18`), the LLM incorrectly grouped “Korean bibimbap and “Vietnamese rice noodles” to "Chinese cuisine" (actually these food exist in both Chinese, Korean and Vietnamese cuisines).
> >
> > Second, MLLMs play a more important role in our system because it translates images to text for the later process. MLLM hallucination often occurs on incorrectly claiming the existence of certain objects, attributes, or spatial relationship of objects in the image. However, our proposed system runs on the **whole dataset level rather than per-image. This makes it insensitive to these common hallucination at fine-grained visual detail level**.
> >
> > Moreover, since our system is training-free, it can be further enhanced with LLM or MLLM hallucination mitigation techniques such as VisualFactChecker [2], which we leave for future work.
>
> [1] Wang, Y., Wang, M., Manzoor, M. A., Liu, F., Georgiev, G., Das, R. J., & Nakov, P. Factuality of Large Language Models: A Survey. In *EMNLP*, 2024.
>
> [2] Ge, Y., Zeng, X., Huffman, J. S., Lin, T. Y., Liu, M. Y., & Cui, Y. Visual Fact Checker: Enabling High-Fidelity Detailed Caption Generation. In *CVPR*, 2024.

---

> ### Author Response · Authors · 2024-11-20
> **Response to Reviewer VuXq (3/5)**
>
> > **A1.2 About bias:** Foundation models are known to inherit biases from their training data [3]. In our system, we employed Hard Positive Prompting techniques to address potential biases:
> > * **MLLM bias mitigation**: We further prompt (condition) the MLLM to generate criterion-specific captions that **describe only the criterion-related content in each image**. This constraints the MLLM from generating non-relevant content influenced by inherent biases.
> > * **LLM bias mitigation**: Similarly, when prompting the LLM to assign image captions to clusters, we condition it to focus solely on the `Criterion` depicted in each image (see `Table 17`).
> >
> > This bias mitigation strategy allowed our system to achieve a high classification accuracy of **87.4%** for `Hair color` and **99.1%** for `Gender` on the 162,770 images in the CelebA dataset for dataset bias detection, as discussed in Appendix I.1, demonstrating its robustness in bias-sensitive applications.
>
> > Inspired by your suggestion, we have conducted a fair clustering experiment to further evaluate our bias mitigation strategy. We sampled images for 4 occupations (Craftsman, Laborer, Dancer, and Gardener) from the FACET [4] dataset, which contains images of 52 occupations. For each occupation, we sampled 10 images of men and 10 of women, totaling 80 images, ensuring a ground-truth gender proportion disparity of 0% per occupation. Using our TeDeSC system, we then grouped these images based on the criterion `Occupation` with three bias mitigation strategies: (1) No mitigation: using general captions and LLM prompts for grouping; (2) Our default mitigation strategy: using criterion-specific captions and LLM prompts; and (3) Our default mitigation strategy with a Negative Prompt: adding a simple negative prompt, `Do not consider gender`, in both the MLLM captioning and LLM grouping prompts. In this experiment, we consider not-biased results to be achieved when the gender proportions within each cluster are equal.
> >
> > The `Table below` presents the gender ratio of the clustering results for each method averaged across the four occupations. As observed, without bias mitigation, TeDeSC exhibits a certain gender bias toward occupations. However, this bias is effectively mitigated with our default bias mitigation techniques, showing comparable performance to adding a manual negative prompt. Thank you for the suggestion! We have included these discussions in the revised paper.
>
>
> |   | Methods  | Male Proportion    | Female Proportion |Gender Disparity |
> | -------- | -------- | -------- | -------- | -------- |
> |     | Ground-truth                            | 50%       | 50%       | 0%    |
> | (1) | TeDeSC w.o. Our bias mitigation strategy    | 40.3%     | 59.7%     | 19.4% |
> | (2) | Ours w. Our bias mitigation strategy   | 47.6%     | 52.5%     | 4.9%  |
> | (3) | Ours w. Our bias mitigation strategy + explicit negative prompt  | 48.4%     | 51.6%     | 3.2%  |
>
>
>
> [3] Bommasani, R., Hudson, D. A., Adeli, E., Altman, R., Arora, S., von Arx, S., ... & Liang, P. On the opportunities and risks of foundation models. *arXiv*, 2021.
>
> [4] Gustafson, L., Rolland, C., Ravi, N., Duval, Q., Adcock, A., Fu, C. Y., ... & Ross, C. Facet: Fairness in computer vision evaluation benchmark. In ICCV, 2023.

---

> ### Author Response · Authors · 2024-11-20
> **Response to Reviewer VuXq (4/5)**
>
> > **Q2:** “Criteria pool refinement process takes the model generated text and work on it. It may lead to hallucination. How would you justify it?”
>
> > **A2:** In the Criteria Pool Refinement step, the LLM is prompted to merge semantically identical criteria (e.g., "Color scheme" and "Primary color," or "Location" and "Place") while discarding noisy or irrelevant ones (e.g., "Presence of earrings") from the initial accumulated criteria. As shown in the actual prompt provided in `Table 11`, **this refinement step is grounded in the input criteria accumulated in the pool.** In our experiments across six datasets and three application studies, we did not observe hallucinated criteria being introduced during this process.
> >
> > **However, your suggestion is insightful and inspired us to further investigate.** To this end, we designed and conducted a control experiment using the Fruit-2c dataset, where we **artificially introduced two “hallucinated” invalid grouping criteria, `Action` and `Clothing Style`, into the refined criteria to evaluate their impact on our system.** We then applied the Caption-based Grouper to group fruit images based on these “hallucinated” criteria, with the grouping statistics reported in the `Table below`.
> >
> > The results demonstrate that **when processing invalid “hallucinated” criteria, nearly all images are grouped into a cluster named `Not visible`.** This happens because, in the absence of relevant visual content in the images, the MLLM-generated captions do not include corresponding descriptors. As a result, the LLM creates a `Not visible` cluster and assigns the images to it. Since the system provides interpretable outputs, users can easily identify and disregard such invalid groupings. This control experiment underscores the robustness of our system against hallucination.
> >
> > We have included this experiment, along with the discussion, as a standalone section in **Appendix M** of the revised paper.
>
> > | Assigned Clusters  | Action   | Clothing Style |
> > | --------           | -------- | --------       |
> > | `Not visible`      | 98.3%    | 96.7%          |
> > | `Others`†          | 1.7%     | 3.3%           |
> > †: For simplicity, all other minority clusters are grouped as `Others`.
>
> ---
>
> > **Q3:** “In image-based proposer, an image grid (8x8) is constructed and passed to a MLLM for criteria generation. What is the significance of such a grid? The grid would have different type of images and will confuse the MLLM.”
>
> > **A3:** This is because the core of the Proposer design lies in its ability to **concurrently reason over the visual content of a large set of images** to **find diverse common themes among different images** and thereby proposes grouping criteria.
> >
> > To achieve this, we arrange 64 images into an 8x8 grid and pass it as a single input image to the MLLM. This approach allows the MLLM to observe a broad and diverse sample of the dataset at once, enabling it to infer clustering criteria that are generalizable across the entire dataset rather than being overly influenced by individual images. This technique is supported by prior work, such as VisDiff [5], where it was shown to be an effective and robust strategy.
> >
> > Additionally, while VisDiff [5] uses LLaVA-1.5 as the MLLM, we employed the stronger LLaVA-NeXT-Interleave model, which is specifically optimized for *multi-image reasoning*. This ensures that the MLLM can handle the diversity within the grid effectively, making our approach a stronger and more capable baseline.
>
> [5] Dunlap, L., Zhang, Y., Wang, X., Zhong, R., Darrell, T., Steinhardt, J., ... & Yeung-Levy, S. Describing differences in image sets with natural language. In *CVPR*, 2024.

---

> ### Author Response · Authors · 2024-11-20
> **Response to Reviewer VuXq (5/5)**
>
> > **Q4:** “The comparison with existing models is limited. Only two methods IC|TC and MMaP are considered.”
>
> > **A4:** The Semantic Multiple Clustering (SMC) task is a novel task that eliminates the strong and impractical assumptions-requiring prior knowledge of the number of clusters, criteria, or criterion names-of traditional Deep Clustering, Multiple Clustering, or Criterion-guided Multiple Clustering. Without such priors, to the best of our knowledge, no existing method is capable of directly accomplishing the SMC task.
> >
> > To address this, beyond our proposed main method, we made considerable efforts to construct additional baselines, including the Image-based and Tag-based methods, for comparison. Furthermore, we adapted IC|TC and MMaP to the SMC task to evaluate their second-stage clustering performance. Importantly, it should be noted that both IC|TC and MMaP used ground-truth criteria and the number of clusters as prior inputs in the comparison, whereas our proposed system operates entirely without human input. We also compared our system against four strong baseline methods (CLIP zero-shot, k-means with CLIP, DINOv1, and DINOv2) in terms of clustering accuracy (CAcc).
> >
> > We believe these comparisons provide valuable insights and guidance for future researchers interested in developing SMC systems. If you have additional baseline suggestions that are concurrent works, we would be more than happy to compare with them and include the results in our revised paper.
>
> ---
>
> > **Q5:** Typos in Page 1 last line and Table 28
>
>  > **A5:** Thank you for pointing out these typos in both the main paper and the appendix. We have corrected them in the revised the paper.
>
> ---
>
> We sincerely hope that our responses have effectively addressed your questions and resolved any concerns. We hope this might encourage you to reconsider your evaluation of our work. Your insightful suggestions have been thoughtfully incorporated into the revised paper.
>
> If there are any remaining questions or areas needing further clarification, please do not hesitate to reach out. We would be more than happy to continue the discussion. Once again, thank you for your constructive feedback, which has greatly contributed to strengthening our paper.
>
> Authors of Submission #1890

---

> ### Author Response · Authors · 2024-11-23
> **Looking Forward to Your Feedback – Thank You from the Authors**
>
> Dear Reviewer VuXq,
>
> Thank you once again for your valuable feedback on our work. We deeply appreciate the opportunity to engage in a meaningful scientific discussion about this work with you.
>
> As the Author-Reviewer Discussion Phase nears its conclusion, we kindly ask if our rebuttal sufficiently addresses the concerns and questions raised in your initial review? Your insights are highly valuable to us, and we are eager to clarify any remaining points or provide additional explanations if needed.
>
> This work represents seven months of dedicated effort, and with this submission, we aspire to share its exploration, methodology, results, and applications with the community in a comprehensive way. We will also open-source the code.
>
> Your insights are  essential to us, and we wholeheartedly hope for the chance to address any remaining concerns you may have.
>
> Thank you for your time and consideration. We look forward to your reply!
>
> Sincerely,
> Authors of Submission #1890

---

> > ### Comment · Reviewer_VuXq · 2024-11-24
> > **Response from Reviewer**
> >
> > Dear Authors of Submission #1890
> >
> > Thank you for your efforts in replying my queries. All my queries/questions are well addressed. I appreciate your work and efforts.
> >
> > best wishes
> > Reviewer VuXq

---

> ### Author Response · Authors · 2024-11-24
> **To Reviewer VuXq from the Authors: Thank You for Your Acknowledgment and Response!**
>
> Dear Reviewer VuXq,
>
> We sincerely appreciate your acknowledgment and confirmation that our responses and the revised paper have adequately addressed all your queries and concerns.
>
> Incorporating your valuable suggestions has significantly strengthened our work in the revised version of the paper. In particular, the additional experiments designed to study the effect of hallucination in the Criteria Pool Refinement process and the exploration of LLM/MLLM bias mitigation strategies, as per your suggestions, have been highly insightful and interesting. These additional studies have allowed us to present this work in a more comprehensive manner to the readers.
>
> Thank you once again for your valuable suggestions and the time you have dedicated. We truly appreciate your effort and insights.
>
> Sincerely,
> Authors of Submission #1890

---

### Official Review · Reviewer_J84w · 2024-11-05

**Soundness:** 3
**Presentation:** 4
**Contribution:** 2
**Rating:** 5
**Confidence:** 4

**Summary:**

The paper proposed a new way to organize image collections by leveraging LLM-assisted semantic multiple clustering. The proposed framework, TeDeSC, uses LLMs to convert images into texts and build a criteria proposer and a semantic grouper to discover potential semantic clusters describing a dataset. The TeDeSC provides us with a way to uncover the semantic structures and underlying biases of a dataset. Empirical studies on real-world datasets and AI-generated images validated its effectiveness.

**Strengths:**

1. The paper is well-organized and clearly written, with figures that effectively illustrate the motivation and methodology.

2. The experimental setup is thorough, with well-designed evaluation criteria and informative visual representations.

3. The proposed method is technically sound. Using LLM and multi-modal learning can improve the performance of many AI-related tasks.

**Weaknesses:**

1.  The proposed method leverages multiple LLMs to generate criteria and semantic labels, with each LLM component requiring dedicated prompt engineering. This introduces numerous design choices that may impact the system’s overall performance. To enhance the framework, it would be helpful if the authors provided quantitative metrics, such as a clearly defined loss function, to clarify the objectives behind the design choices and to better evaluate the approach's effectiveness.

2. ​​​​There is a growing trend to apply large language models (LLMs) to enhance various AI-related tasks. While this paper is innovative from an application perspective, it would benefit from a deeper discussion of its scientific contributions.

3. The overall design is primarily empirical. It could be strengthened by incorporating more theoretical analysis on the integration of its components. For example, a discussion on why using LLMs can improve image clustering, compared to previous studies, would provide valuable insights.

4. Biases in benchmark datasets have garnered considerable attention in recent years. For instance, [1] investigated geographic biases within the COCO dataset (used as a benchmark in this paper), and [2] explored spurious correlations between attributes like "Blond" and "Female" in the CelebA dataset. In response, developers of LLM products such as GPT and DALL-E have curated their training datasets and models to mitigate such spurious correlations. While the proposed method benefits from these mitigated biases by incorporating pre-trained LLMs into its pipeline, its contribution may be less impactful than suggested in Section 5, especially when compared to baseline methods trained from scratch on biased datasets.

[1] https://arxiv.org/pdf/1906.02659
[2] https://arxiv.org/pdf/2007.10075

**Questions:**

In addition to the weakness, there is one more question as below.

1. If we choose different LLMs, how they will influence the performance of the proposed framework?

---

> ### Author Response · Authors · 2024-11-20
> **Response to Reviewer J84w (1/5)**
>
> Thank you for your positive and encouraging feedback! We are delighted that you found our paper well-organized and clearly written, with figures that effectively illustrate both the motivation and methodology. We greatly appreciate your recognition of our thorough experimental setup, well-designed evaluation criteria, and informative visual representations. Additionally, we are pleased that you acknowledge the technical soundness of our proposed method. Below, we address your comments and suggestions point by point.

---

> ### Author Response · Authors · 2024-11-20
> **Response to Reviewer J84w (2/5)**
>
> > **Q1:** “If we choose different LLMs, how they will influence the performance of the proposed framework?”
>
> > **A1:** Thank you for this question! We have indeed studied this point in the paper. In `Appendix F.1`, we examined the influence of both different LLMs and MLLMs on the performance of our proposed framework, with the comparative results presented in `Figure 11`. We directed readers to this section in the main paper (Line #440). Specifically:
> > - In `Figure 11(b)`, we fixed the MLLM to LLaVA-NeXT-7B and evaluated the influence of four different LLMs: 1) GPT-4-turbo, 2) GPT-4o, 3) Llama-3-8B, and 4) Llama-3.1-8B.
> > - In `Figure 11(a)`, we fixed the LLM to Llama-3.1-8B and examined the performance of three different MLLMs: 1) GPT-4V, 2) BLIP-3-4B, and 3) LLaVA-NeXT-7B.
> >
> > Regarding the impact of LLMs, we observed that GPT-4-turbo demonstrated marginally better performance compared to our default open-source Llama-3.1-8B. However, apart from the Card-2c dataset, the system's performance remained largely consistent regardless of the choice of LLM in our study. This consistency suggests that the reasoning tasks in SMC are relatively straightforward for modern LLMs, given their capability to handle complex problems. **Our empirical results indicates that our proposed framework is fairly robust and not sensitive to the choice of LLM.**
>
> ---
>
> > **Q2:** “To enhance the framework, it would be helpful if the authors provided quantitative metrics, such as a clearly defined loss function, to clarify the objectives behind the design choices and to better evaluate the approach's effectiveness.”
>
> > **A2:** Thank you for this insightful suggestion. We agree that finding a "nearly optimal" solution in the unconstrained space of design choices is one of the core challenges in LLM prompt engineering, especially since this process is difficult to quantify with a clear loss function to guide prototyping. This becomes even more challenging in the SMC task studied in this work, as the task is open-ended and the grouping criteria are inherently subjective and application dependant.
> >
> > In this work, we developed our prompts following the **Iterative Prompt Engineering methodology** introduced by Isa Fulford and Andrew Ng. As detailed in **Appendix C**, we provided every exact LLM (and MLLM) prompt used in our framework and broke down each prompt to explain the objectives and purposes behind each design choice. These explanations include elements such as system prompts, input formatting, task and sub-task instructions, and output instructions (see `Tables 7–19`).
> >
> > More importantly, the four metrics established for the SMC task in this work—True Positive Rate (TPR), Diversity Score, Clustering Accuracy (CAcc), and Semantic Accuracy (SAcc)—are carefully designed to evaluate the impact and effectiveness of these design choices. Specifically:
> > - **Criteria Proposer Stage**: The objective of LLM prompting at this stage is to discover **comprehensive, valid**, and **diverse (non-redundant)** criteria.
> >   - **TPR**: Assesses the extent to which the predicted set of criteria covers the ground-truth set, **reflecting validity and comprehensiveness**.
> >   - **Diversity Score**: Evaluates how well the proposed criteria avoid redundancy and capture distinct aspects of the data, **ensuring diversity**.
> > - **Semantic Grouper Stage**: The objective of LLM prompting here is to uncover criterion-specific data substructures (groupings) that **align with the ground truth**.
> >   - **SAcc**: Measures how well the predicted substructures **semantically align** with the ground-truth clusters (e.g., similarity in cluster names).
> >   - **CAcc**: Measures how **structurally similar** the predicted groupings are to the ground truth (e.g., similar data distributions).
> >
> > In summary, in this work we focused on creating a general framework (including prompts) for organizing unstructured image collections and did not conduct dataset-specific prompt optimizations. The prompts are flexible and can be tweaked by users to suit the needs of the downstream business application. **We have included this discussion in a Future Work section in` Appendix L` of the revised paper.**

---

> ### Author Response · Authors · 2024-11-20
> **Response to Reviewer J84w (3/5)**
>
> > **Q3:** “While this paper is innovative from an application perspective, it would benefit from a deeper discussion of its scientific contributions.”
>
> > **A3:** Thank you for acknowledging the novelty of our application study.
> >
> > Existing models in the vision and language space have predominantly been used to reason at an image level. However, the SMC task presents unique challenges, as **it requires simulataneous reasoning over an entire image collection to discover grouping criteria**.
> >
> >  Therefore, from a methodological perspective, **our key scientific contribution** is the proposal of a **novel and general framework, TeDeSC**, which utilizes LLMs and MLLMs to **concurrently reason over large image sets (up to 162,000 CelebA images in our experiments) to address these unique challenges in the SMC task.** This differentiates our approach from other methods that use LLMs in a image-level manner. We carefully discussed these distinctions in comparison to recent LLM and MLLM tool use works in **Appendix A (Line#1164-1193)**.
> >
> > Specifically, our contributions in this paper are as follows:
> > 1. **Introduced a novel task** of Semantic Multiple Clustering (SMC) that eliminates the strong limitations of existing deep clustering and multiple clustering settings.
> > 2. **Proposed two realistic benchmarks** -- COCO-4c and Food-4c -- to further enrich the evaluation of methods. Alongside, we layed down the evaluation metrics to properly evaluate the open-ended SMC task.
> > 2. **Proposed a novel and generalizable framework, TeDeSC**, for the SMC task. In addition, we implemented and experimented with **three different approaches** to building the TeDeSC framework.
> > 3. **Explored three practical applications enabled by the SMC task and TeDeSC framework.** These applications highlight the potential of SMC in real-world scenarios and underscore its importance for future research.
> >
> > We believe these scientific contributions provide a comprehensive foundation for future research on the SMC task, making a distinct scientific impact that extends beyond simply leveraging LLMs or MLLMs.

---

> ### Author Response · Authors · 2024-11-20
> **Response to Reviewer J84w (4/5)**
>
> > **Q4:** “It could be strengthened by incorporating more theoretical analysis on the integration of its components. For example, a discussion on why using LLMs can improve image clustering, compared to previous studies, would provide valuable insights”
>
> > **A4:** We agree with your suggestion that a deep dive will further strengthen our work. Following your suggestion, we have included a dedicated section in `Appendix P` to elaborate on the core working principle of our system. A summary of this discussion is provided below.
>
> > **Why LLMs Improve Image Clustering:**
> >
> > A nice aspect of this work lies in our framework’s ability to translate a large volume of unstructured images into natural language and leverage the powerful text understanding and summarization capabilities of LLMs to address the challenging Semantic Multiple Clustering (SMC) task. This concept draws inspiration from the use of LLMs in the **Topic Discovery task** in the NLP domain [1]. Our core motivation is: *If LLMs can discover topics from documents and organize them, then by converting images into text, we can similarly employ LLMs to organize unstructured images.*
> >
> > Traditional clustering methods often rely on pre-defined criteria, pre-set numbers of clusters, fixed feature representations (which require training), and are typically not interpretable. These limitations reduce their applicability to diverse datasets in open-world scenarios, as they require significant human priors and retraining for each new dataset.
> > In contrast, LLMs excel at understanding, summarizing, and reasoning over high-level semantics expressed in natural language across diverse knowledge domains (e.g., everyday content, cultural content, or even medical content). They operate in a zero-shot, interpretable manner, making them uniquely suited to the SMC task. The goal of SMC is to discover meaningful and interpretable clustering criteria without requiring prior knowledge or training.
> > By integrating LLMs and MLLMs into the carefully designed TeDeSC framework, we enable the discovery and refinement of criteria directly from the dataset's content, followed by automatic dataset grouping. This allows our framework to overcome the rigid assumptions of traditional clustering methods, making it **automatic, generalizable, and training-free.** It offers a novel perspective on how clustering tasks can evolve beyond traditional paradigms.
>
> > **Integration of LLMs in Our Framework:** we have provided further discussions on the integration of system components and their comparison with previous studies in **Appendices A and C** in the initial submission.
> >
> > - **Criteria Proposer**: Using LLMs to discover diverse and interpretable grouping criteria by summarizing and reasoning over dataset captions.
> > - **Criteria Pool Refinement**: Leveraging LLMs as intelligent text processors to unify, clean, and refine the initial set of criteria.
> > - **Semantic Grouper**: Grouping images based on captions using LLMs, which outperforms Groupers relying on VQA (e.g., BLIP-2) or tagging models (e.g., CLIP). This is because VQA models may suffer from weaker instruction following, while tagging models are typically object-centric and less flexible.
> >
> > Together, these components form a cohesive pipeline that is both generalizable across diverse datasets and applications and capable of addressing the unique challenges of SMC.
>
> [1] Eklund, A., & Forsman, M. Topic modeling by clustering language model embeddings: Human validation on an industry dataset. In *EMNLP: Industry Track*, 2022.

---

> ### Author Response · Authors · 2024-11-20
> **Response to Reviewer J84w (5/5)**
>
> > **Q5:** “While the proposed method benefits from these mitigated biases by incorporating pre-trained LLMs into its pipeline, its contribution may be less impactful than suggested in Section 5, especially when compared to baseline methods trained from scratch on biased datasets”
>
> > **A5:** Regarding the experiments in `Figure 8` of Section 5, we simply use our TeDeSC framerwork to discover the biases present in CelebA dataset followed by running a well-known debiasing algorithm DRO using the predicted biases. Then we compare this performance with DRO that uses biases discovered by other methods (eg., B2T) and ground-truth biases (DRO+GT). All these models that uses DRO (including ours) have been trained from scratch. They only differ by the biases used by the DRO algorithm. Thus, we believe that the comparison is fair and the DRO+GT should be considered as an upperbound.
> >
> > Further to the bias mitigation experiments in `Figure 8`, we applied our proposed method to real-world, more "wild" datasets in Section 5:
> > 1. Novel Bias Detection in Text-to-Image Diffusion Models: **Detecting novel biases in images generated by SDXL and DALL-E3**.
> > 2. Social Media Image Popularity Analysis: **Analyzing Flickr images from the SPID dataset (a non-mainstream dataset collected in 2019)**.
> >
> > We believe that TeDeSC does not rely solely on the *bias mitigated pre-trained LLMs* because our method successfully uncovered *uncommon* biases in DALL-E3 generated images that are not generally flagged by human annotators, such as "Hair style" and "Grooming," associated with the Nurse and Teacher professions (see `Figure 9`). Additionally, our method uncovered *not safe for work* (NSFW) images in the SPID dataset, as highlighted in `Figure 10`, which had not been identified in any prior literature. These application studies demonstrate our method's generalizability to more diverse datasets and uncommon biases, and does not rely solely on the bias mitigated pre-trained LLMs.
>
> ---
>
> We really enjoyed discussing the above points with you. Your insightful suggestions have been helpful in strengthening our paper, and we hope our responses have thoroughly addressed your concerns. We hope this might encourage you to reconsider your evaluation of our work.
>
> If you have any additional questions or need further clarification, please don’t hesitate to reach out. We would be more than happy to engage in further discussion with you. Thanks you!
>
>
> Authors of Submission #1890

---

> ### Author Response · Authors · 2024-11-23
> **Looking Forward to Your Feedback – Thank You from the Authors**
>
> Dear Reviewer J84w,
>
> Thank you once again for your valuable feedback on our work. We deeply appreciate the opportunity to engage in a meaningful scientific discussion about this work with you.
>
> As the Author-Reviewer Discussion Phase nears its conclusion, we kindly ask if our rebuttal and the revised paper sufficiently addresses the concerns and questions raised in your initial review? Your insights are highly valuable to us, and we are eager to clarify any remaining points or provide additional explanations if needed.
>
> This work represents seven months of dedicated effort, and with this submission, we aspire to share its exploration, methodology, results, and applications with the community in a comprehensive way. We will also open-source the code.
>
> Your insights are  essential to us, and we wholeheartedly hope for the chance to address any remaining concerns you may have.
>
> Thank you for your time and consideration. We look forward to your reply!
>
> Sincerely,
> Authors of Submission #1890

---

### Official Review · Reviewer_UE1x · 2024-11-10

**Soundness:** 1
**Presentation:** 2
**Contribution:** 2
**Rating:** 3
**Confidence:** 3

**Summary:**

The paper proposes a LLM-based method (the "TeDeSC" method) to address a proposed variant of the "semantic multiple clustering" problem, where both the clustering criteria and the groupings within each criterion are not known a priori and have to be generated with respect to the given data set.

Given a set of images, the proposed TeDeSC (Text Driven Semantic Multiple Clustering) method leverages the ability of LLMs and MLLMs to generate the clustering criteria for the images and the groupings in each criterion.  For generating the clustering criteria, multi-modal LLMs (MLLMs) are used first to generate a caption for each image, and then the generated captions are fed to LLMs to generate a set of criteria that are relevant to the given images.  For generating the groupings within a criterion C, each image is first fed to a MLLM to generate a caption, conditioned on the criterion C; then, the captions of all of the images will go through a three-step process using LLMs to assign to each image a triplet of group labels, at three levels of semantic granularity.

Along the development of the proposed method, previously existent benchmark data sets are expanded with richer ground-truth clustering criteria and groupings.  The expanded ground-truth clustering criteria and groupings provide better coverage over the many valid grouping options in collections of images with high scene complexity.

**Strengths:**

(S1) Besides the main proposed approach "TeDeSC", the paper also describes other interesting approaches of using LLMs to build the "criteria proposer" and the "semantic grouper".  The descriptions of those other approaches set a nice context to highlight and differentiate the main features of the proposed "TeDeSC" approach.

(S2) The three applications described in Section 5 are quite motivating.  Those applications illustrate how an approach for "semantic multiple clustering" (such as the proposed "TeDeSC" method) can be used in discovering spurious correlations and human bias in data, and in data understanding.  However, I feel that the paper should focus more on describing the details of the TeDeSC method, rather than using two pages to describe the three applications.

(S3) The new benchmark data sets produced in this work may be interesting for future work.

**Weaknesses:**

(W1) The paper puts too much content in the appendix.  Much of that content should be in the main text to make the paper self-contained.  For example, the paper can be more readable if it has more descriptions on the proposed approach, on the evaluation metrics and methodology, etc.

(W2) More descriptions are needed about the proposed methods.

(W3) More descriptions are needed about the evaluation metrics and methodology.

(W4) The figures are too small (for example, Figures 4, 5, and 6).

(W5) The definition of the proposed variant of the "semantic multiple clustering" problem seems to be not fully defined.  The definition doesn't specify the objective that the generated clusters should achieve (e.g., optimal with respect to some notions, etc.).

**Questions:**

(Q1) In Section 3.1, "Caption-based Proposer": For each subset of captions, the proposed method prompts a LLM to elicit a set of partitioning criteria.  How consistent are the sets of partitioning criteria elicited from different subsets of captions?  Would a LLM generate criteria using different English words which are semantically the same?  How does the proposed method accumulate the subsets of captions and consolidate partitioning criteria that are semantically similar?

(Q2) In Section 3.2, "Caption-based Grouper": How diverse are the initial set of class names that are generated and assigned to the images?  How diverse are the output of the "multi-granularity cluster refinement"?  By the way, can we show some experimental data (perhaps in the "Experiments" section) about the number of images in the data sets and the number of distinct groups at each granularity level?

(Q3) In Section 4, "Evaluation metrics for Criteria Discovery": In the formula of TPR, how does it compute the intersection of the set of generated criteria (the set R) and the ground-truth criteria (the set Y)?  How does a "criterion token" in the set R match with a "criterion token" in the ground-truth set Y?  Do the criteria tokens in the set R and the set Y come from a small superset of common tokens?

(Q4) In Section 4, "Evaluation metrics for substructure uncovering": Can we formulate the evaluation method of "Semantic Consistency" in a math formula?  For example, it will be helpful to show the math formula of the Semantic Accuracy (SAcc) metric.

(Q5) Figure 4:  Can we have some descriptions about how to read the figures in Figure 4?

(Q6) Figure 5:  Can we have some descriptions about how to read the figures in Figure 5?

(Q7) In Section 4.2, Table 2, "Comparison with criterion-conditioned clustering methods": How does the computation of CAcc(%) and SAcc(%) done, if the TeDeSC method generates more groupings than the ground-truth groupings?  Does having more groupings than the ground-truth groupings boost the value of CAcc(%) and SAcc(%)?

---

> ### Author Response · Authors · 2024-11-20
> **Response to Reviewer UE1x (1/6)**
>
> Thank you for your constructive feedback. We greatly appreciate your acknowledgment of the novelty of our proposed TeDeSC framework and your recognition of the additional variants we developed for the new Semantic Multiple Clustering task as "interesting." We find it encouraging that you found the three application studies "motivating" and our newly proposed benchmarks "interesting for future work." Below, we address your questions and suggestions point by point. We have incorporated your suggestions into the revised paper.

---

> ### Author Response · Authors · 2024-11-20
> **Response to Reviewer UE1x (2/6)**
>
> > **Q1:** “The paper puts too much content in the appendix. Much of that content should be in the main text to make the paper self-contained. For example, the paper can be more readable if it has more descriptions on the proposed approach, on the evaluation metrics and methodology, etc. More descriptions are needed about the proposed methods. More descriptions are needed about the evaluation metrics and methodology.”
>
> > **A1:** Thank you for your suggestions. In this work we have additionally proposed a new task (Semantic Multiple Clustering), a new framework (TeDeSC), two new benchmarks (COCO-4c and Food-4c), experimented on six benchmarks across 19 criteria and demonstrated three applications, which severely contrains the amount of available space for a over detailed methodology and metrics description. Thus, in the main paper we had to make a trade-off between the comprehensiveness of the paper as a whole and thoroughness of each section.
> >
> > Yet, we believe that we have included all necessary information to make the paper self-contained. For instance, we have used commonly used metrics in prior literature: TPR (equivalent to Recall), Clustering Accuracy [2], and Semantic Accuracy [1]. Given these are standard metrics, we did not expand upon their details in the main paper. However, following your suggestion, we have included appropriate references to the Appendix (eg., detailed prompts, detailed metrics) in each section of the main paper for improved navigation. We hope that this will sufficiently improve the readability of the paper and yet allow us to comprehensively discuss all the important aspects of our studied problem in the main paper. Note that the reviewers`Reviewer J84w`, `Reviewer Npux` and `Reviewer rRnH` all appreciated that our manuscript is comprehensive, clear, well-structured and contains significant application studies.
> >
> > Finally the Appendix is extensive because we wanted to include all the details related to the benchmarks, implementation details, full prompts, failure case analysis, user study statistics, clustering results visualization, and so on. We firmly believe that the extensiveness simply asserts the comprehensiveness of the work and promotes reproducibility.
>
> ---
>
> > **Q2:** “How consistent are the sets of partitioning criteria elicited from different subsets of captions? Would a LLM generate criteria using different English words which are semantically the same? How does the proposed method accumulate the subsets of captions and consolidate partitioning criteria that are semantically similar?”
>
> > **A2:** You are correct that the batching mechanism indeed influences the discovered partitioning criteria. Note that batching of captions in Criteria Proposal was adopted due to the limitation of the LLM context window (128k). **To tackle the noisy criteria proposal across batches we designed the Criteria Refinement step (Sec. 3.1).** The refinement step consolidates the semantically similar criteria and discards the noisy ones, making the final criteria consistent and coherent with the image collections.
> >
> > In detail, the LLM can indeed generate and accumulate criteria in the initial criteria pool $\tilde{\mathcal{R}}$ containing overlaps or semantically similar criteria expressed differently in English (e.g., "Color Scheme" vs. "Primary Colors", or other synonyms). This issue is dealt by the Criteria Refinement step by feeding all the initial criteria to a LMM and then prompting the LLM to refine the noisy, overlapping criteria into a final, and unique criteria set $\mathcal{R}$ (details of this prompt can be found in `Table 11`).
>
> [1] Conti, A., Fini, E., Mancini, M., Rota, P., Wang, Y., & Ricci, E. Vocabulary-free image classification. In NeurIPS, 2023.
>
> [2] Han, K., Rebuffi, S. A., Ehrhardt, S., Vedaldi, A., & Zisserman, A. Autonovel: Automatically discovering and learning novel visual categories. IEEE TPAMI, 2021.

---

> > ### Comment · Reviewer_UE1x · 2024-12-02
> >
> > (Follow-up on A1)
> > I personally prefer reading a paper that gives more space to describe the insights on the proposed method.
> > For example, I like the prompting technique of generating the "criterion-specific captions", which is described in the appendix (from line 1612 to line 1615, and in Tables 16 and 17).  I don't know whether such a technique is a common practice and therefore is not worth highlighting in the main text.  I feel that putting content like those in the main text can make me more easily appreciate the innovation of the proposed method.  In the current manuscript which has 39 pages of appendix materials, I wonder how easy a reader can find those interesting information in the appendix.
> >
> > Meanwhile, it will be even better if there are discussions about authors' experiences with the "criterion-specific captions", to really understand the advantages of the technique.

---

> > > ### Author Response · Authors · 2024-12-02
> > > **Author Response to the Follow-Up Question 2 from Reviewer UE1x**
> > >
> > > > **Q2.1:** About prompt details in the appendix.
> > >
> > > **A2.1:** Thank you for recognizing our novel prompting technique for generating "criterion-specific captions," designed to capture only criterion-related content in each image. We introduced this in Lines 245-246 of the Method section, with detailed prompts provided in the Appendix (Tables 16 and 17).
> > >
> > > The exact prompts are included in the Appendix *because the key innovation lies in the strategy of employing criterion-specific captions*, *not* in the prompts themselves. Since the prompt design space is flexible and multiple designs can achieve similar functionality, we provided detailed discussions in the Appendix. Moreover, space constraints in the main text also influenced this decision, as we introduced two additional novel baseline methods (image-based and tag-based) for the Proposer and Grouper, resulting in a large set of prompts (Tables 8-19), making it impractical to include all details in the main text. Thus, we provided comprehensive prompt details and discussions in the Appendix to balance clarity and depth, while ensuring reproducibility. This allows the main text to focus on component-wise solutions for the SMC task.
> > >
> > >
> > > ---
> > >
> > >
> > > > **Q2.2:** "In the current manuscript which has 39 pages of appendix materials, I wonder how easy a reader can find those interesting information in the appendix."
> > >
> > > **A2.2:** To help readers easily locate the prompt details, we **provided `hyperlinks`** in the system overview **`Figure 3`** (upper part, Line 162) that **direct** them to the corresponding sections in the appendix for the Criteria Proposer and Semantic Grouper. Additionally, we emphasized and linked the locations of these details throughout the Method section (Lines 185, 192, 230). We believe these thoughtful designs enable readers to easily navigate the paper and find the relevant information.
> > >
> > >
> > > ---
> > >
> > > > **Q2.3:** "Meanwhile, it will be even better if there are discussions about authors' experiences with the 'criterion-specific captions,' to really understand the advantages of the technique."
> > >
> > > **A2.3:** Thank you for the suggestion. The advantages of our "criterion-specific captions" technique are twofold:
> > >
> > > 1. It can capture the visual information *most relevant* to the target criterion by conditioning the MLLM. Compared to general captions (e.g., "Describe this image in detail"), it ensures that criterion-specific information is captured while reducing noisy, irrelevant details, which improves the subsequent semantic grouping step.
> > >
> > > 2. It can help *mitigate inherent biases* in MLLMs by conditioning the model to focus only on criterion-related content, minimizing the influence of unrelated or biased information (e.g., gender). We studied this effect in a fair clustering experiment presented in `Appendix N (Line 2900, Table 34)` of the revised paper.
> > >
> > > We will include this discussion in the camera-ready version of the paper.

---

> ### Author Response · Authors · 2024-11-20
> **Response to Reviewer UE1x (3/6)**
>
> > **Q3:** “How diverse are the initial set of class names that are generated and assigned to the images? How diverse are the output of the "multi-granularity cluster refinement"? By the way, can we show some experimental data (perhaps in the "Experiments" section) about the number of images in the data sets and the number of distinct groups at each granularity level?”
>
> > **A3:** **Diversity of class names before and after multi-granularity cluster refinement**: We provide tables below that report the number of: (i) ground-truth cluster names (GT), (ii) initially assigned cluster names (Pred-Init), and (iii) refined cluster names at each granularity level (Coarse: Pred-L1, Middle: Pred-L2, Fine: Pred-L3), for each dataset and each criterion. As observed, the multi-granularity cluster refinement step effectively de-noises and consolidates the initial cluster names into well-defined granularity levels, meeting varying user needs. **Following your suggestion, we have included these additional experimental results of the clustering in Table 28 (Appendix E.2, Line#1890)** of the revised paper, where we provide expanded numerical results for the comparison of semantic groupers.
>
>
> | COCO-4c       | GT  | Pred-Init | Pred-L1 | Pred-L2 | Pred-L3 |
> |---------------|------|------|------|------|------|
> | Activity      | 64   | 203 | 12   | 23   | 52   |
> | Location      | 19   | 145  | 7    | 14   | 28   |
> | Mood          | 20   | 122 | 15   | 25   | 30   |
> | Time of day   | 6    | 96  | 2    | 8   | 31   |
>
> | Food-4c     | GT  | Pred-Init | Pred-L1 | Pred-L2 | Pred-L3 |
> |-------------|------|------|------|------|------|
> | Food type   | 101  | 301  | 7    | 37   |  127  |
> | Cuisine     | 15   | 141  | 9    | 18   | 53   |
> | Course      | 5    | 97   | 4    | 12   | 78   |
> | Diet        | 4    | 139  | 5    | 8    | 64   |
>
> | Action-3c   | GT  | Pred-Init | Pred-L1 | Pred-L2 | Pred-L3 |
> |-------------|------|------|------|------|------|
> | Action      | 40   | 71   | 8    | 15   | 51   |
> | Location    | 10   | 82   | 5    | 10   | 67   |
> | Mood        | 4    | 95   | 6    | 18   | 55   |
>
> | Clevr-4c    | GT  | Pred-Init | Pred-L1 | Pred-L2 | Pred-L3 |
> |-------------|------|------|------|------|------|
> | Color       | 10   | 25   | 6    | 12   | 17   |
> | Texture     | 10   | 23   | 2    | 5    | 12   |
> | Shape       | 10   | 22   | 5    | 11   | 14   |
> | Count       | 10   | 11   | 2    | 4    | 11   |
>
> | Card-2c     | GT  | Pred-Init | Pred-L1 | Pred-L2 | Pred-L3 |
> |-------------|------|------|------|------|------|
> | Rank        | 14   | 147  | 4    | 7    | 16   |
> | Suit        | 5    | 56   | 4    | 7    | 30   |
>
> | Fruit-2c    | GT  | Pred-Init | Pred-L1 | Pred-L2 | Pred-L3 |
> |-------------|------|------|------|------|------|
> | Species     | 34   | 54   | 8    | 25   | 38   |
> | Color       | 15   | 66   | 5    | 15   | 39   |
>
> > In **Appendix B**, we have already provided a summary of the ground-truth number of classes (clusters) in `Table 3` across the six benchmarks, along with the exact annotated cluster names in `Tables 4–5`. **Following your suggestion, we have added the number of images (samples) in each dataset to `Table 3` (Appendix B.1)** in the revised paper to provide a clearer overview of the dataset statistics.
>
> | Dataset | Number of Images |
> | -------- | -------- |
> | COCO-4c     | 5,000     |
> | Food-4c     | 25,250     |
> | Action-3c     | 1,000     |
> | Clevr-4c     | 10,000     |
> | Card-2c     | 8,029     |
> | Fruit-4c     | 103     |

---

> ### Author Response · Authors · 2024-11-20
> **Response to Reviewer UE1x (4/6)**
>
> > **Q4:** “Can we formulate the evaluation method of "Semantic Consistency" in a math formula?”
>
> > **A4:** Yes. **Semantic Accuracy (SAcc) formulation**: Under the $l$-th target criteria $R_l$, for each image $\mathbf{x}_n \in \mathcal{D}$, we compute the semantic similarity $s_n$ between its assigned cluster name $\mathbf{p}_l \in \mathcal{P}_l$ and the ground-truth label $\mathbf{c}_l \in \mathcal{C}_l$ under the current criterion $R_l$ as $\langle \mathcal{E}(\mathbf{p}_l), \mathcal{E}(\mathbf{c}_l) \rangle$, where $\mathcal{E}$ is the Sentence-BERT encoder and $\langle \cdot, \cdot \rangle$ represents the cosine similarity function.
> >
> > The average similarity across all the $N=|\mathcal{D}|$ images in the dataset is reported as **SAcc**$=\frac{1}{N}\sum_{n=1}^{N} s_n$, reflecting how well the predicted substructure semantically aligns with the ground-truth.
> >
> > As the Semantic Accuracy (SAcc) used for the "Semantic Consistency" evaluation is a well-formulated evaluation metric introduced in [1], we did not elaborate on it in detail due to space constraints in the main paper. **Following your suggestion, we have included the formulation in Section 4 of the revised main paper (Line#278-280).**
>
> ---
>
> > **Q5:** “How does the computation of CAcc(%) and SAcc(%) done, if the TeDeSC method generates more groupings than the ground-truth groupings? Does having more groupings than the ground-truth groupings boost the value of CAcc(%) and SAcc(%)?”
>
> > **A5:** Having more groupings (or clusters) *does not* boost the value of CAcc(%) or SAcc(%). Given a testing dataset, the metrics considered can only be maximized when the number of predicted clusters is *identical* to the number of ground-truth (GT) clusters, as shown previously in [3]. Unlike the competitors (i.e., IC|TC and MMaP) reported in `Table 2` of the main paper, our method **neither assumes nor uses** the GT number of clusters. We explain below why this is the case.
> >
> > To recap, Clustering Accuracy (CAcc) (defined in [2]) is evaluated by running the Hungarian Matching algorithm [4] to find the optimal assignment between the predicted cluster indices and GT label indices. As carefully studied and discussed in the GCD [3] literature:
> > - If the number of predicted clusters is higher than the total number of GT clusters, the extra clusters -- not matched by Hungarian algorithm -- are assigned to the null set, and will subsequently be predicted as misclassification. Below is an illustrative example for such a scenario. Lets say there are 2 GT clusters of 20 images in each and the model predicts (denoted as Pred) 4 clusters with the following prediction distribution, will result in a CAcc of 50%:
> >
> > | Model predictions | Number of images | GT Cluster #1 | GT Cluster #2 |
> > | -------- | -------- | -------- | -------- |
> > | Pred as cluster #1 (matched to GT #1) | 10  | 10 (Correct)     | -     |
> > | Pred as cluster #2 (matched to GT #2) | 10  | -     | 10 (Correct)     |
> > | Pred as cluster #3 (null cluster or no match) | 10   | -     | -   |
> > | Pred as cluster #4 (null cluster or no match) |10   | -   | -    |
> > | **Total correct predictions (20/40), CAcc = 50%** |-  | -   | -    |
>
> > - Conversely, if the number of predicted clusters is *lower* than the number of GT clusters, extra classes are assigned to the null cluster, and all instances with those ground-truth labels are considered incorrect.
> >
> > Thus, if the predicted number of clusters is higher or lower with respect to the GT clusters, this will lead to sub-optimal CAcc on the evaluation dataset. Only when the number of predicted clusters matches the number of GT clusters, the CAcc can be maximized, under the assumption that the model is predicting well. Thank you for raising this point. We have added this discussion in Appendix Q to further clarify our evaluation methods.
>
>
> [1] Conti, A., Fini, E., Mancini, M., Rota, P., Wang, Y., & Ricci, E. Vocabulary-free image classification. In *NeurIPS*, 2023.
>
> [2] Han, K., Rebuffi, S. A., Ehrhardt, S., Vedaldi, A., & Zisserman, A. Autonovel: Automatically discovering and learning novel visual categories. IEEE *TPAMI*, 2021.
>
> [3] Vaze, S., Han, K., Vedaldi, A., & Zisserman, A. Generalized category discovery. In *CVPR*, 2022.
>
> [4] Harold W Kuhn. The hungarian method for the assignment problem. *Naval research logistics quarterly*, 1955.

---

> > ### Comment · Reviewer_UE1x · 2024-12-02
> >
> > (Follow-up on A5)
> > Thank you for the explanations.
> > It is very useful for me to understand that a method will have a better CAcc metric value if the number of clusters (groups) is close to that of the ground truth.
> >
> > In lines 286 to 289, it is mentioned that this study uses the best CAcc values among the three granularities (coarse, medium, and fine), and this is described as a fair evaluation strategy, as the existing methods rely on knowing the number of ground-truth clusters.  I can see that this is a choice that the proposed method made about what information to take from the ground truth.  I can feel that this choice (or tradeoff) may have advantages in real world use cases (such as in the applications outlined in the section 5).  However, I feel that this trade-off on what information to take from the ground truth, can be made more upfront in the paper (for example, around line 88, when the paper talks about the flexibility of the proposed method), to better understand the merits of the proposed method.

---

> > > ### Author Response · Authors · 2024-12-02
> > > **Author Response to the Follow-Up Question 3 from Reviewer UE1x**
> > >
> > > > **Q3:** "I feel that this trade-off on what information to take from the ground truth, can be made more upfront in the paper (for example, around line 88, when the paper talks about the flexibility of the proposed method), to better understand the merits of the proposed method."
> > >
> > > **A3:** We are glad that you found our explanations on CAcc useful. Yes, you are correct. This choice indeed offers advantages in real-world use cases: because our method and its outputs are *human-interpretable*, users can flexibly select their preferred granularity in practical applications. We designed the evaluation strategy with these real-world scenarios in mind, as the ground truth can be seen as representing a user's (annotator's) preference.
> > >
> > > Thank you for the suggestion! We will incorporate a discussion of this design choice after Line 88, alongside the description of the flexibility of our method, to make this merit of the proposed method more clear in the camera-ready version of the paper.

---

> ### Author Response · Authors · 2024-11-20
> **Response to Reviewer UE1x (5/6)**
>
> > **Q6:** “In the formula of TPR, how does it compute the intersection of the set of generated criteria (the set R) and the ground-truth criteria (the set Y)? How does a "criterion token" in the set R match with a "criterion token" in the ground-truth set Y? Do the criteria tokens in the set R and the set Y come from a small superset of common tokens? ”
>
> > **A6:** We do not have the concept of a "criterion token" in the paper. We are assuming that, by "criterion token," you are referring to the criterion text. For TPR computation, we follow the approach described in [5]. To compute the intersection between the generated set of criteria $\mathcal{R}$ and the ground-truth set $\mathcal{Y}$, we use exact text matching, as both criteria in $\mathcal{R}$ and $\mathcal{Y}$ are expressed in natural language. A criterion in $\mathcal{R}$ is considered a match with a criterion in $\mathcal{Y}$ only if the text strings are identical. This approach ensures an unambiguous and straightforward comparison between the generated and ground-truth criteria.
> >
> > Additionally, the criteria in $\mathcal{R}$ and $\mathcal{Y}$ are not drawn from a predefined superset. Instead, they are open-ended and generated independently by the method and the annotators. While this makes exact matching not straightforward, it ensures that only precisely matching criteria are counted as true positives. Although exact text matching is stricter than semantic matching, it aligns with the open-ended nature of the task and the goal of maintaining high precision in evaluation. Using exact matching also avoids introducing subjectivity or ambiguity into the evaluation process, which could arise with semantic similarity measures.
>
> [5] Csurka, G., Hayes, T. L., Larlus, D., & Volpi, R. What could go wrong? Discovering and describing failure modes in computer vision. In *ECCV Workshop*, 2024.
>
> ---
>
> > **Q7:** “The definition of the proposed variant of the "semantic multiple clustering" problem seems to be not fully defined. The definition doesn't specify the objective that the generated clusters should achieve (e.g., optimal with respect to some notions, etc.).”
>
> > **A7:** The underlying definition of Semantic Multiple Clustering (SMC) remains the same as Multiple Clustering (MC) [6], which is clustering a dataset into semantic clusters under different grouping themes (or criteria). Differently from MC, the proposed SMC assumes no knowledge of the criteria and the number of clusters, making the task more open-ended in nature [7]. We have summarized the key differences of the task definitions in `Table 1` of the main paper.
> > Given the SMC task is open-ended (e.g., the number of valid grouping criteria is subjective and inherently unlimited), we consider the user-preferred groupings, only at the evaluation phase, as the "optimal" results. In relation to the user-preferred groupings, we specify the objectives that an SMC system should meet in Section 4:
> > 1. **Objectives of Criteria Discovery (Line#265–274)**: The predicted criteria should cover as many valid and distinct criteria as possible. This is evaluated using True Positive Rate (TPR) and Diversity score with respect to the ground-truth (user-preferred) criteria set.
> > 2. **Objectives of Substructure Uncovering (Line#275–285)**: For each criterion, the predicted substructure should correctly organize images that share a common semantic meaning related to the criterion into interpretable clusters, where the results should structurally and semantically align well with the ground-truth (user-preferred) groupings. Thus, the quality of structural alignment is evaluated using Clustering Accuracy (CAcc), and the quality of semantic alignment is evaluated using Semantic Accuracy (SAcc).
> >
> > By defining these objectives and linking them to our evaluation metrics, we provide a clear and measurable framework for assessing the performance of an open-ended SMC system.
>
> [6] Yu, G., Ren, L., Wang, J., Domeniconi, C., & Zhang, X. (2024). Multiple clusterings: Recent advances and perspectives. Computer Science Review, 52, 100621.
>
> [7] Lin, C., Jiang, Y., Qu, L., Yuan, Z., & Cai, J. (2024). Generative Region-Language Pretraining for Open-Ended Object Detection. In Proceedings of the IEEE/CVF Conference on Computer Vision and Pattern Recognition (pp. 13958-13968).

---

> ### Author Response · Authors · 2024-11-20
> **Response to Reviewer UE1x (6/6)**
>
> > **Q8:** “The figures are too small (for example, Figures 4, 5, and 6). Can we have some descriptions about how to read the figures?”
>
> > **A8:** Thanks for your feedback. Given the multi-faceted nature of the newly introduced task SMC that requires to be evaluated under various dimensions -- comprehensiveness and diversity of the criteria (TPR and diversity score) and the quality of the semantic substructure (CAcc and SAcc) -- across 6 benchmarks, we resorted to advanced yet compact plots to showcase the results. However, in the revised paper, **following your suggestion we have now significantly improved the clarity and size of Figures 4, 5, 6,** added instructions in the captions on how to interpret the figures and further added detailed breakdowns of the results behind those figures in the Appendix. We have also added cross-references next to each figure to improve the navigation between the main paper and Appendix.
>
> ---
>
> We greatly appreciate your constructive feedback on our work. We hope our detailed responses in the rebuttal, along with the revised version of the paper, have clarified all of your concerns and will encourage you to reconsider your evaluation of our work.
>
> If you have any further questions or require additional clarification, please don’t hesitate to reach out. We would be delighted to engage in further discussion and provide any additional information you may need. Thank you once again for your time and effort in reviewing our work!
>
> Authors of Submission #1890

---

> ### Author Response · Authors · 2024-11-23
> **Looking Forward to Your Feedback – Thank You from the Authors**
>
> Dear Reviewer UE1x,
>
> Thank you once again for your valuable feedback on our work. We deeply appreciate the opportunity to engage in a meaningful scientific discussion about this work with you.
>
> As the Author-Reviewer Discussion Phase nears its conclusion, we kindly ask if our rebuttal and the revised paper sufficiently addresse the concerns and questions raised in your initial review? Your insights are highly valuable to us, and we are eager to clarify any remaining points or provide additional explanations if needed.
>
> This work represents seven months of dedicated effort, and with this submission, we aspire to share its exploration, methodology, results, and applications with the community in a comprehensive way. We will also open-source the code.
>
> Your insights are  essential to us, and we wholeheartedly hope for the chance to address any remaining concerns you may have.
>
> Thank you for your time and consideration. We look forward to your reply!
>
> Sincerely,
> Authors of Submission #1890

---

> ### Comment · Reviewer_UE1x · 2024-12-02
>
> (Follow-up on A2)
> Thank you for the explanations.
> So, the refinement step is also done by LLMs with a prompt.  I see (in Table 11) that the prompt specifies a goal explanation and a task instruction:
> * Goal Explanation: My goal is to refine this list by merging similar criteria and rephrasing them using more precise and informative terms. This will help create a set of distinct, optimized clustering criteria.
> * Task Instruction: Your task is to first review and understand the initial list of clustering criteria provided. Then, assist me in refining this list by:
>    * Merging similar criteria.
>    * Expressing each criterion more clearly and informatively.
>
> I wonder whether the number of criteria in the refined set is still relatively large.
> Looking at the formula of TPR (line 266), I think that a method that outputs a larger set of criteria (the set \mathcal{R}) has an advantage in scoring higher in TPR.  The TPR metric does not penalize a large set of criteria that may contain irrelevant texts.
>
> (A2_Q1) Does the Caption-based proposer achieve better TPR metric scores due to its larger output set of criteria?  Is there a table in the paper that reports the number of refined criteria?
> (A2_Q2) Are there "guardrails" used in preventing the generation of irrelevant criteria in the output set of refined criteria?

---

> > ### Author Response · Authors · 2024-12-02
> > **Author Response to the Follow-Up Question 1 from Reviewer UE1x**
> >
> > > **Q1.1:** "(A2_Q1) Does the Caption-based proposer achieve better TPR metric scores due to its larger output set of criteria? Is there a table in the paper that reports the number of refined criteria?"
> >
> > **A1.1:** The Caption-based Proposer achieves better TPR **not because** it outputs a larger set of criteria. The `Table below` shows the number of proposed criteria generated by each Proposer across six datasets (Note that all Proposers conducted the Criteria Refinement step). As we can observe, the **Caption-based Proposer**, which achieves the best TPR, **often produces a smaller and more precise set of criteria**. In contrast, **the Image-based and Tag-based Proposers**, which underperform in TPR, **typically generate a larger set of criteria that might include more noise.**
> >
> > We will include the `Table below` that reports the number of refined criteria in the camera-ready version of the paper. As the current rebuttal stage does not allow for uploading a revised PDF, this update will be incorporated in the final submission.
> >
> >
> > `Table below:` Number of proposed criteria from different Criteria Proposers. All proposers performed the Criteria Refinement step.
> > |  | Image-based Proposer | Tag-based Proposer | Caption-based Proposer |
> > | -------- | -------- | -------- | -------- |
> > | COCO-4c     | 48     | 19     | 15     |
> > | Food-4c     | 28     | 24     | 10     |
> > | Action-3c     | 25     | 15     | 12     |
> > | Clevr-4c     | 8     | 11     | 8     |
> > | Card-2c     | 43     | 15     | 8     |
> > | Fruit-2c     | 6     | 11     | 10     |
> >
> > ---
> >
> > > **Q1.2:** "(A2_Q2) Are there "guardrails" used in preventing the generation of irrelevant criteria in the output set of refined criteria?"
> >
> > **A1.2:** We do *not* have a "guardrails" system to prevent generating irrelevant criteria in the proposed system. The visual information (e.g., captions) translated from images is sent to the LLM as **conditions** to guide the criteria discovery.
> >
> > Furthermore, in Appendix M, we analyzed the effect of irrelevant (invalid) criteria on the proposed system by manually adding invalid criteria, "Action" and "Clothing style," into the proposed criteria set for Fruit-2c dataset. As shown in Table 33, the system automatically assigned nearly all images to a cluster named "Not visible" for these invalid (irrelevant) criteria. Because the system generates interpretable outputs, users can easily identify and disregard such invalid groupings.
> >
> > ---
> >
> > > **Q1.3:** "The TPR metric does not penalize a large set of criteria that may contain irrelevant texts."
> >
> > **A1.3:** As highlighted in Lines 273-274, the number of valid grouping criteria (True Positives) in an unstructured large image collection is often inherently unlimited and subjective, given the open-ended nature of the Semantic Multiple Clustering (SMC) task. *This unlimited nature makes it challenging—even for human annotators—to determine what constitutes irrelevant criteria (False Positives) in the open-ended SMC task.* This difficulty in evaluating irrelevant criteria is evident even in the simple case of four images provided below. Therefore, we proposed TPR as the primary evaluation metric for criteria discovery. This approach shares similar motivation with Object Proposal task [1], where evaluation methods commonly focus on *recall-related metrics* to measure the ability to *recover* relevant criteria.
> >
> > ```
> > Consider a set of 4 images taken in different gyms. How many valid grouping criteria could exist for these images? Potential criteria might include: Interior design style, Lighting color, Floor material, Density of sports equipment, Brand of equipment, Number of windows, Number of people present, Age composition of individuals, Activity type, or even more specific criteria like the Number of people using phones.
> >
> > With such a wide range of valid possibilities, it becomes extremely challenging to determine which criteria are irrelevant. This difficulty highlights the subjective and open-ended nature of identifying False Positives (irrelevant criteria) in the SMC task, even in this toy example.
> > ```
> >
> > [1] Shi, Hengcan, Munawar Hayat, Yicheng Wu, and Jianfei Cai. Proposalclip: Unsupervised open-category object proposal generation via exploiting clip cues. In *CVPR*, 2022.

---

> ### Author Response · Authors · 2024-12-02
> **Author Response to Reviewer UE1x: Thank you for the Follow-up Discussion**
>
> Dear Reviewer UE1x,
>
> Thank you once again for your follow-up questions. Incorporating your suggestions into the revised paper has further strengthened our work.
>
> We hope our responses and the revisions have sufficiently addressed your questions. If so, we would greatly appreciate it if you would reconsider your score for our pioneering work on the novel Semantic Multiple Clustering task, reflecting the discussion.
>
> If you have any additional questions, we would be happy to engage in further discussion during the remaining days of the rebuttal period.
>
> Sincerely,
>
> Authors of Submission #1890

---

### Author Response · Authors · 2024-11-20
**Global Response to All Reviewers**

We sincerely thank all the Reviewers and Area Chairs for their dedicated efforts in reviewing our paper. We greatly appreciate the positive, insightful, and constructive feedback, which we believe has significantly strengthened our paper in its revised form.

We are encouraged that `Reviewers J84w, VuXq, Npux, and rRnH` consistently found our paper to be well-organized, clearly written, and supported by a clear problem statement and informative figures that effectively illustrate the motivation and methodology. We are thrilled that `Reviewer Npux` acknowledged how our effective use of figures and tables enhances understanding. Moreover, we are encouraged by the consistent recognition of our newly proposed task as "interesting and important" (`Reviewer rRnH`) and as having "significant contribution, filling a clear gap in the existing literature" (`Reviewer Npux`). Our proposed framework was acknowledged as "interesting" (`Reviewer UE1x`), "innovative" (`Reviewer Npux`), and "technically sound" (`Reviewer J84w`). Furthermore, we are pleased that the Reviewers consistently recognized our experiments and evaluation as "comprehensive" (`Reviewers Npux and rRnH`) and "well-designed" (`Reviewer J84w`), with the three "motivating" (`Reviewer UE1x`) application studies ("underscore the work's significance across different domains" per `Reviewer Npux`). Lastly, we are encouraged by the observations of `Reviewers UE1x and Npux` that our newly proposed benchmarks are interesting and better reflect real-world complexity.

We have thoroughly addressed all the concerns raised by the reviewers in our individual responses and incorporated their suggestions into the revised paper. *The now updated paper revision incorporates all changes, which are highlighted in Forest Green font.* A summary of the modifications made in response to each reviewer’s feedback is provided below:

* In response to **`Reviewer UE1x`**, we:
  - Improved Figures 4, 5, 6, and 7 by enlarging them, refining their elements, and enhancing captions to better guide readers in interpreting the plots.
  - Provided a formal formulation for the Semantic Accuracy evaluation criterion in Section 4 of the revised main paper.
  - Added the number of images in each benchmark to Table 3 and included the number of predicted clusters at each granularity in Table 28.

* In response to **`Reviewer J84w`**, we:
  - Added a Future Work discussion in Appendix L to explore closed-loop optimization for developing our method further.
  - Included a dedicated section in Appendix P discussing why using LLMs can improve image clustering.

* In response to **`Reviewer VuXq`**, we:
  - Conducted additional experiments and discussions to investigate the effect of invalid "hallucinated" grouping criteria on system performance in Appendix M.
  - Provided additional experiments with an in-depth discussion on the effect of model hallucination and bias on our system in Appendix N.

* In response to **`Reviewer Npux`**, we:
  - Investigated the effect of multi-granularity clustering in Appendix O.
  - Validated our framework’s bias mitigation with additional experiments in Appendix N.
  - Explored the impact of invalid False Positive grouping criteria on system performance in Appendix M.
  - Included a Future Work discussion in Appendix L outlining potential applications to other data types.
  - Provided a detailed computational cost analysis of our method in Appendix K.

* In response to **`Reviewer rRnH`**, we:
  - Added a comprehensive computational cost and time analysis in Appendix K, along with a discussion of how our framework is parallelized.
  - Further clarified the "ill-posed nature" of the task in the Introduction.

Finally, we sincerely hope the ACs and Reviewers will fully consider the following contributions of our work when making the final decision:
1. The proposal of the novel and practical Semantic Multiple Clustering (SMC) task.
2. The introduction of two new benchmarks, COCO-4c and Food-4c, for the newly proposed SMC task.
3. The development of the novel TeDeSC framework for the SMC task, alongside exploring various approaches to building SMC systems.
4. Extensive evaluation of our proposed framework across six benchmarks, covering 19 grouping criteria, to validate its effectiveness.
5. Study of the TeDeSC framework to three diverse, real-world applications, yielding valuable and insightful results.

We sincerely thank the Reviewers and Area Chairs for their time and thoughtful feedback. If you have any additional questions or concerns, please do not hesitate to reach out. We are happy to provide further clarifications.

Best regards,

Authors of Submission #1890

---

> ### Author Response · Authors · 2024-11-27
> **Global Response to All Reviewers - Further Improvements in the Revision**
>
> Thank you once again to all Reviewers and Area Chairs for your time and effort in reviewing our paper.
>
> After thoughtfully revisiting and reflecting on all the valuable suggestions provided, we have made the following additional improvements in the latest uploaded revision:
>
> ---
>
> Main Paper:
> - Added a technical *Challenge Analysis* paragraph in the Introduction section, discussing the challenges and difficulties of the newly proposed SMC task. (`Lines 63–79`)
> - Further *clarified the objective of the SMC task* in the Task Definition section. (`Lines 123–126`)
>
> Appendix:
> - Included more detailed discussions of Multiple Clustering methods in Appendix A. (`Lines 1121–1126`)
> - Added a discussion on the *Challenges of employing LLMs to facilitate the SMC task*. (`Lines 2995–3010`)
>
> ---
>
> These revisions primarily enhance the clarity and description of the newly introduced SMC task.
>
> *Additionally, we wholeheartedly invite the Reviewers to review our individual responses under your review posts, as well as the revised paper, to check whether your concerns have been sufficiently addressed or if further clarification is needed.*
>
> The discussion period has been extended to Dec 2nd, and we welcome any additional questions or feedback. Your insights are invaluable for improving our work, and we sincerely look forward to hearing from you.
>
> Thank you!
>
> Sincerely,
>
> Authors of Submission #1890

---

### Meta-Review · Area_Chair_BzbC · 2024-12-13

**Metareview:**

I have read all the materials of this paper including the manuscript, appendix, comments, and response. Based on collected information from all reviewers and my personal judgment, I can make the recommendation on this paper, reject. No objection from reviewers who participated in the internal discussion was raised against the reject recommendation.

**Research Question**

This paper considers the multiple clustering problem with the extra LLM model.

**Challenge Analysis**

The motivation of this paper is unclear. After communicating with the authors, the authors aim to tackle how to accurately translate visual content from images into text that LLMs can effectively reason with. In the revised version, two specific challenges are outlined: 1) the current VLM cannot effectively identify valid clustering criteria, and 2) no prior knowledge of user-preferred clustering granularity is provided.

**Philosophy**

I did not find any philosophy in this paper to tackle the above challenges.

**Solution**

Instead, the authors directly provided their solutions. The technical part is engineering-like without too much rationality (One reviewer points out that the overall design is primarily empirical.). For example, it is unclear why the proposed method can find the valid clustering criteria. In the introduction, the authors aim to find “all” and “valid” criteria. I do not know if finding “all” valid criteria is possible; moreover, the authors need to define what is a valid criterion. I am also confused about criteria refinement.

**Experiments**

1.	What is the ground truth? It is a single criterion or multiple criteria.
2.	The evaluation metrics mainly rely on the LLM’s prior; thus, the evaluation is self-biased. It is suggested to employ downstream tasks for evaluation. In general, I did not see a solid evaluation process, especially on multiple clustering.
3.	The authors aim to answer which criteria proposer or semantic groups is the best. Again, this is a self-defined question. Why these questions are meaningful or interesting?
4.	I do not think the experimental results can well support the targeted challenges.

**Presentation**

The presentation needs to be further improved (Several reviewers have similar concerns). In the first version, I did not find a clear motivation. For the technical part, a roadmap or overview might be helpful. Moreover, the logic can be enhanced as well.

I understand the authors put tremendous effort into this paper. However, the current version is far below the bar of a top machine learning conference.

**Additional Comments On Reviewer Discussion:**

No objection from reviewers who participated in the internal discussion was raised against the reject recommendation.

---

### Decision · Program_Chairs · 2025-01-22

Reject